# Cholinergic signals preserve haematopoietic stem cell quiescence during regenerative haematopoiesis

Claire Fielding[1,2,3], Andrés García-García [1,2,3], Claudia Korn[1,2,3], Stephen Gadomski[1,2,3,4,5], Zijian Fang[1,2,3], Juan L. Reguera[6], José A. Pérez-Simón [5,6], Berthold Göttgens [1,2] & Simón Méndez-Ferrer [1,2,3,7,8 ✉]

The sympathetic nervous system has been evolutionary selected to respond to stress and activates haematopoietic stem cells via noradrenergic signals. However, the pathways preserving haematopoietic stem cell quiescence and maintenance under proliferative stress remain largely unknown. Here we found that cholinergic signals preserve haematopoietic stem cell quiescence in bone-associated (endosteal) bone marrow niches. Bone marrow cholinergic neural signals increase during stress haematopoiesis and are amplified through cholinergic osteoprogenitors. Lack of cholinergic innervation impairs balanced responses to chemotherapy or irradiation and reduces haematopoietic stem cell quiescence and self-renewal. Cholinergic signals activate α7 nicotinic receptor in bone marrow mesenchymal stromal cells leading to increased CXCL12 expression and haematopoietic stem cell quiescence. Consequently, nicotine exposure increases endosteal haematopoietic stem cell quiescence in vivo and impairs hematopoietic regeneration after haematopoietic stem cell transplantation in mice. In humans, smoking history is associated with delayed normalisation of platelet counts after allogeneic haematopoietic stem cell transplantation. These results suggest that cholinergic signals preserve stem cell quiescence under proliferative stress.

[1] Wellcome-MRC Cambridge Stem Cell Institute, Cambridge CB2 0AW, UK. [2] Department of Hematology, University of Cambridge, Cambridge CB2 0AW, UK. [3] National Health Service Blood and Transplant, Cambridge Biomedical Campus, Cambridge CB2 0AW, UK. [4] Skeletal Biology Section, National Institute of Dental and Craniofacial Research, National Institutes of Health, Department of Health and Human Services, Bethesda, MD 20892, USA. [5] NIH-Oxford-Cambridge Scholars Program in partnership with Medical University of South Carolina, Charleston, SC 29425, USA. [6] Department of Hematology, University Hospital Virgen del Rocio, 41013 Sevilla, Spain. [7] Instituto de Biomedicina de Sevilla (IBiS/CSIC), Universidad de Sevilla, 41013 Seville, Spain. [8] Departamento de Fisiología Médica y Biofísica, Universidad de Sevilla, 41009 Seville, Spain. ✉email: sm2116@cam.ac.uk

Haematopoietic stem cell transplantation (HSCT) is routinely performed to regenerate the haematopoietic and immune systems of patients with cancer, immune or metabolic disorders. However, haematopoietic stem cell (HSC) responses are heterogeneous, but the underlying mechanisms are not yet clear. HSCs have reduced quiescence and cannot self-renew in peripheral blood, compared with bone marrow (BM). Sympathetic noradrenergic fibres innervate the BM[1] and regulate traffic and activation of HSCs and leucocytes[2–5]. Additionally, stress-induced noradrenergic activity increases HSC proliferation[6]. However, preserving HSC quiescence is critical to prevent HSC attrition, but the mechanisms regulating HSC quiescence under stress are understudied[7,8]. Cholinergic signals cooperate with noradrenergic signals to regulate HSC and leucocyte migration[9]. However, whether cholinergic signals regulate HSC proliferation has not been described. Noradrenaline and acetylcholine (ACh) are generally postganglionic neurotransmitters of (noradrenergic) sympathetic nervous system (SNS) and (cholinergic) parasympathetic nervous system, respectively. However, some sympathetic neurons convert to cholinergic phenotype during postnatal development in sweat glands and the periosteum[10]. Furthermore, the function of sympathetic cholinergic fibres in bone has remained unknown[11].

Here, we show the source and function of cholinergic signals during regenerative haematopoiesis. Cholinergic-neural signals activate bone-associated nestin+ BM mesenchymal stem cells (BMSCs) and regulate HSC quiescence locally in endosteal BM niches. These results illustrate the regulation of stem cell quiescence by the cholinergic system. This mechanism seems to allow stem cells to meet physiological demands and respond to stress, without losing potency.

## Results

**Cholinergic expansion during stress haematopoiesis.** The autonomic nervous system has been evolutionarily selected to efficiently respond to stress. Therefore, we have studied the role of sympathetic cholinergic signals during stress haematopoiesis. WT mice underwent BM transplantation (BMT) following lethal irradiation. BM ACh concentration transiently increased 2 weeks after BMT (Fig. 1a), suggesting a role for cholinergic signals during stress haematopoiesis. To investigate the source of BM ACh, we performed genetic lineage tracing of neuroglial cells in *Wnt1-Cre2* mice, and cholinergic cells using *ChAT-Ires-Cre* mice and *ChAT-Gfp* mice. Matching the transient ACh increase, cholinergic nerve fibres peaked 2 weeks after transplantation (Fig. 1b and Supplementary Fig. 1a). Signalling through the GDNF family receptor alpha-2 (GFRα2) promotes the survival of cholinergic neurons[12,13]. Matching the transient increase in cholinergic innervation, BM GFRα2 mRNA expression increased 9-fold 2 weeks after transplantation (Fig. 1c). In sharp contrast, BM tyrosine hydroxylase (TH)+ noradrenergic fibres decreased steadily over 4 weeks (Fig. 1d and Supplementary Fig. 1b). In a separate study, we found that cholinergic neural signals are transmitted in the skeletal system through cholinergic osteoprogenitors. Therefore, we measured ChAT+ osteoprogenitors and found them similarly expanded 2 weeks after irradiation (Fig. 1e–h and Supplementary Fig. 1c–e). These results suggest that both neural and non-neuronal BM cholinergic signals increase during haematopoietic regeneration.

**Cholinergic regulation of HSC quiescence.** Since GFRα2 is required for the survival of cholinergic neurons[12,13], we used *Gfra2−/−* mice as a model to study the cholinergic regulation of

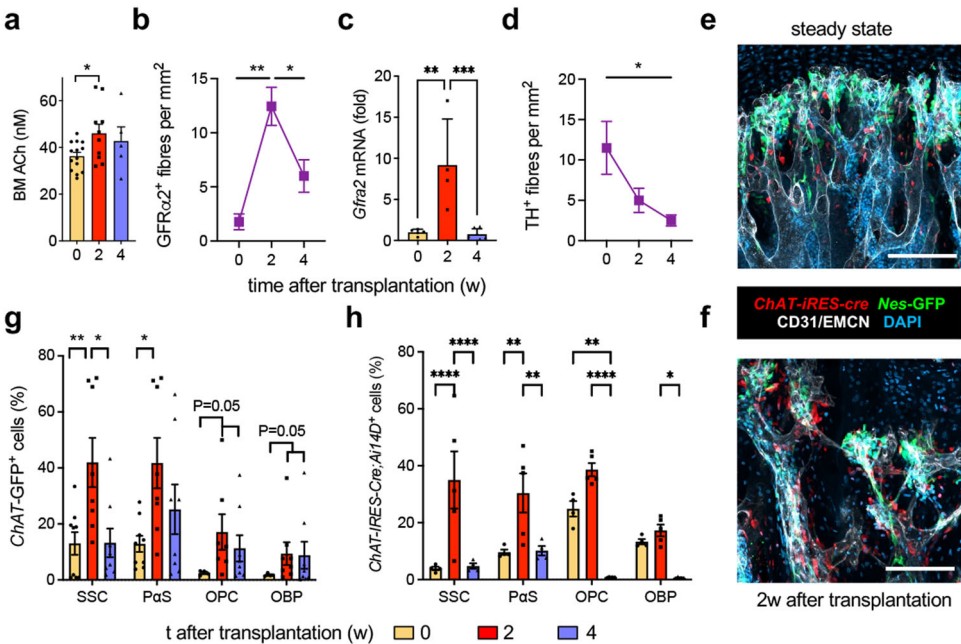

**Fig. 1 Cholinergic expansion during haematopoietic regeneration. a**, **c** Acetylcholine (ACh) content (**a**; *N* = 5,9,14; *P* = 0.03) and *Gfra2* mRNA expression (**c**; *N* = 5,4,7; *P* = 0.002,0.0008) in endosteal BM before or 2–4 weeks after transplantation. **b**, **d** Density of (**b**) cholinergic (GFRα2+; *N* = 4,5,9; *P* = 0.003,0.02) or (**d**) noradrenergic (tyrosine hydroxylase, TH+; *N* = 4,8,6; *P* = 0.01) genetically marked nerve fibres in *Wnt1-Cre2;Ai14D* mice. **e**, **f** Representative genetic tracing showing the expansion of cholinergic stromal cells (red) adjacent to *Nes*-GFP+ BMSCs (green) associated with CD31+ or endomucin (EMCN)+ blood vessels (white) near the growth plate of *ChAT-IRES-Cre;Ai14D;Nes-GFP* tibias (*N* = 3). Nuclei are counterstained with DAPI (blue). Scale, 50 µm. **g**, **h** Transient expansion of endosteal BM cholinergic PDGFRα+Sca1− skeletal stem cells (SSC), PDGFRα+Sca1+ (PαS) cells, PDGFRα−CD51+Sca1+ bone-lining osteoprogenitors (OPCs) or PDGFRα−CD51+Sca1− osteoblast precursors (OBPs) genetically traced in (**g**) *ChAT-Gfp* mice (0 weeks, SSC, PαS, *N* = 9. OPC, OBP, *N* = 4; 2 weeks, SSC, PαS, *N* = 8. OPC, *N* = 7, OBP, *N* = 8; 4 weeks, SSC, PαS, OPC, OBP, *N* = 8) or (**h**) *ChAT-Ires-Cre* mice (0 and 4 weeks, *N* = 4; 2 weeks, *N* = 5). **a**, **c**, **g**, **h** Each dot is a mouse. Data are mean of biological replicates ± SEM. *P < 0.05, **P < 0.01, ***P < 0.001, ****P < 0.0001. ANOVA and Tukey's multiple comparisons test.

HSCs. First, we analysed HSC proliferation separately in the endosteal or central BM of *Gfra2*[−/−] mice and WT mice (Supplementary Fig. 2a). In agreement with a previous study[14], BM lin[−]sca1[+]ckit[+] (LSK) CD150[+]CD48[−] HSCs were >5-fold more abundant in the central WT BM, which contained 7-fold more nucleated cells. However, the frequency of quiescent HSCs was 4-fold higher in the endosteal WT BM. Notably, *Gfra2*[−/−] mice showed 5-fold-reduced frequency of quiescent HSCs in the endosteal BM, whereas HSC proliferation remained unchanged in the central BM (Fig. 2a, b). An identical result was obtained using a different marker combination (LSK Flt3[−]) to label HSCs (Fig. 2c).

We have previously shown that decreased parasympathetic activity in *Gfra2*[−/−] mice derepresses sympathetic activity and causes abnormal BM egress of HSCs and leucocytes via sympathetic activation of the β₃-adrenergic receptor[9] (encoded by the *Adrb3* gene). To investigate the possible contribution of the noradrenergic system to decreased HSC quiescence in the endosteal BM of *Gfra2*[−/−] mice, we intercrossed these mice with *Adrb3*[−/−] mice. Unlike HSC mobilisation[9], increased endosteal HSC proliferation was not the consequence of derepressed noradrenergic activity in *Gfra2*[−/−] BM because it was not rescued in *Gfra2*[−/−]*Adrb3*[−/−] compound mice (Fig. 2c). These results suggest that cholinergic signals inhibit HSC proliferation locally in the endosteal BM niche.

**Niche α7nAChR promotes HSC quiescence.** GFRα2 has been shown to promote HSC self-renewal and ex vivo expansion through its co-receptor RET expressed in HSCs[15,16], but its role during regenerative haematopoiesis is unclear. Furthermore, GFRα2 protein was detected on the membrane surface of some (<5%) mature haematopoietic cells but was not detectable in immunophenotypically defined HSCs (Supplementary Fig. 2b, c). To clearly dissect haematopoietic-cell-autonomous and HSC-extrinsic regulation in *Gfra2*[−/−] mice, we generated chimeric mice through long-term transplantations of BM cells into lethally irradiated recipients. The decreased quiescence of endosteal HSCs in *Gfra2*[−/−] mice (see Fig. 2c) was not observed in WT mice carrying *Gfra2*[−/−] haematopoietic cells (Supplementary Fig. 2d, e), suggesting that HSC quiescence is extrinsically regulated by cholinergic signals through the microenvironment.

The chemokine CXCL12/SDF1α produced by non-haematopoietic niche cells regulates HSC quiescence[17,18]. We measured CXCL12 concentration in long-term BM chimeras using *Gfra2*[−/−] or control *Gfra2*[+/−] mice as donors or recipients to discriminate haematopoietic-cell-autonomous from niche regulation. In agreement with disrupted cholinergic regulation of the BM microenvironment in *Gfra2*[−/−] mice, only *Gfra2*[−/−] recipient mice (but not WT recipients of *Gfra2*[−/−] donor haematopoietic cells) exhibited a 35% reduction in BM CXCL12 concentration (Fig. 3a).

CXCL12 is highly produced by HSC niche-forming BMSCs marked by the regulatory elements of Nestin (*Nes*)[19]. Therefore, we intercrossed *Gfra2*[−/−] mice with *Nes-gfp* mice and isolated endosteal and central (see Supplementary Fig. 2a) BM CD45[−]Ter119[−]CD31[−]Nes-GFP[+/−] cells to investigate their possible regulation by cholinergic signals. *Cxcl12* mRNA expression was 2.5-fold higher in endosteal (vs. central) *Nes*-GFP[+] cells in control mice, but not in *Gfra2*[−/−] mice. This was due to 3-fold-reduced *Cxcl12* mRNA expression in endosteal *Nes*-GFP[+] cells from *Gfra2*[−/−] mice (Fig. 3b). In *Gfra2*[−/−] mice, *Cxcl12* expression was specifically deregulated in *Nes*-GFP[hi] cells (Fig. 3c), which reside in the endosteal BM[9] and promote HSC quiescence[20].

To further investigate how cholinergic signals regulate CXCL12, we treated WT mice with cholinergic antagonists selective for nicotinic or muscarinic receptors. Only the nicotinic antagonists

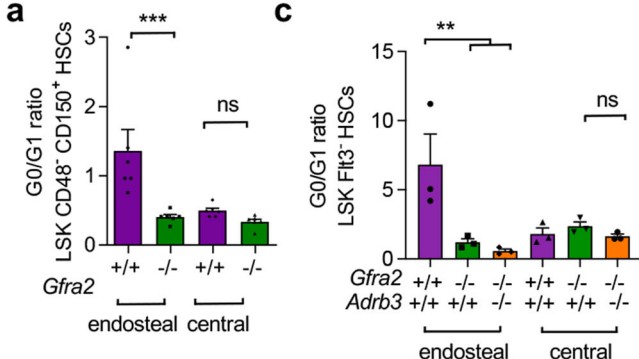

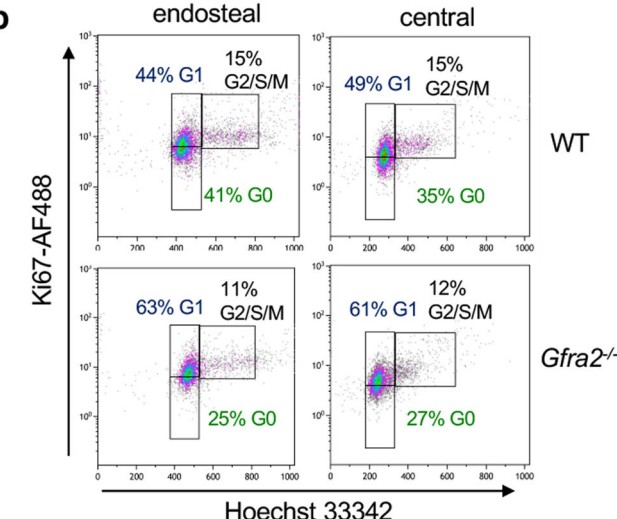

**Fig. 2 Cholinergic signals promote HSC quiescence in endosteal BM niches. a** Ratio of quiescent HSCs (G0:G1 phases of the cell cycle) in the endosteal or central BM of *Gfra2*[−/−] or WT mice (N = 6; P = 0.0006). **b** Representative flow cytometry diagrams showing ki67/Hoechst 33342 of lin[−] sca1[+] ckit[+] (LSK) CD48[−] CD150[+] HSCs isolated from the endosteal or central BM of *Gfra2*[−/−] mice or control *Gfra2*[+/−] mice (N = 6). The frequencies of the gated populations are indicated. **c** Ratio of quiescent HSCs in the endosteal or central BM of *Gfra2*[−/−], *Adrb3*[−/−], compound *Gfra2*[−/−];*Adrb3*[−/−] or WT mice (N = 3; P = 0.006). **a**, **c** Each dot is a mouse. Data are mean of biological replicates ± SEM. *P < 0.05, **P < 0.01, ***P < 0.001. ANOVA and Bonferroni's multiple comparisons test.

decreased BM CXCL12 content by 40% (Fig. 3d), suggesting that cholinergic signals are transduced by nicotinic receptors in the BM microenvironment. Therefore, we treated different BM stromal cells with acetylcholine or nicotine (Supplementary Fig. 3a). CXCL12 production was highest in MS-5 cells (resembling nestin[+] BMSCs) and was equally induced by ACh and nicotine, but was not affected by the muscarinic receptor antagonist atropine (Fig. 3e; Supplementary Fig. 3b). These results are consistent with more abundant expression of nicotinic receptors than muscarinic receptors in *Nes*-GFP[+] BMSCs[19]. Among nicotinic receptors, mRNA expression of α₇ (*Chrna7*) was high in *Nes*-GFP[+] cells (Supplementary Fig. 3c)[19]. Moreover, CXCL12 secretion was 5-fold lower in primary BM cultures from *Gfra2*[−/−] mice or *Chrna7*[−/−] mice (Fig. 3f), suggesting that cholinergic signals regulate BM CXCL12 expression through the α₇ nicotinic ACh receptor (α7nAChR). Supporting this possibility, CXCL12 was 3-fold lower in the BM extracellular fluid from *Chrna7*[−/−] mice (Fig. 3g). Together, these results suggest that cholinergic signals induce CXCL12 expression via α7nAChR in niche cells.

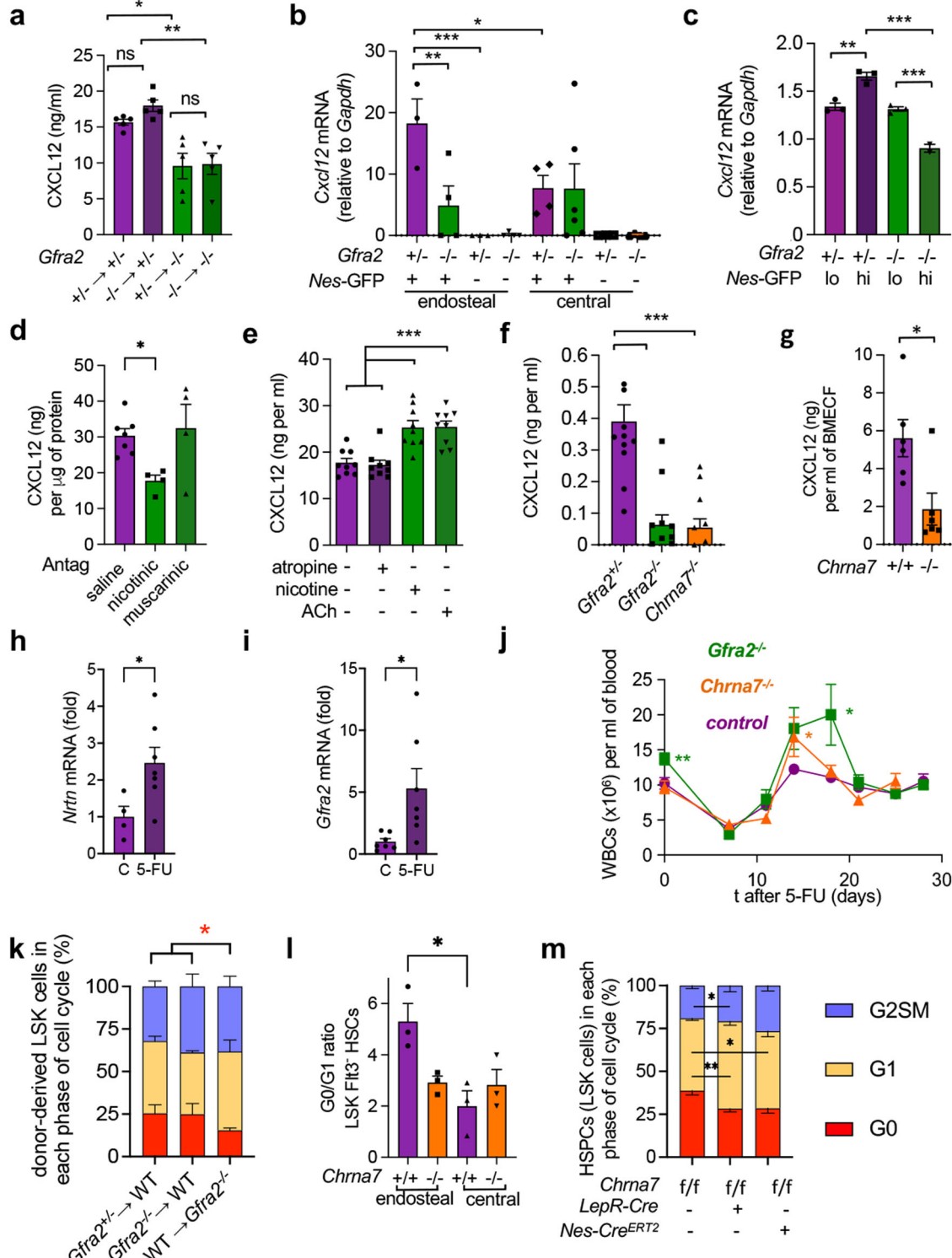

## Preserved HSC quiescence after chemotherapy

Since CXCL12 is essential to preserve HSC quiescence[17,18], we asked whether its regulation by cholinergic signals might affect HSC quiescence and activation during emergency haematopoiesis. 5-Fluorouracil (5-FU) is commonly used to treat cancer since it triggers apoptosis of proliferative cells but spares quiescent cells[21]. BM *Gfra2* and its ligand neurturin (*Nrtn*) mRNA expression increased 7 days after 5-FU (Fig. 3h, i), resembling the NRTN-GFRα2 induction after BMT. *Gfra2*−/− mice and *Chrna7*−/− mice exhibited a decreased ability to regulate haematopoietic recovery and fine-tune immune responses

acutely after 5-FU (14–18d); in contrast, haematopoiesis normalised in these mice after 20d, when stress haematopoiesis reverted to homeostasis (Fig. 3j), underscoring the role of cholinergic signals during stress haematopoiesis.

To confirm that cholinergic niche regulation is required to preserve HSC quiescence under proliferative stress, *Gfra2*−/− or control donor/recipient mice were transplanted with 500 endosteal HSCs and $10^5$ congenic BM helper cells; 4 weeks after transplantation, HSC quiescence was compromised only in the cholinergic-neural-deficient endosteal BM (Fig. 3k and Supplementary Fig. 4).

**Fig. 3 α7nAChR regulates CXCL12 and HSC quiescence in the endosteal BM niche. a** CXCL12 concentration in BM supernatant of chimeric mice 16 weeks after transplantation ($N = 5$; $P = 0.0003$). **b** *Cxcl12* mRNA expression (qPCR) in endosteal and central BM *Nes*-GFP-positive (+) or negative (−) CD45−CD31−Ter119− stromal cells from *Nes-gfp;Gfra2−/−* mice or control *Nes-gfp;Gfra2+/−* mice ($N = 6$; $P = 0.005$). **c** *Cxcl12* mRNA expression in BM stromal *Nes*-GFP^hi/lo cells ($N = 3$). **d** CXCL12 concentration in BM supernatant after in vivo treatment with cholinergic nicotinic or muscarinic antagonists, or saline ($N = 7, 4, 4$; $P = 0.04$). **e** CXCL12 secretion by MS-5 murine BM stromal cells treated for 24 h with nicotine, acetylcholine (ACh), the muscarinic antagonist atropine (10 μM) or vehicle ($N = 9$ technical replicates; $P < 0.0001$). **f** CXCL12 secretion in primary BM cultures from *Gfra2−/−*, *Chrna7−/−* or control *Gfra2+/−* mice ($N = 12$; $P = 0.004$). **g** CXCL12 concentration in BM supernatant of *Chrna7−/−* or WT mice ($N = 6$; $P = 0.01$). **h, i** mRNA expression of (**h**, $N = 4,5$) Neurturin (*Nrtn*) and (**I**, $N = 7$) its receptor *Gfra2* in BM of WT mice 7 days after 5-fluorouracil (5-FU) or control treatment. **j** Circulating white blood cells (WBCs) before and until 28 days after 5-FU treatment of *Gfra2−/−* ($N = 13$), *Chrna7−/−* ($N = 7$) or control mice ($N = 21$ pooled from all experiments). **k** Cell cycle profiles of endosteal BM HSPCs 4 weeks after transplantation (*Gfra2+/−* → WT, $N = 3$; *Gfra2−/−* → WT, $N = 3$; WT → *Gfra2−/−*, $N = 4$). **l** Ratio of quiescent HSCs in the endosteal or central BM of *Chrna7−/−* or WT mice ($N = 3$, $P = 0.02$). **m** Cell cycle profiles of endosteal BM HSCs 4 weeks after transplantation into *LepR-Cre;Chrna7f/f* mice ($N = 6$), *Nes-Cre^ERT2;Chrna7f/f* mice ($N = 3$) or control littermates ($N = 10$ pooled from two independent experiments). **a–i** and **l** Each dot is a mouse. Data are mean of biological replicates ± SEM. *$P < 0.05$, **$P < 0.01$, ***$P < 0.001$. **a–d** and **f** ANOVA and Bonferroni's multiple comparisons test, **g–i** Unpaired two-tailed *t*-test, **e** and **k–m** ANOVA and Tukey's multiple comparisons test.

Furthermore, reduced CXCL12 in *Chrna7−/−* BM (Fig. 3f, g) correlated with reduced endosteal HSC quiescence in *Chrna7−/−* BM (Fig. 3l), phenocopying *Gfra2−/−* mice (Fig. 2c). Importantly, α7nAChR deletion in Leptin-receptor-Cre-targeted HSC niche cells[22], which overlap with *Nes*-GFP+ BMSCs[23,24], or in *Nes-Cre^ERT2* targeted HSC niche cells, similarly increased endosteal (not central) BM HSC proliferation (Fig. 3m and Supplementary Fig. 5). These results suggest that ACh limits HSC proliferation by inducing CXCL12 expression in endosteal BMSCs via α7nAChR.

**HSCs cycling in cholinergic-deficient niche**. To evaluate the functional consequences of decreased HSC quiescence in the cholinergic-neural-deficient niche, we performed HSC long-term competitive repopulation assays using *Gfra2−/−* or control mice as donors or recipients. Consistent with increased WT HSC proliferation in cholinergic-deficient niches and with its normalisation in WT niches (see Fig. 3k), multilineage haematopoietic reconstitution from *Gfra2−/−* HSCs was unchanged, and that from WT HSCs remained high, 4–16 weeks after transplantation into *Gfra2−/−* mice (Fig. 4 and Supplementary Fig. 6). This was opposite to, and not explained by, the competitive disadvantage of CD45.1+ (compared with CD45.2+) cells[25] (Fig. 4, blue lines).

**Reduced HSC self-renewal in cholinergic-deficient niche**. To determine whether decreased HSC quiescence in *Gfra2−/−* mice compromises the HSC self-renewing programme, we compared the gene expression profiles (RNA-Seq) of endosteal and central HSCs isolated from WT mice and *Gfra2−/−* mice. Notably, gene set enrichment analysis (GSEA) showed that central (compared with endosteal) BM WT HSCs exhibited increased ribosomal, mitochondrial and GFRα1-related pathways (Fig. 5a, Supplementary Fig. 7a–d and Supplementary Data 1), suggesting a higher activation and different response to GFR signalling. Furthermore, the expression of target genes of the GFRα1/2 co-receptor RET, which promotes HSC self-renewal and expansion[15,16], was reduced in HSCs from *Gfra2−/−* mice (Fig. 5b). In contrast, myc-related glycolysis and Notch1-dependent pathways, which regulate HSC maintenance[26,27] and GFRα1/2-dependent, but RET-independent, maintenance of cardiac progenitors[28], were enriched in endosteal (compared with central) WT HSCs (Supplementary Fig. 7e, f). Similarly, neurite outgrowth-related pathways and interleukin-6 (IL-6)-dependent transcription were reduced in HSCs from *Gfra2−/−* mice (Supplementary Fig. 7g, h). In contrast, targets co-activated by Notch1 and Myc[29] were enriched in endosteal (compared with central) HSCs from WT, but not in *Gfra2−/−* mice (Fig. 5c and Supplementary Data 2). Matching cell cycle, GSEA revealed an abnormal upregulation of proliferation-associated gene sets in HSCs from endosteal *Gfra2−/−* BM. Similarly, mRNA levels of genes that are highly expressed in mobilized HSCs[30] were

increased in the endosteal HSCs from *Gfra2−/−* mice. In contrast, gene signatures associated with the most primitive long-term- (LT)-HSCs[30] and the HSC-fingerprint[31] were decreased in endosteal HSCs, compared with HSCs from central *Gfra2−/−* BM (Fig. 5d, Supplementary Fig. 7i–n and Supplementary Data 1–2).

To investigate the impact of the cholinergic regulation of HSCs on their self-renewal ability, 500 donor-derived HSCs were isolated from the primary recipient mice and were retransplanted (together with congenic helper BM cells) into lethally irradiated secondary recipients of the same genetic background as the primary recipient mice (Fig. 6a). Contrasting their normal reconstitution capacity in primary recipients (see Fig. 4b–g) and consistent with their reduced self-renewing gene programme (see Fig. 5d), endosteal HSCs from *Gfra2−/−* mice failed to reconstitute secondary recipients (Fig. 6b–g). Notably, WT HSCs continued to yield increased reconstitution shortly (4w) after secondary transplantation into *Gfra2−/−* mice; however, by 8 weeks their reconstitution capacity dropped sharply as a sign of decreased self-renewal or premature exhaustion (Fig. 6b–g). Altogether, these results suggest that cholinergic signals preserve HSC quiescence and self-renewal in the endosteal BM niche under proliferative stress.

**Nicotine impairs hematopoietic regeneration**. Since the cholinergic HSC regulation is transduced by nicotinic receptors in the niche (Fig. 3d–m and Supplementary Fig. 5), to complement the loss-of-function models, we treated mice with nicotine, as gain-of-function. A 3-day nicotine treatment increased quiescent endosteal BM HSCs (Supplementary Fig. 8). Therefore, transplanted mice were treated with nicotine or vehicle over 7 weeks (Fig. 7a). Decreased blood multilineage reconstitution in nicotine-treated mice (Supplementary Fig. 9a–h) was explained by decreased donor-derived chimerism in peripheral blood, persisting 7 weeks after transplantation (Fig. 7b–d). This was likely caused by increased quiescence of transplanted HSCs (Fig. 7e and Supplementary Fig. 9i). As a complementary loss-of-function model, transplanted mice were treated with a nicotinic receptor antagonist (Fig. 8a). Nicotinic receptor blockade increased BM HSC proliferation after transplantation (Fig. 8b, c). These results suggest that nicotinic signalling reduces HSC proliferation and haematopoietic reconstitution after transplantation.

**Smoking delays recovery after transplantation**. To investigate the possible association of nicotinic signalling with human HSC proliferation after transplantation, we retrospectively analysed the platelet recovery of 248 patients following allogeneic HSC transplantation, taking into consideration their smoking history before transplantation (Supplementary Data 3). Multivariate analysis indicated that normalisation of platelet counts was

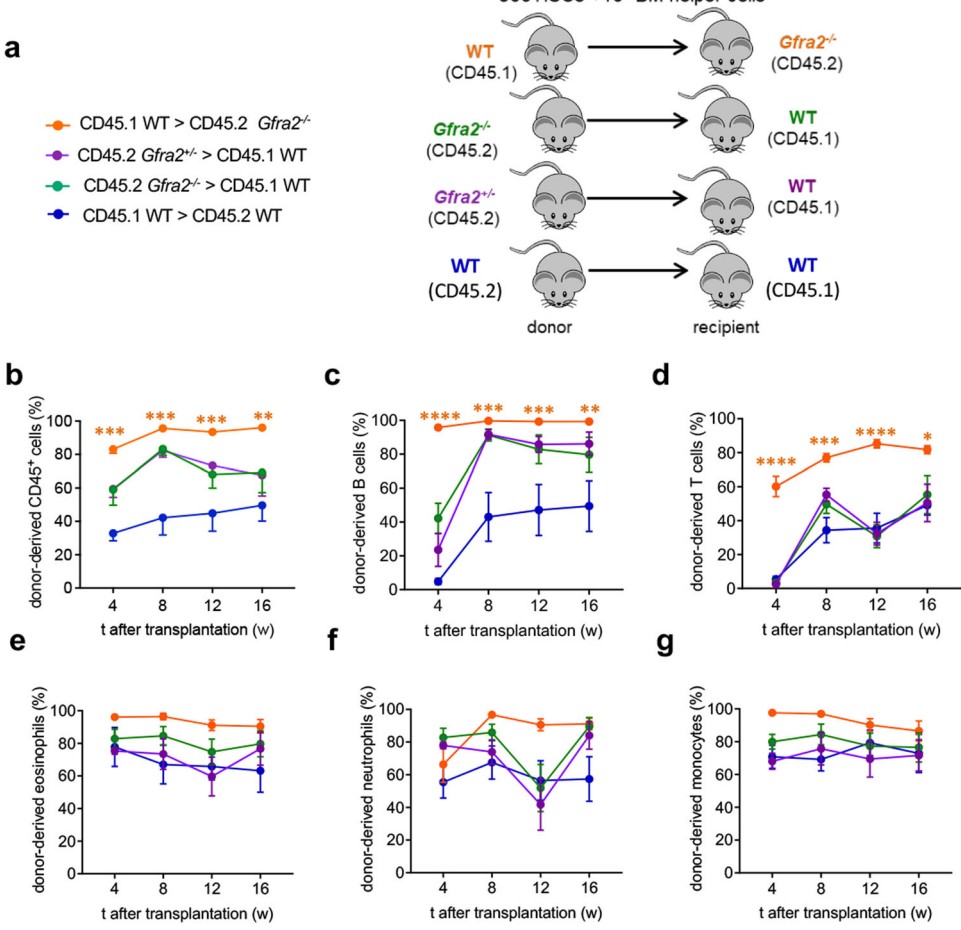

**Fig. 4 Cholinergic-deficient niche compromises HSC quiescence, leading to increased reconstitution of primary recipients. a** Scheme illustrating competitive HSC transplantation in primary recipients. 500 endosteal LSK CD48$^-$ CD150$^+$ HSCs and $10^5$ BM helper cells were i.v. injected into lethally irradiated congenic mice of indicated genotype. **b–g** Donor-derived haematopoietic chimerism (**b**), B cells (**c**), T cells (**d**), eosinophils (**e**), neutrophils (**f**) and monocytes (**g**) at specified time following primary transplantation of HSCs from donors and recipients of specified genotypes ($Gfra2^{+/-}$ > CD45.1, $N = 5$; $Gfra2^{-/-}$ > CD45.1, $N = 6$; CD45.1 > $Gfra2^{-/-}$, $N = 4$; CD45.1 > $Gfra2^{+/+}$, $N = 4$). Data are mean of biological replicates ± SEM. *$P < 0.05$, **$P < 0.01$, ***$P < 0.001$. ANOVA and Tukey's multiple comparisons test.

significantly delayed in previous smokers (Fig. 9), independently of the type of donor (matched family vs. alternative donor) and the number of CD34$^+$ cells infused, which also influenced platelet reconstitution kinetics, as expected[32,33]. Since increased transplant-related mortality has been noted in smokers treated for chronic myeloid leukaemia[34], these results suggest that cholinergic nicotinic signalling might impact clinical haematopoietic recovery after transplantation.

## Discussion

This study shows that cholinergic signals contribute to preserve HSC quiescence in endosteal BM niches. Cholinergic signalling increases in these niches upon haematopoietic stress (myeloablation, irradiation) and preserves HSC quiescence under proliferative stress, thereby helping protect HSCs from exhaustion.

Daily release of noradrenaline by sympathetic fibres innervating the BM activates the β$_3$-adrenergic receptor and reduces CXCL12 expression, permitting HSC egress to the circulation[2]. Moreover, the SNS-β$_3$-adrenergic-receptor-CXCL12 axis causes HSC mobilisation and correlates with HSC hyperproliferation under chronic stress[6]. However, preserving HSC quiescence under stress is critical to prevent HSC attrition, but the underlying mechanisms remain largely unknown[7,8]. Besides noradrenergic fibres, cholinergic nerve fibres have been found in

periosteal regions and inside the BM during postnatal stages[35,36], but their function had remained elusive. We recently demonstrated that dual cholinergic signals (acting centrally and peripherally) regulate HSC traffic between the BM and the bloodstream[9], but whether cholinergic signals regulate HSC quiescence, which is essential to preserve HSC self-renewal, was unknown. Non-myelinating Schwann cells associated with peripheral nerves reduce HSC proliferation[37], but it remained unclear whether and how different neurotransmitters released by these nerve fibres, and particularly acetylcholine (ACh), regulate HSC quiescence[11].

The current study expands the cholinergic regulation of HSC and leucocyte traffic[9] to the preservation of HSC quiescence and self-renewal under proliferative stress. The results suggest that the sympathetic nervous system (SNS) can simultaneously promote two opposing processes (stem cell activation/migration and induction of quiescence) through different neurotransmitters in separate BM niches. In this model, a stress-response system (SNS) could simultaneously trigger migration of activated stem cells in one niche (through noradrenaline), but at the same time protect the stem pool from exhaustion by promoting quiescence in another niche (through acetylcholine). This data illustrates a gatekeeper mechanism that can simultaneously trigger migration of activated stem cells in one niche, but at the same time protect

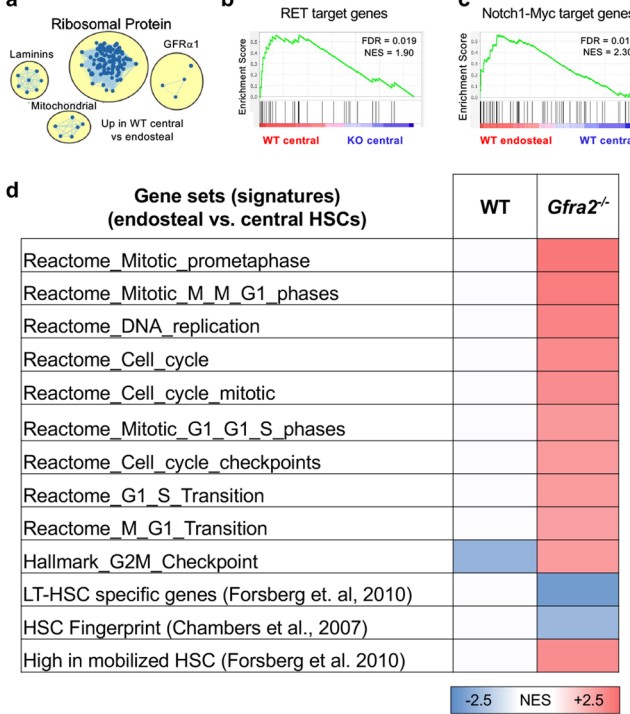

**Fig. 5 Reduced quiescence and self-renewing programme in endosteal HSCs from cholinergic-neural-deficient mice.** Gene sets enrichment analysis (GSEA) comparing endosteal and central lin⁻CD48⁻sca1⁺ckit⁺CD150⁺ HSCs of WT mice and *Gfra2⁻/⁻* mice ($N = 4$ biological replicates). **a** Enrichment map showing upregulated pathways in central (compared with endosteal) BM HSCs. **b** Downregulation of RET target genes in HSCs from *Gfra2⁻/⁻* mice. **c** Enriched Notch1-Myc target gene expression in endosteal WT HSCs. **d** Datasets from the Reactome, Pathway Interaction Database and Hallmark database as well as published HSC datasets were used for comparison. The normalised enrichment score (NES) of significantly enriched datasets (False discovery rate (FDR) < 0.1) is depicted as heatmap. Red, blue, and white colours indicate upregulation, downregulation, and no change, respectively.

the stem pool from exhaustion by promoting quiescence in another niche (Fig. 10).

Importantly, the expression of *Gfra2* and its ligand *Nrtn*, and the presence of cholinergic (but not noradrenergic) fibres increases in the BM under haematopoietic stress induced by myeloablation with 5-FU or after irradiation. Therefore, increased cholinergic nicotinic signalling in endosteal BM niches preserves HSC quiescence during emergency haematopoiesis.

Three independent experimental paradigms of our study (acute myelosuppression in two independent cholinergic-deficient lines, *Gfra2⁻/⁻* and *Chrna7⁻/⁻* mice, and BM reconstitution following lethal irradiation) convey the same message—increased HSC proliferation when cholinergic signals are impaired in the BM niche. However, this decreased HSC quiescence is associated with reduced self-renewing gene programme and diminished self-renewal in serial HSC transplantations. Decreased HSC quiescence in *Gfra2⁻/⁻* mice can be transiently reverted when the haematopoietic cells are transplanted in a WT microenvironment, although HSCs from *Gfra2⁻/⁻* mice fail to reconstitute secondary recipients. Conversely, the *Gfra2⁻/⁻* microenvironment decreases WT HSC quiescence and self-renewal in serial HSC transplantations. Altogether, these data points towards the cholinergic regulation of the HSC niche—rather than HSCs themselves—as a gatekeeper mechanism to protect HSC quiescence under stress.

The observed endosteal and central HSC distribution is in agreement with previous studies showing that the majority of HSCs are located in the perisinusoidal (central) niche[14,38]. However, consistently with our work, many studies highlight the presence of a smaller subset of HSCs associated with bone surfaces (endosteal niches), which were initially proposed to harbour quiescent HSCs[39–45] that expand to regenerate the damaged BM[46,47]. Our results support these contentions and more recent descriptions of low-permeability blood vessels (including endosteal arterioles and capillaries named "transition zone vessels") that serve as specialised niches for quiescent HSCs, whilst highly permeable sinusoids allow for HSC activation and trafficking[20,48–50]. We speculate that dynamic regulations in these two niches might regulate the reversible switch between dormant and activated HSCs based on physiological requirements (homoeostasis vs stress)[51,52].

Our findings reveal *Nes*-GFP⁺ *LepR-Cre*-targeted BMSCs as a target for cholinergic HSC regulation in the endosteal BM niche. CXCL12 levels are reduced in Nes-GFP^hi cells associated with decreased HSC quiescence in the endosteal BM of cholinergic-neural-deficient mice. Since CXCL12 is essential to preserve HSC quiescence[17,18] and our data shows that nicotine induces CXCL12 secretion by BM stromal cells, this appears to be one mechanism of cholinergic regulation of HSC quiescence through the niche. *Nes*-GFP^hi cells have been previously associated with HSC quiescence in arteriolar niches[20] and this might also be the case for endosteal transition zone vessels[48]. It is possible that other stromal cells expressing cholinergic receptors, such as endothelial cells[53] and monocytes/macrophages[54–56], might be targeted by this cholinergic regulation. Therefore, future studies should elucidate the possible regulation of other HSC niche cells by cholinergic signals.

Previous studies have suggested the expression of cholinergic nicotinic receptors in HSPCs and their possible activation by nicotine[57]. Specifically, α7nAChR has been proposed to mark HSPCs with a prominent role during inflammatory responses[58]. We have shown that cholinergic signals activate α7nAChR and induce CXCL12 expression in endosteal BMSCs as one mechanism promoting HSC quiescence. Indeed, our results show that α7nAChR deletion in *LepR-Cre*-targeted SSC-enriched cells[22], which overlap with *Nes*-GFP⁺ cells[23,24], or in *Nes-Cre^ERT2*- targeted HSC niche cells both increase endosteal (not central) BM HSC proliferation. Consequently, nicotine exposure increases endosteal HSC quiescence in vivo and impairs haematopoietic regeneration after HSC transplantation in mice. Our retrospective analysis of platelet recovery after clinical allogeneic HSC transplantation shows a significant delay in the normalisation of platelets in smokers, compared with non-smokers, independently of the stem cell dose and type of donor. Although other factors could contribute to delayed haematopoietic reconstitution in smokers, our data suggest that the cholinergic nicotinic signalling might affect clinical haematopoietic recovery after transplantation. These data are consistent with the reportedly inhibitory effects of nicotine during in vivo haematopoiesis[59] and in vitro megakaryopoiesis[60], and suggest that the cholinergic nicotinic signalling might affect clinical haematopoietic recovery after transplantation. Our data might relate to the increased transplant-related mortality previously noted in high-dose smokers who received HSC transplantation for the treatment of chronic myeloid leukaemia[34].

In summary, these results suggest that the cholinergic nicotinic regulation of the endosteal HSC niche preserves HSC quiescence under haematopoietic proliferative stress. They also reveal how adult stem cells can be regulated by different signals from the autonomic nervous system to meet physiological demands at various sites and time, and to efficiently respond to stress without being exhausted.

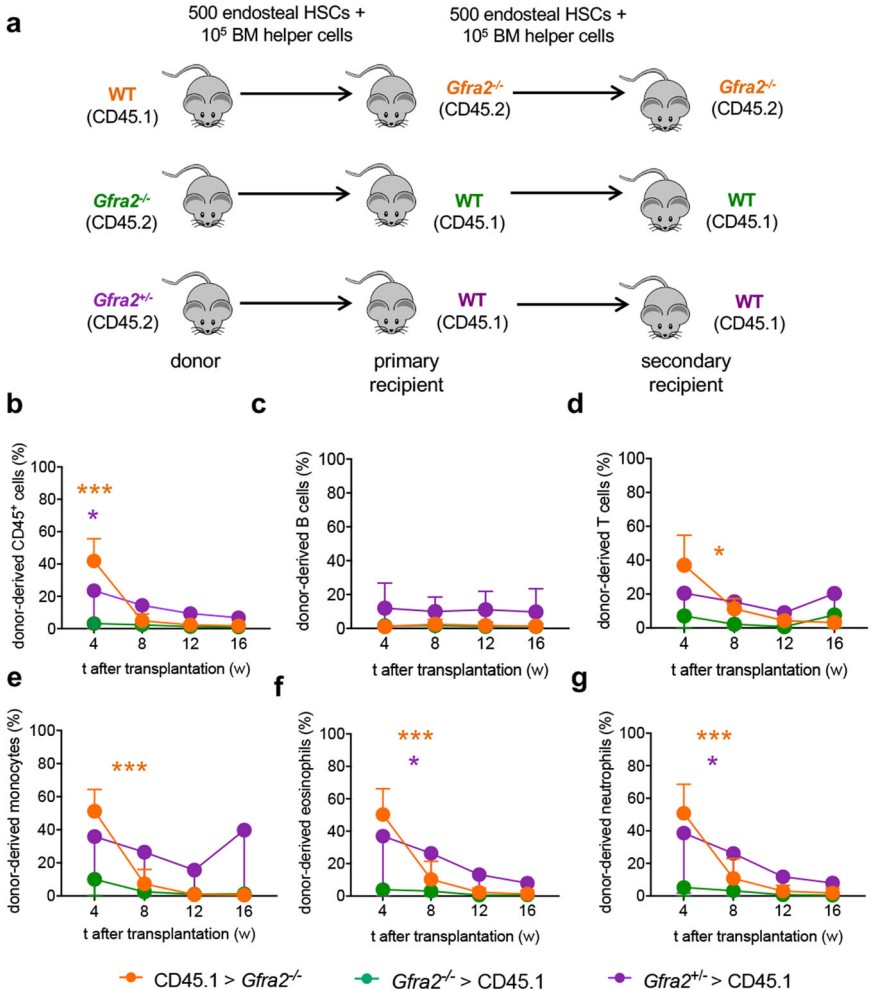

**Fig. 6 Reduced HSC self-renewal in a cholinergic-neural-deficient niche. a** Scheme illustrating competitive serial HSC transplantations. 500 endosteal LSK CD48⁻ CD150⁺ HSCs isolated from primary recipients were i.v. injected together with $10^5$ BM helper cells into lethally irradiated secondary recipients of the same genotype as the primary recipient mice. **b–g** Donor-derived haematopoietic chimerism (**b**), B cells (**c**), T cells (**d**), monocytes (**e**), eosinophils (**f**) and neutrophils (**g**) and at specified time following secondary transplantation of mice with specified genotype ($Gfra2^{+/-} >$ CD45.1, $N = 3$; $Gfra2^{-/-} >$ CD45.1, $N = 4$; CD45.1 $> Gfra2^{-/-}$, $N = 4$). Data are mean of biological replicates ± SEM. *$P < 0.05$, ***$P < 0.001$. ANOVA and Tukey's multiple comparisons test.

## Methods

**Human study design**. Patients provided written consent before transplant for the collection and analysis of anonymised data, which were maintained in the Hospital registry, in line with legal and regulatory requirements.

**Animals**. Age and sex-matched $Gfra2^{-/- REF}$12, $Nes$-$gfp$[61] (generously provided by G.E. Enikolopov), $FVB/N$-$Adrb3tm1Lowl/J$ (Stock number 006402), $B6;129\times1$-$Nrtn^{tm1Jmi}/J$ (Stock number 012238), $B6.129S7$-$Chrna7^{tm1Bay}/J$ (Stock number 003232), $ChATBAC$-$eGFP$ (Stock number 007902), $\alpha7nAChRflox$ (Stock number 026965), $B6.129(Cg)$-$Lepr^{tm2(cre)Rck}/J$ (Stock number 008320), $Nes$-$cre^{ERT2(REF 62)}$ (generously provided by G. Fishell), $B6.Cg$-$Commd10Tg(Vav1$-$icre)$ $A2Kio/J$ (Stock number 008610) (Jackson Laboratories), and congenic CD45.1 and CD45.2 C57BL/6 J mice (Charles River) were used in this study. For genetic lineage tracing, $B6.Cg$-$Gt(ROSA)26Sor^{tm14(CAG$-$tdTomato)Hze}/J$ ($Ai14D$) reporter mice (Stock number 007908) were crossed with $B6.129S$-$Chat^{tm1(cre)Lowl}/MwarJ$ mice ($Chat$-$IRES$-$Cre$) (Stock number 031661) and $Wnt1$-$Cre2$ (Stock number 022501) (Jackson Laboratories). Mice were housed in specific pathogen-free facilities. All experiments using mice followed protocols approved by the Animal Welfare Ethical Committees at the University of Cambridge (PPL 70/8406 and PPL P0242B783). All experiments were compliant with EU recommendations.

**Cell culture**. MS-5 cells (DSMZ, ACC 441), a stromal cell line established by irradiation of adherent cells in long-term bone marrow cultures[63], were grown in monolayers in α-MEM supplemented with 10% FBS, 2 mM l-glutamine and 2 mM sodium pyruvate (Invitrogen). ST-2 cells, another stromal cell line established from Whitlock-Witte type long-term bone marrow cultures[64], were grown in RPMI 1640 medium containing 10% FBS. MC3T3-E1 cells, fibroblastic cell line established

from the skull of an embryo/foetus C57BL/6 mouse[65], were purchased from ATCC and grown in α-MEM medium supplemented with 10% FBS. MLO-Y4 cells, osteocyte cell line originally derived from the long bones of transgenic mice expressing SV40 large T-antigen oncogene under the control of osteocalcin promoter[66], were kindly provided by Prof. Lynda Bonewald and seeded in collagen I pre-coated-6 well dishes and grown with α-MEM medium supplemented with 5% FBS and 5% iron-supplemented calf serum (Invitrogen). All cultures were maintained with 1% penicillin–streptomycin (Invitrogen) at 37 °C in a water-jacketed incubator with 5% $CO_2$ and 1:5 split with 0.05% trypsin-EDTA (Invitrogen) every three or 4 days, when cells reached about 80% confluence. Routine tests confirmed the absence of mycoplasma contamination in the cultures.

**BM extraction, flow cytometry and fluorescence-activated cell sorting**. For BM hematopoietic cell isolation, bones were crushed in a mortar and filtered through a 40-μm strainer to obtain single-cell suspensions. Tissues were depleted of red blood cells by commercial lysis (Biolegend, 420301) for 8 min at 4 °C. Blood samples were directly lysed.

Cells were incubated with the appropriate dilution (2–5 μg/ml) of fluorescent antibody conjugates and 4′,6-diamidino-2-phenylindole (DAPI, 1:2000) for dead cell exclusion, and analysed on LSRFortessa flow cytometer (BD Biosciences, Franklin Lakes, NJ) equipped with FACSDiva Software (BD Biosciences). The following antibodies were used: Rat Anti-Mouse CD45R/B220 (Clone RA3-6B2, BD Biosciences, 553088), Ckit (Clone 2B8, BioLegend, 105825), Biotin Mouse Lineage Panel (CD11b, Gr1, Ter119, B220, CD3e) (BD Biosciences, 559971), Human/Mouse GFR alpha-2/GDNF R alpha-2 Antibody (Bio-Techne, AF429), CD45.1 (Clone A20, Insight Biotechnology, 60-0453-U100), CD45.2 (Clone 104, BioLegend UK Ltd, 109823), Sca1 (Clone D7, Biolegend, 108114), Goat anti-rabbit

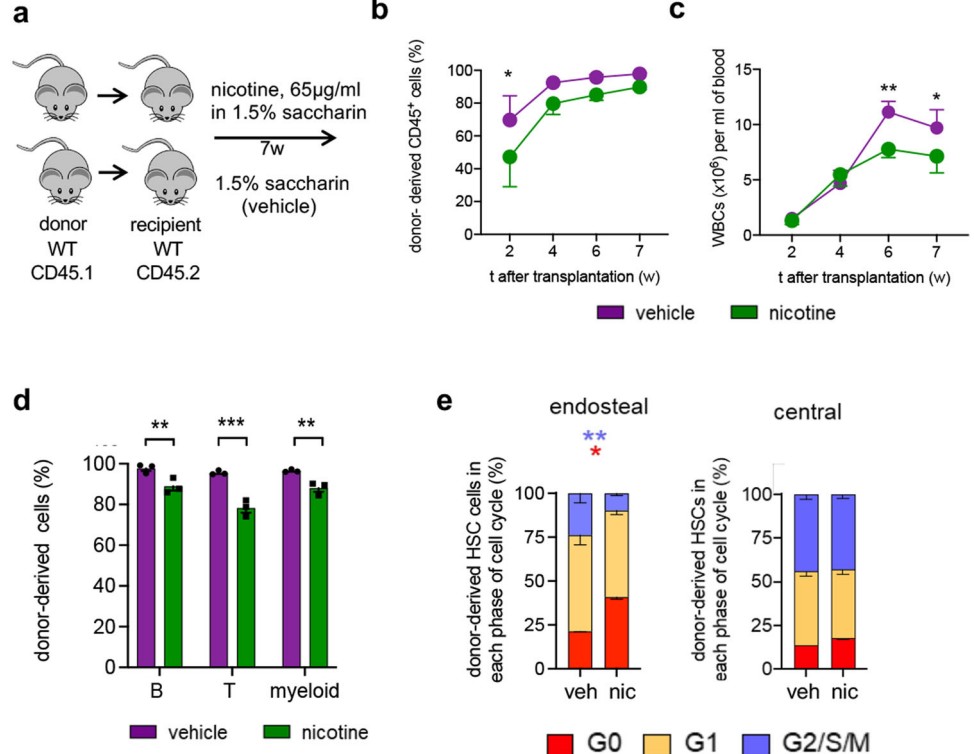

**Fig. 7 Nicotine induces HSC quiescence after transplantation. a** Scheme illustrating experimental design. Irradiated WT CD45.2 recipients were transplanted with $2 \times 10^6$ BM nucleated cells from CD45.1 mice and treated with 65 µg/ml nicotine or vehicle in their drinking water over 7 weeks. **b, c** Circulating donor-derived (CD45.1$^+$) haematopoietic cells (**b**, $P = 0.03$) and white blood cells (**c**, $P = 0.006, 0.04$) at specified time after transplantation ($N = 3$). **d** Donor-derived B cells ($P = 0.007$), T cells ($P < 0.0001$) and myeloid cells ($P = 0.009$) from mice treated with nicotine or vehicle at 7 weeks ($N = 3$). **e** Cell cycle profiles of donor-derived lin$^-$sca1$^+$ckit$^+$ (LSK) CD34$^-$flt3$^-$ HSCs isolated from the endosteal or central BM 7 weeks after transplantation and chronic treatment with nicotine or vehicle ($N = 3; P = 0.005, 0.04$). (**b–e**) Data are mean of biological replicates ± SEM. *$P < 0.05$, **$P < 0.01$, ***$P < 0.001$. (**b–e**) ANOVA and Sidak's comparisons.

AF488 (Thermo Fisher Scientific, A-11008), Sca1 PE Cy7 (Clone E13-161.7, BioLegend UK Ltd, 108113), CD34 (Clone HM34, BioLegend UK Ltd, 128603), CD150 (Clone TC15-12F12.2, BioLegend UK Ltd, 115909), CD48 (Clone HM48-1, BioLegend UK Ltd, 103439), CD41 (Clone MWReg30, BD Biosciences, 561850), CD49b (Clone HMa2, BD Biosciences, 558759), CD135 (Clone A2F10, BioLegend UK Ltd, 135307), Flt3 (Clone A2F10.1, BD Biosciences, 560718), Rabbit anti ki67 (Abcam, AB15580), CD31 (Clone MEC 13.3, BD Biosciences, 553371), Hoechst 33342 Solution (20 mM) (Thermo Fisher Scientific, 62249), Dylight 650 Donkey anti-rat (Thermo Fisher Scientific, SA5-10029), Ly-6G/Ly-6C (Gr1) (Clone HK1.4, Biolegend, 108411), CD11b (Clone M1/70, Biolegend, 101207), CD3e (Clone 145-2C11, Biolgend, 100349). Biotinylated antibodies were detected with fluorochrome-conjugated streptavidin (BD Biosciences, 554061). All antibodies were used at 1:200 except for biotinylated lineage antibody mix and rabbit anti ki67 at 1:100, cells incubated in 300 ul for staining.

We isolated HSPCs from bone-associated and non-associated marrow fractions as previously described[41]. Briefly, long bones were flushed gently to obtain hematopoietic cells less tightly associated with the bone and the flushed-bones were then crushed in a mortar to obtain hematopoietic cells of the endosteal compartment. Cells were stained with the above-detailed antibodies and analysed by flow cytometry or sorted (FACS Aria cell sorter, BD Bioscience). HSCs were immunophenotypically defined as lin$^-$ CD48$^-$ sca-1$^+$ c-kit$^+$ CD150$^+$ cells.

To isolate nestin$^+$ cells, bones were cleaned off surrounding tissue, crushed in a mortar with a pestle, and digested with collagenase (catalogue number C2674, Sigma; 0.25% collagenase in PBS supplemented with 20% foetal bovine serum) in water bath at 37 °C for 30 min with agitation. Cells were filtered through a 40-µm strainer and erythrocytes were lysed as described above. The resulting BM-enriched cell suspensions were pelleted, washed and resuspended in PBS containing 2% FCS for further analyses. BM stromal CD45$^-$ (30-F11) CD31$^-$ (MEC 13.3) Ter119$^-$ cells were further purified according to GFP fluorescence using an LSRFortessa flow cytometer (BD Biosciences) for immunophenotypic analysis, or a BD FACS Aria or BD Influx Sorter (BD Biosciences) for cell sorting.

To separate endosteal and non-endosteal nestin$^+$ cells, we gently flushed the long bones as described above. We digested both the flushed fraction and the remaining bone samples with collagenase (catalogue number C2674, Sigma; 0.25% collagenase in PBS supplemented with 20% foetal bovine serum) in a water bath at 37 °C for 30 min with agitation.

Cell cycle analysis was performed through Hoescht 33342 (H42)/ PironinY (PY) staining as described[67] or through Hoechst 33442 (H42)/ ki67. Briefly, for Hoescht 33342 (H42)/ PironinY (PY) staining, sorted lin$^-$sca-1$^+$c-kit$^+$flt3$^-$ cells were collected in α-MEM medium supplemented with 2% foetal calf serum and 10 mM Hepes. Cells were incubated with H42 (5 µg/ml) 35 min in α-MEM medium supplemented with 2% foetal calf serum, 10 mM Hepes and 50 µM verapamil at 37 °C. PY (1 µg/ml) was then added and cells were incubated for 20 min at 37 °C. Finally, cells were resuspended in a medium with 50 µM verapamil and analysed by flow cytometry. For Hoechst 33342(H42)/ki67 method, single cell suspensions were obtained and stained for cell surface markers and fixed Cytofix/Cytoperm (BD Biosciences, 554714) for 10 min at RT. Cells were then washed in Perm/Wash (BD Biosciences, 554714) and resuspended in ki67 (Abcam, AB15580, 1:100) for a minimum of 45 mins at 4 °C or overnight. Cells were washed and resuspended in goat anti-rabbit AF488 (Thermo Fisher Scientific, A-11008) at 1:200 for 20 mins on ice and subsequently in Hoechst at 1:2000 for 5 mins at 4 °C. A final wash was carried out and cells were resuspended in 300 µl Perm/Wash for analysis.

**Long-term competitive repopulation assay.** To assess endosteal HSCs, 500 sorted HSCs (Lin$^-$Sca1$^+$cKit$^+$CD48$^-$CD150$^+$) from *Gfra2$^{-/-}$*, *Gfra2$^{+/-}$* (CD45.2) or congenic CD45.1 mice isolated from crushed flushed-bones as donor cells together with either 105 CD45.1$^+$ or CD45.2$^+$ competitor bone marrow nucleated cells. In secondary transplantations, 500 sorted HSCs (Lin$^-$Sca1$^+$cKit$^+$CD48$^-$CD150$^+$) isolated from crushed flushed-bones from the primary recipients were sorted and used as donor cells along with either 10$^5$ CD45.1$^+$ or CD45.2$^+$ competitor bone marrow nucleated cells. All recipient mice were split dose irradiated with 12 Gy. Hematopoietic reconstitution was assessed in the peripheral blood of recipient mice several times after transplantation by measuring blood CD45.2 or CD45.1 chimerism in the different mature hematopoietic lineages: B cells (B220$^+$), T cells (CD3$^+$), monocytes (CD11b$^+$ Gr1$^-$), eosinophils (CD11b$^+$ Gr1$^{low}$) and neutrophils (CD11b$^+$ Gr1$^{high}$).

**Choline acetyltransferase$^+$ cell detection.** To assess which cell populations express choline acetyltransferase (ChAT) under steady state and after stress, transplantations were performed. To assess the stromal ChAT$^+$ cells, *ChAT-GFP* and *ChAT-IRES-Cre;Ai14D* (CD45.2) recipient mice were split dose irradiated with

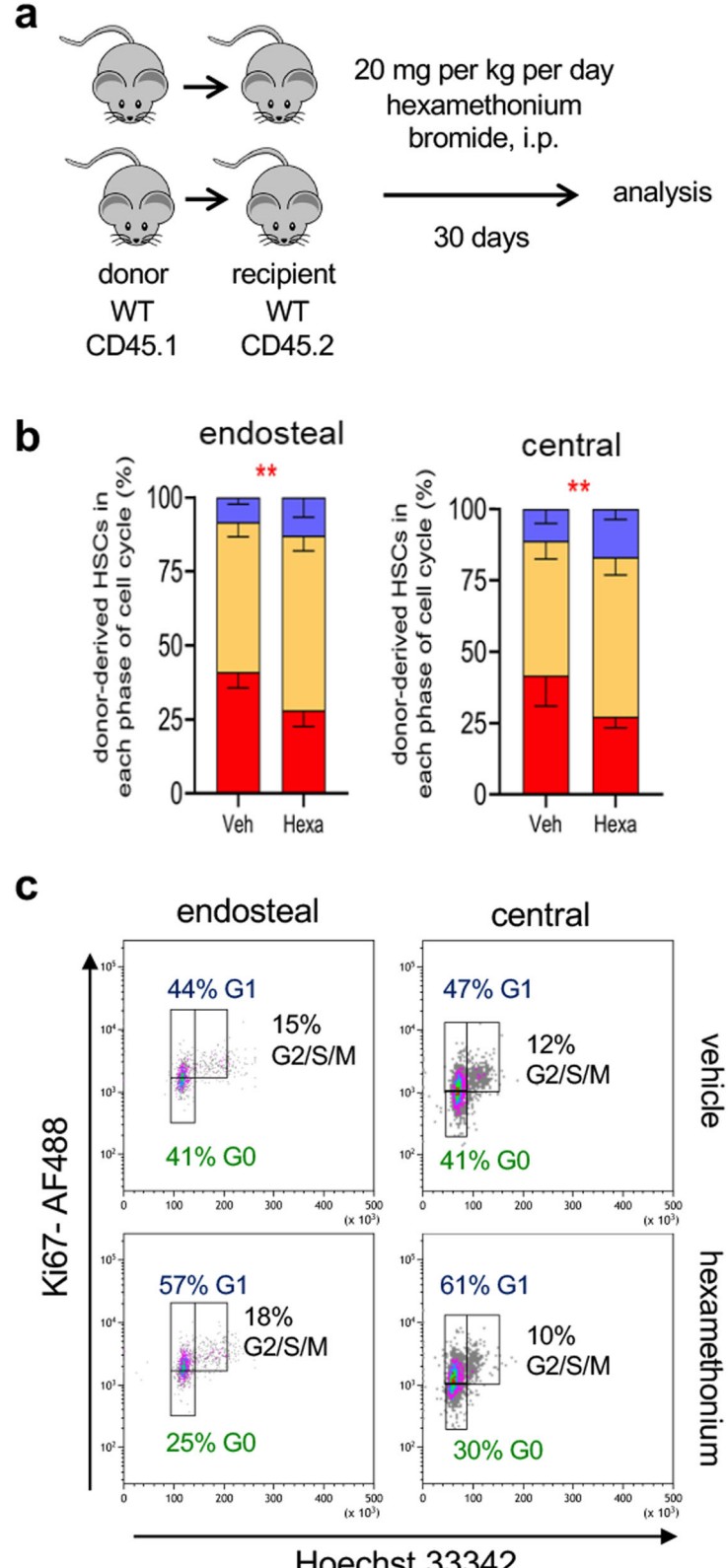

**Fig. 8 Nicotinic receptor blockade reduces HSC quiescence after transplantation. a** Scheme illustrating the treatment of sub-lethally irradiated WT CD45.2 recipients of $2 \times 10^6$ BM nucleated cells from CD45.1 mice, treated with Hexamethonium bromide (20 mg/kg/d over 30d, i.p.) or vehicle. **b** Cell cycle profiles of donor-derived (CD45.1+) lin-sca1+ckit+ (LSK) CD34-flt3- HSCs isolated from the endosteal or central BM 30 days after transplantation and chronic treatment with hexamethonium bromide or vehicle ($N = 5$;$P = 0.002,005$). Data are mean of biological replicates ± SEM. **$P < 0.01$. ANOVA and Sidak's comparisons. **c** Representative flow cytometry plots showing HSCs in different phases of the cell cycle. The frequencies of the gated populations are indicated.

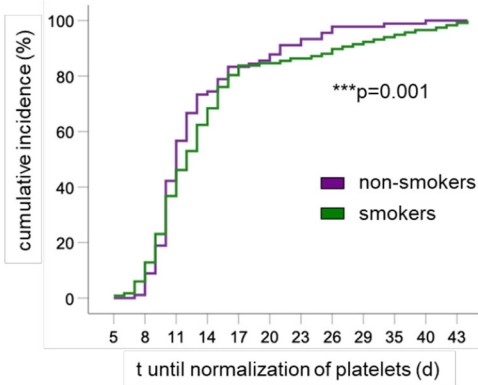

**Fig. 9 Smoking history is associated with delayed platelet normalisation after transplantation.** Cumulative incidence of time ($t$) until normalisation of circulating platelets in patients undergoing allogeneic HSC transplantation, taking into account their smoking history ($N = 248$). Multivariate Cox Analysis included the type of donor (matched family vs. alternative donor) and the number of CD34$^+$ cells infused as independent variables. Omnibus tests of model coefficients.

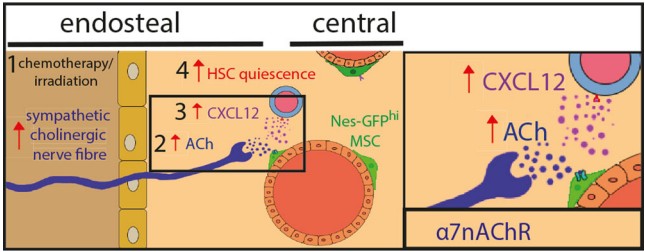

**Fig. 10 Sympathetic cholinergic fibres preserve HSC quiescence in endosteal BM niche.** 1 Sympathetic cholinergic signals increase in the endosteal bone marrow after chemotherapy or irradiation. 2 Acetylcholine (ACh) is released by sympathetic fibres in the endosteal niche for haematopoietic stem cells (HSCs). 3 ACh activates α7 nicotinic ACh receptor (α7nAChR) and induces CXCL12 secretion by *Nes*-GFP$^{hi}$ cells enriched in HSC niche-forming mesenchymal stem cells (MSCs). 4 Cholinergic-induced CXCL12 promotes HSC quiescence and self-renewal.

12 Gy and transplanted with 2 million CD45.1 bone marrow nucleated cells. Mice were culled at 0, 2 weeks and 4 weeks post-transplantation and the following stromal populations were analysed: CD45$^-$TER119$^-$CD31$^-$PDGFRα$^+$Sca1$^-$ skeletal stem cells (SSCs), CD45$^-$TER119$^-$CD31$^-$PDGFRα$^+$Sca1$^+$ (PαS) cells[68], CD45$^-$TER119$^-$CD31$^-$PDGFRα$^-$CD51$^+$Sca1$^+$ bone-lining osteoprogenitors (OPCs) and CD45$^-$TER119$^-$CD31$^-$PDGFRα$^-$CD51$^+$Sca1$^-$ osteoblast precursors, OBPs[69–71].

**In vivo treatments.** Haematopoietic recovery in mice was assayed administering a single dose of 5-FU (150 mg/kg, i.p., diluted in PBS), and bleeding mice for the first time before the 5-FU injection and then every 3–4 days for haematological analysis.

To measure GFRα2$^+$ or TH$^+$ fibres following irradiation, 10$^6$ bone marrow nucleated cells from CD45.1 were used as donor cells in *Wnt1-Cre2/tdTomato* (CD45.2) recipient mice. All recipient mice were split dose irradiated with 9.5 Gy.

To assess the role of Chrnα7 signalling in stromal cells, *Chrnα7-floxed;LepR-Cre* (CD45.2) recipient mice were split dose irradiated with 12 Gy and transplanted with 2 million CD45.1 bone marrow nucleated cells. Mice were culled at 4 weeks post-transplantation and cell cycle status was analysed.

To generate recipients to assess the role of Chrnα7 signalling in stromal cells, *Chrnα7-floxed;Nes-CreERT2* (CD45.2) mice, pregnant dams were treated with 4 mg tamoxifen on days 1 and 3 via oral gavage. Induced recipients mice were split dose irradiated with 12 Gy and transplanted with 2 million CD45.1 bone marrow nucleated cells. Mice were culled at 4 weeks post-transplantation and cell cycle status was analysed. To assess the role of Chrnα7 signalling in HSCs, CD45.1 recipient mice were split dose irradiated with 12 Gy and transplanted with 2 million *Chrnα7-*

*floxed;Vav-Cre* (CD45.2) bone marrow nucleated cells. Mice were culled at 4 weeks post-transplantation and cell cycle status was analysed.

For pharmacological cholinergic blockade experiments, mice were i.p injected with nicotinic antagonists (mecamylamine, 3 mg/kg; and hexamethonium, 20 mg/kg), muscarinic antagonists (scopolamine, 3 mg/kg; and methylatropine nitrate, 3 mg/kg) or vehicle at Zeitgeber time (ZT) 5 (5 h after light onset). Mice were culled and analysed at ZT13.

For the nicotinic agonist and antagonist experiments, CD45.2 C57BL/6 J mice were treated with 100 μg/ml nicotine (Sigma, N3876-5ML) in 1.5% saccharin (Sigma) via their drinking water. To assess in the transplantation setting, CD45.2 C57BL/6 J recipient mice were split dose irradiated with 9.5 Gy and transplanted with 2 million CD45.1 bone marrow nucleated cells. Mice were administered with 65ug/ml nicotine 24 h later via their drinking water for the remainder of the study, changed every 2–3 days. For the antagonist experiments, CD45.2 C57BL/6 J recipient mice were split dose irradiated with 12 Gy and transplanted with 2 million CD45.1 bone marrow nucleated cells. Mice were treated with 20 mg/kg/day Hexamethonium bromide (i.p.) daily for 30 days.

**Immunofluorescence.** Immunofluorescence staining of cryostat sections was performed as previously described[72]. Briefly, tissues were permeabilized with 0.1% Triton X-100 (Sigma) for 10 min at RT and blocked with TNB buffer (0.1 M Tris–HCl, pH 7.5, 0.15 M NaCl, 0.5% blocking reagent, Perkin Elmer) for 1 h at RT. Primary antibody incubations were conducted overnight at 4 °C. Secondary antibody incubations were conducted for 2 h at RT. Repetitive washes were performed with PBS + 0.05% Triton X-100. Stained tissue sections were counterstained for 5 min with 5 μM DAPI and rinsed with PBS. Slides were mounted in Vectashield Hardset mounting medium (Vector Labs). *ChAT-Ires-Cre*-positive nerve fibre staining was performed on half-bones according to previous reports (Acar et al., 2015), whereby bones were longitudinally bisected using a cryostat, blocked O/N in staining buffer (5% donkey serum, 0.5% IgepAl, 10% DMSO) supplemented with 1% BlokHen (Aves Labs, Cat. No. BH-1001) and stained with primary and secondary antibodies for 3 days in staining buffer with daily intervening washes in PBS at RT. We used the following primary antibodies: tyrosine hydroxylase (1:1000, rabbit polyclonal antibody, Millipore), Gfra2 (1:200, goat polyclonal antibody, R&D), Chicken anti-GFP (Aves Labs, Cat. No. GFP-1020), Rat anti-CD31 (1:100, BD Biosciences, Cat. No. 550274, Clone MEC 13.3), Rat anti-Endomucin (1:200, Santa Cruz, Cat. No. sc-65495, clone V.7C7) and anti-DsRed polyclonal antibody (Takara Cat. No. 632496). The following antibodies were used for secondary staining: Alexa Fluor 488 donkey anti-chicken (Jackson Immuno, Cat. No. 703-545-155), Alexa Fluor 647 Donkey anti-rat (Abcam, Cat. No. ab150155), Alexa Fluor 647 Donkey anti-goat and Alexa Fluor 647 Donkey anti-rabbit. Sections were scanned with Zeiss Axioscan Z1 slide scanning microscope. For all stainings, control and experimental samples were processed simultaneously and were blindly analysed using Zen lite software.

**ELISA.** CXCL12 protein levels were measured by conventional ELISA. Briefly, 96-well plates were coated overnight at 4 °C with 2 μg/ml of monoclonal CXCL12/SDF-1 antibody (MAB350, R&D Systems). After blocking, BM extracellular fluids were incubated with the antibody for 2 h at room temperature, followed by addition of biotinylated anti-human and mouse CXCL12/SDF-1 antibody (BAF310, RD). Streptavidin-horseradish peroxidase conjugate (RPN1231V, Dako) was used to detect the signal, and the reaction was developed with horseradish peroxidase substrate (TMB, ES001-500ML, Chemicon, Millipore). Standard curve was performed with recombinant SDF-1 alpha (350-NS, R&D).

ELISA for norepinephrine/epinephrine (Bi-CAT ELISA ALPCO) was performed according to the manufacturers' recommendations. Acetylcholinesterase measures were performed with Choline/Acetylcholine Assay Kit, Fluorometric protocol (Abcam, Cat. No. ab65345).

**RNA isolation, reverse transcription and quantitative real-time PCR (qPCR).** Total RNA extraction was performed with TRIzol (Invitrogen), followed by treatment with DNase to eliminate contaminating genomic DNA with RNase-free DNase Set (Qiagen) and RNA clean-up with RNeasy mini kit (Qiagen). Alternatively, for small cell numbers, RNA was isolated using the Dynabeads® mRNA DIRECT™ Micro Kit (Invitrogen). Reverse transcription was performed using the Reverse Transcription System (Promega), following the manufacturer's recommendations. The expression level of each gene was determined by the relative standard curve method, using a standard curve prepared from serial dilutions of a mouse or human reference total RNA (Clontech). The expression level of each gene was calculated by interpolation from the standard curve. All values were normalised to *Gapdh* as the endogenous control. The sequence of oligonucleotides used for quantitative real-time RT-PCR are available in Supplementary Data 4.

**RNAseq sample preparation.** Pools of 30 viable LSK CD48$^-$CD150$^+$ cells were sorted (BD Influx™ cell sorter) into 4 μl lysis buffer (0.5 U/μl SUPERase In RNase Inhibitor in 0.2 % (v/v) Triton X-100) containing 12.5 mM DTT and 2.5 mM dNTP. RNAseq was performed following Smart-seq2 protocol[73]. Briefly, RNA was primed with oligo-dT primers and reverse transcribed using Superscript II Reverse Transcriptase (200 U/μl, Thermo Fisher 18064071). KAPA HiFi Hotstart

ReadyMix (KAPA Biosystems, KK2601) and IS PCR primers were used to amplify cDNA, after which cDNA was purified with Agencourt AMPure XP beads (Beckman Coulter, A63881). cDNA quality was checked with Agilent Bioanalyser 2100 using Agilent High Sensitivity DNA chip and quantity was measured with Quant-iT™ Picogreen double-stranded DNA assay kit (Thermo Fisher P7589). Pooled libraries of four replicates per sample were prepared using the Illumina Nextera XT DNA preparation kit. Amplified libraries were purified using Agencourt AMPure XP beads, quality-checked using Agilent high-sensitivity DNA chip on Agilent Bioanalyser 2100 and quantified using KAPA qPCR quantification kit (KAPA Biosystems, KK4824). Sequencing of three replicates per condition were performed on the Illumina Hi-Seq 4000 (single end, 50 bp read length) by the CRUK Cambridge Institute Genomics Core facility.

**Bioinformatic analysis of RNAseq**. RNAseq reads were aligned to Mus musculus genome (Ensembl version 38.81) using GSNAP (version 2015-09-29) with parameters (-B 5 –t 24 –n 1 –Q –N 1). Reads in features were counted with htseq-count (HTSeq version 0.5.3p3) with the parameter (-s no). Quality control was performed with the following cut-offs: more than one and a half million uniquely mappable reads, <20% of reads mapping to mitochondrial genes over mitochondrial + nuclear genes and >8500 high coverage genes identified. Counts were normalized using size factors as calculated by DESeq2 using a 10% FDR, and then log10 transformed. Highly variable genes were selected using the method described by Brennecke et al. PCA was then performed in R using the prcomp function.

Gene set enrichment analyses were performed as described[74] (http://www.broadinstitute.org/gsea/index.jsp), using a weighted statistic, ranking by signal to noise ratio, 1000 gene-set permutations, and a custom gene set database gene lists manually compiled from the literature.

GEO accession number—GSE94078

**Statistical analysis and reproducibility**. We used similar response variables for the calculation of sample size. For each hypothesis, the total cell number and composition of the different cell populations was determined by flow cytometry, in peripheral blood (after each treatment) and in the bone marrow (at the end of the experiment). The compared groups were set similarly in all procedures. For each scenario, we established the number of conditions in which to test the hypothesis, and two groups were randomly assigned to each condition (control group and experimental group). Results were scored blindly. Data shown in figures are means ± SEM; $n$ and $p$-values are indicated in every single figure. One-way ANOVA and Bonferroni comparison were used for multiple group comparisons, and unpaired two-tailed $t$-tests for two-group comparisons. The data met the assumptions of the tests. Significant statistical differences between groups were indicated as: $*P < 0.05$; $**P < 0.01$; $***P < 0.001$. Statistical analyses and graphics were carried out with GraphPad Prism software and Microsoft Excel.

**Reporting summary**. Further information on research design is available in the Nature Research Reporting Summary linked to this article.

## Data availability

The RNAseq data generated in this study have been deposited in the GEO database under accession code GSE94078. Source data are provided with this paper.

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

## Acknowledgements

We thank M. Airaksinen for *Gfra2$^{-/-}$* mice; C. Kapeni, A. Rodríguez-Romera., J.-Y. Lee and other members of the S.M.-F group, A.R. Green, D. Pask, T. Hamilton, W.W.Y. Lau and E. Diamanti (University of Cambridge) for support; the Central Biomedical Services and Cambridge NIHR BRC Cell Phenotyping Hub for technical assistance; S.G. was supported by the NIH-OXCAM Program and the Gates Cambridge Trust. A.G.-G. received fellowships from Ramón Areces and LaCaixa Foundations. C.K. was supported by Marie Curie Career Integration grant H2020-MSCA-IF-2015-70841. This work was supported by core support grants from MRC to the Cambridge Stem Cell Institute; National Health Service Blood and Transplant (United Kingdom), European Union's Horizon 2020 research (ERC-2014-CoG-648765), MRC-AMED grant MR/V005421/1 and a Programme Foundation Award (C61367/A26670) from Cancer Research UK to S.M.-F. This research was funded in part by the Wellcome Trust [203151/Z/16/Z]. For the purpose of Open Access, the authors have applied a CC BY public copyright license to any author accepted manuscript version arising from this submission.

## Author contributions

C.F., A.G.-G., C.K., S.G. and S.M.-F. performed experiments and analyses and prepared figures. J.L.R. and J.A.P.-S. retrospectively analysed human data. Z.F. and B.G. helped with RNAseq and its analysis. S.M.-F. designed and supervised the study and wrote the manuscript. All authors edited the manuscript.

## Competing interests

The authors declare no competing interests.
