## [Peer Review File · Nature Communications]

Cholinergic signals preserve haematopoietic stem cell quiescence under stressEditorial Note: Parts of this Peer Review File have been redacted as indicated to maintain the confidentiality of unpublished data.

Reviewers' comments:

Reviewer #1 (Remarks to the Author):

General Comments

The manuscript by Garcia-Garcia, et. al., from the group of Mendez-Ferrer is in an important and relevant area of research. However, there are numerous problems with the paper that greatly dampens enthusiasm for the work from a group that has made important advances in the past.

Recent progress shows that quiescent hematopoietic stem cells (HSCs) reside at the top of the hematopoietic hierarchy and harbor the highest self-renewal and long-term reconstitution capacity. Thus, understanding mechanisms regulating HSC quiescence would expand our knowledge of HSC biology and has the potential to improve clinical application for hematopoietic cell transplantation. In the present study, the authors propose that the sympathetic cholinergic nervous system regulates HSC quiescence and self-renewal. However, the data presented in the manuscript is not sufficient to support their conclusions. Much of the data depends on phenotypic characterization of HSCs with a large paucity of functional data on the HSCs as assessed by limiting dilution engrafting transplant data. This is a major flaw with the paper as presented, especially in context of situations of stress-induced changes in hematopoiesis where phenotype does not always correspond with the functional capabilities of the HSCs.

Specific Comments

1. Page 5 Line 114, "Supporting this possibility, reduced HSC numbers in perinatal BM normalized in adulthood (Fig. 3e-i)". I do not see how the authors draw this conclusion from Fig. 3e-i. The statistics showed no significant difference between Gfra2 +/- and -/-.
2. In Fig 3 the authors used LSKCD150+CD41- to define HSC. Then in Fig 4 they used LSKFlt3-, which is not correct to be used to define HSC.
3. Fig. 3b,c, the authors claimed that Gfra2 knockout mice showed less HSCs in the bone marrow one week after birth, suggesting that Gfra2-dependent sympathetic cholinergic fibres might contribute to the developmental bone marrow HSC colonization. The phenotypic HSC data should be proven by transplantation.
4. Page 5 Line 121, "the frequency of quiescent HSPCs (G0 to G1 ratio) was 4-fold higher in T endosteal marrow", however in their transplantation data (Fig7b,c), engraftment levels showed no differences between endosteal and non-endosteal fractions even when the same number of cells were transplanted. This seems very contradictory.
5. In Fig 4b, the authors showed no difference of G0/G1 ratio between Gfra2 +/+ and -/- for non-endosteal fraction, but in Fig 7 the authors showed that Gfra2 -/- had HSC self-renewal defects even in the non-endosteal fractions. This also appears contradictory. It seems Gfra2 affected HSC engraftment independent of its function in regulation of HSC quiescence.
6. In Fig 4c, the authors showed that 6OHDA increased HSC quiescence in endosteal BM. Functional engrafting assays by doing in vivo transplantation are needed to support this conclusion.
7. In Fig 5k, the authors showed that nicotinic antagonist could mimic the Cxcl12 phenotype of Gfra2 -/-. Does this antagonist affect HSC quiescence and self-renewal?
8. In Fig7, hematopoietic reconstitution was only assessed in the peripheral blood of recipient mice.

Reconstitution levels in the BM would be more relevant and need to be shown. To draw conclusions from transplantation experiments, the authors must perform limiting dilution assays to quantify functional HSC frequency in Gfra2^{+/-} and ^{-/-}.

9. In Fig 7d, the authors transplanted CD45.2⁺ donor cells into CD45.2⁺ recipients. Thus, they could not distinguish donor contributions from host recovery. They should use another congenic recipient (e.g. CD45.1⁺).

10. In order to prove that decreased HSC quiescence in Gfra2 knockout mice is caused by extrinsic mechanisms, the authors need to transplant wild type bone marrow cells or HSCs into wild type and Gfra2 knockout recipient mice.

11. Cholinergic nicotinic antagonist decreased bone marrow Cxcl12 content by 40%. The authors should check if this antagonist changes the quiescence of endosteal HSCs.

12. Gfra2 knockout mice and Chrna7 knockout mice showed augmented hematopoietic recovery after 5-FU challenge. Other stress situations such as recovery after irradiation needs to be evaluated?

13. The authors need perform a limiting dilution assay to quantify the repopulating HSCs in Gfra2 ^{-/-} mice and Chrna7 ^{-/-} mice.

14. Page 7. If Gfra2^{-/-} HSC are hyperproliferative, one wouldn't expect more HSC numbers after treatment of mice with 5FU. Explain.

15. Do Gfra2^{+/-} mice have any functional phenotype since you used them for a control as noted on line 187 of page 8.

16. Page 5, line 107. What about protein levels?

17. Show all data where you note data not shown.

18. Fig. 7b,c, why are the authors only showing the data of donor derived monocytes? What about whole CD45.2 chimerism and other cell populations? Besides, for non-endosteal cells, why was there no difference in the primary transplantation, but significant difference in the secondary recipients?

Other Comments

1. The English presentation can be improved.

2. Make sure that the Figures and Supplemental Figures are cited in the correct order.

3. The authors failed to describe current knowledge about regulation of HSC quiescence. More information on this is needed.

4. Fig. 1c. The authors used red fluorescence labeled antibody to stain GFP, and at the same time used green fluorescence labeled antibody to stain VACHT. Will the fluorescence of GFP interfere with the VACHT staining?

5. Fig. 1 l and m. From the pictures, it looks like myelinated axons were also reduced and significantly enlarged. This should be clarified.

--

Reviewer #2 (Remarks to the Author):

This is a very interesting manuscript. Cholinergic sympathetic neurons are a fascinating population of neurons and there is a lot of interest in what they do. The idea that they are the specific population of sympathetic neurons that regulate haematopoietic stem cell quiescence and self renewal is exciting.

Haematopoietic stem cell quiescence and self-renewal are not the main focus of my research, so I will confine my comments to data and interpretations related to the neuronal analyses, and cholinergic denervation/inhibition presented in this paper.

The authors posed the question of whether cholinergic innervation in bone exists. VACht is used by the authors as a marker for cholinergic innervation and is appropriate, but it labels all cholinergic neurons not just those that are sympathetic. Importantly, the authors have claimed to show with ChAT-IRES-cre mice intercrossed with Ai35D reporter mice (to genetically label cholinergic neurons) that the VACht labeling they report at the endosteal surface is indeed related to cholinergic innervation. However, the panels in fig 1c-f that they refer to are not of high enough power to show convincing labelling of cholinergic nerve terminals, and the supplementary figure 2b, presented at higher power, shows at best poor GFP labelling in structures that do not at all look like nerve terminal endings. This draws attention to whether their labelling of cholinergic nerve terminals in bone is adequate. Whilst some controls were performed (for example primary antibody omission), other more appropriate controls (for example pre-adsorption or validation in knockout animals) were not. There was not any information about specificity controls for the goat polyclonal Merck VACht antibodies available on the Merck website either.

The authors used a GFRa2^{-/-} mouse model to study cholinergic regulation of HSCs. GFRa2 is present in both sympathetic and sensory neurons, including in sensory neurons that innervate the bone marrow. The authors also cite/provide evidence that GFRa2 mRNA is present on HSCs themselves. In addition, whilst they showed that these mice have reductions in VACht labelling, there was still a lot of VACht labelling present. These findings highlight the notion that the use of GFRa2 is not adequate as a marker for cholinergic sympathetic neurons alone, and that the use of GFRa2^{-/-} mice is therefore confounded by effects on other cell types (not just sympathetic cholinergic neurons) that might be causing the effects they are seeing on HSC function.

Lineage tracing using the TH-cre; Ai14D mice was used to confirm the VACht labelling was at least in some cases (those presented in figure 2c-f) of sympathetic origin. This was useful. However, in supplementary figure 4 the authors also show evidence that there are many VACht⁺ profiles that are not coincident with TH⁺ sympathetic neuronal labeling in these same TH reporter mice. Chemical sympathectomy with 6OHDA reduced but did not abolish VACht labelling. These findings reinforce the idea that there is other cholinergic innervation, not of sympathetic origin, that could be contributing to the effects seen on HSC function.

The electron microscopy to show apparent reduced numbers of unmyelinated fibers is presented with only a single sample and without quantification, so it is difficult to gauge if the result occurs from sampling bias.

Whilst sympathetic cholinergic innervation exists and is documented clearly in the limbs of mice, there is still some debate about whether it exists in other species, particularly the rat. The authors again use GFRa2 as the marker for cholinergic innervation in the rat exercise stress experiments, but there is no evidence that this is appropriate for this species. This needs to be confirmed, along with providing clear evidence of cholinergic innervation in the femur of rats, if one is to accept the interpretation of the authors relating to the role of sympathetic cholinergic neurons on stress induced effects to HSC function. In addition, the sympathectomy in the rats was performed in the early postnatal period, before the proposed cholinergic switch documented in mice, and so the changes observed could be the result of both adrenergic and cholinergic sympathetic innervation.

Overall, many of the findings are consistent with the hypothesis that sympathetic cholinergic neurons

might regulate HSC proliferation and self renewal, but given the limitations of the findings described above, it is difficult to see how this could be asserted with the strictest of confidence. The effects ACh through nicotinic receptors on BM stromal cells could be the result of ACh released from cells other than of sympathetic cholinergic origin.

--

Reviewer #3 (Remarks to the Author):

This manuscript is the first to demonstrate a switch in bone marrow sympathetic innervation from noradrenergic to cholinergic within first week after birth in mice with functional consequences for HSC quiescence and self-renewal. The manuscript introduces novel concepts regarding maintenance of haematopoietic stem cell (HSC) quiescence. Experimental design is elegant with extensive use of lineage tracing and gene-deleted animals to characterise nerve fibres innervating the bone marrow complemented by functional transplant assays.

General Comment

1. The cholinergic switch and innervation studies described in this manuscript are innovative, interesting and well performed. However a role for bone marrow cholinergic nerve fibres in regulation of haematopoiesis is limited to direct role on Nestin+ MSC and CXCL12 as sole mediator. Why are possible roles for cholinergic signalling in regulation of other niche cell populations (eg. endothelial cells, tissue-resident macrophages and Nestin negative MSC) largely ignored? Most of these niche cells also express cholinergic receptors and reported to regulate HSC via CXCL12 as well as other factors.

Specific comments on Figures and associated text are listed stepwise below.

Fig 1. (g-j) Comment regarding 'Endomucin .. marks endothelial cells'. Endomucin is commonly utilised to distinguish venous endothelium (endomucin positive) from arteriole vasculature (largely negative -see M Corada et al., Nature Comms 2013, PMID 24153254 and Fig2 in review article Arterial versus venous endothelial cells, PMID: PMC4105978). This is relevant as bone marrow HSC residing near arterioles (which may not stain with endomucin) are reported to have differing self-renewal potential compare to HSC near the (endomucin positive) post-capillary venules. Could this impact interpretation of the staining data in Figure 1 g-h as endomucin may only be staining a portion of the vasculature (not arterioles) at the bone marrow endosteum?

SuppFig2 trabecular bone marrow stains (a-d) with associated controls (h-j). These data suggest that genetically-traced Cholinergic neurons co-stain with CD31+ endothelial cells at the endosteum. This is not mentioned in manuscript text and suggest the manuscript's final discussion - regarding tentative direct role of Nestin-GFP+ MSC in regulating HSC quiescence via direct cholinergic sympathetic signalling - should be more nuanced as several other cells of the HSC niche, including endothelium and macrophages (see example refs below) may also express cholinergic receptors.

Cooke JP et al., 2008. Trends in Cardiovascular Medicine Endothelial Nicotinic Acetylcholine Receptors and Angiogenesis (PMC 2673464)

Chernyavsky AI, Int Immunopharmacol. 2010. Auto/paracrine control of inflammatory cytokines by acetylcholine in macrophage-like U937 cells through nicotinic receptors.

Fig 3 and associated text.

Line 111 comment 'HSC...enter quiescence in a Cxcl12-dependent manner ..in postnatal weeks 3 to 4' ... Reference 23 has been misquoted here. Although correct that reference 23 (PMC1578623) does indeed state that HSC enter quiescence between postnatal weeks 3 to 4 (see Ref 23, Fig3), at no stage does it state that this is in a 'CXCL12-dependant manner'. Instead reference 23 claims that post-natal HSC had an engraftment defect while cycling (which could be rectified by administering

CXCL12 blockade to host) and that FACs sorted 3wk postnatal mouse HSC separated based on cell-cycle (using phenotype lin-Sca-1+, CD43+, Mac1+) show different levels of CXCL12 mRNA expression in themselves. This is quite distinct from 'HSC enter quiescence in CXCL12 dependant manner.' Please remove sentence.

Similarly the comment (also line 111) 'HSC abruptly enter quiescence... between postnatal weeks 3 to 4....concurrently with the completion of the cholinergic switch' ... has also not been yet proven. Figs 1 to 3 in current manuscript were performed in 1 week old mice only. Later figures refer to adult mice or rats. No time-course to demonstrate that cholinergic switch occurs 'concurrently' with abrupt HSC quiescence has been performed. Many other biological processes also happen to mice between post-natal week 3 to 4 including weaning and exposure to new food and associated bacteria, a shift in microbiota and immune settings etc.

As the data contained in Fig 1 to 4 are all correlative only, perhaps the wording in places could be more nuanced to reflect this such as the statement in line 108 should be clearly explained as hypothesis.

Fig 4d. (RE Gfra2 chimeras, lines 134 to 139). Normally to confirm an effect is HSC extrinsic, wildtype HSC would be also injected into Gfra2-/- recipients . Without data from such a reverse chimera it is not possible to ... 'clearly dissect haematopoietic cell autonomous and non-autonomous effects' .. as stated in line 134. The wording 'clearly dissect' should be modified accordingly. The reasons why the reverse chimera was not shown could include irradiation-induced nerve damage in recipient mice confusing the result. However if so this raises further questions of whether BM cholinergic nerves do survive the conditioning regimes used in these transplants and how data from Fig4d and should then be interpreted.

Fig 5(a-h) statement on line 140 'We found that endosteal Nes-GFP+ cells are innervated with cholinergic fibres'. This is possibly true from the data shown, however in supplementary Fig2(a-e) there is far more convincing co-localisation with CD31+ endothelial cells which also express nicotinic receptors as do macrophages (see example refs provided in SuppFig2 above). This is not at all mentioned in text and raises questions on why wasn't a more open approach taken to investigate the contributions of a range of known HSC niche cells including; (1) other MSC populations especially non-Nestin+ CXCL12 abundant PGDFRa+ MSC (as described in Greenbaum et al., Nature 2013), (2) bone marrow endothelium and (3) tissue resident perivascular macrophages.

Fig5(i). Control Nes-GFP neg cells used for CXCL12 Q-PCR were of phenotype CD45-CD31-Ter119-. A far more appropriate and relevant Nes-GFP neg control cell population could have been achieved by Sca-1+ PDGFRa+ co-staining. These important non-Nestin MSC also express abundant CXCL12 and depletion of CXCL12 within this population which are within the Nestin neg Prx1-Cre x fl CXCL12 MSC population results in a complete loss of dormant BM phenotypic HSC and severely reduced reconstitution potential in primary transplant (see Ref Greenbaum A D et al., Nature 201). This relevant study should be included in references.

Reviewers' comments:**Reviewer #1 (Remarks to the Author):**

General Comments

The manuscript by Garcia-Garcia, et. al., from the group of Mendez-Ferrer is in an important and relevant area of research. However, there are numerous problems with the paper that greatly dampens enthusiasm for the work from a group that has made important advances in the past.

Recent progress shows that quiescent hematopoietic stem cells (HSCs) reside at the top of the hematopoietic hierarchy and harbor the highest self-renewal and long-term reconstitution capacity. Thus, understanding mechanisms regulating HSC quiescence would expand our knowledge of HSC biology and has the potential to improve clinical application for hematopoietic cell transplantation. In the present study, the authors propose that the sympathetic cholinergic nervous system regulates HSC quiescence and self-renewal. However, the data presented in the manuscript is not sufficient to support their conclusions. Much of the data depends on phenotypic characterization of HSCs with a large paucity of functional data on the HSCs as assessed by limiting dilution engrafting transplant data. This is a major flaw with the paper as presented, especially in context of situations of stress-induced changes in hematopoiesis where phenotype does not always correspond with the functional capabilities of the HSCs.

We thank the reviewer for considering our manuscript in an important and relevant area of research and for highlighting our previous contributions. We agree that limiting dilutions are needed to quantify HSC numbers exactly, which we have not done or claimed here. Many studies have successfully used long-term competitive repopulation assays without limiting dilutions (as we did and showed in Fig. 7) to suggest differences between two groups (i.e Signer, Morrison *et al.*, Nature 2014; Dong, Scadden *et al.*, Nature 2016). We also agree that stress-induced changes in haematopoiesis may change the immunophenotype of HSPCs and we actually discussed this important concept in a review article (Mendez-Ferrer et al. Annals NY Acad Sci 2015; section entitled "Considerations about stem cell immunophenotype and function" on pages 33-36). This is precisely the reason why we carefully validated some of our main contentions (increased HSPC proliferation and decreased HSC self-renewal) through multiple approaches (cell cycle analysis, RNAseq from purified HSCs, long-term competitive repopulation assays, response to myeloablation and bacterial infection as a new additional stress setting to specifically respond to point 12 by the Reviewer, explained in detail below). However, we would like to clarify that whereas we used stress haematopoiesis challenges to reveal the function of quiescent HSPCs (as most studies in this area), immunophenotypically-defined HSPCs in Fig. 3, Fig. 4, Supp. Fig. 5 and Supp. Fig. 6 correspond to unperturbed mice.

To further convince the reviewer, we have investigated in a separate study the role of cholinergic signals in day/night oscillations in HSC traffic. We found accumulation of HSPCs (measured as CFU-C) in circulation at night. To measure circulating HSCs at Zeitgeber time (ZT)13 (13h after light onset, with 12h light:12h darkness cycles), we performed competitive long-term repopulation assays using limiting dilutions of blood harvested at ZT13. *Gfra2*^{-/-} mice showed 3.5 times more HSCs in bloodstream early at night (**Figure R1**).

Figure R1. Limiting dilution assays to quantify functional HSCs. HSCs, measured by long-term competitive repopulation assay, in peripheral blood harvested at ZT13 from *Gfra2*^{-/-} mice (green) and control *Gfra2*^{+/+} mice (purple). The log fraction of mice which failed reconstitution is plotted against the transplanted blood volume using ELDA software; Likelihood ratio test of single-hit model, $p = 0.006$, Chi square test. Blood HSC concentrations are indicated ($n = 5$).

Specific Comments

1. Page 5 Line 114, “Supporting this possibility, reduced HSC numbers in perinatal BM normalized in adulthood (Fig. 3e-i)”. I do not see how the authors draw this conclusion from Fig. 3e-i. The statistics showed no significant difference between *Gfra2* ^{+/+} and ^{-/-}.

There appears to be some confusion in this point. Reduced HSC numbers in perinatal BM were shown in Fig. 3b and quoted in L. 105-106 (now L. 99-100 in the revised manuscript): “*Gfra2*^{-/-} mice exhibited a normal number of foetal liver HSCs (Fig. 3a) but showed ~40% less HSCs in the BM one week after birth (Fig. 3b, c).” The reviewer is correct that at adult stage there are no differences (Fig. 3e-i), and this is what we meant in the quoted sentence: (L112-114) “Therefore, we hypothesized that cholinergic fibres might induce HSC quiescence and a lack of the fibres might cause increased HSC proliferation. Supporting this possibility, reduced HSC numbers in perinatal BM normalized in adulthood (Fig. 3e-i).” The reason that HSC numbers normalize from perinatal to adult stage is that HSPCs are more proliferative in *Gfra2*^{-/-} mice, as demonstrated by cell cycle analysis (Fig. 4b), HSC response to acute myelosuppressive stress (Fig. 6a) and RNAseq (new Fig. 6c).

2. In Fig 3 the authors used LSKCD150⁺CD41⁻ to define HSC. Then in Fig 4 they used LSKFlt3⁻, which is not correct to be used to define HSC.

Whenever possible (like in Fig. 3) we used SLAM markers to immunophenotypically define HSCs. However, our H42/Pyronin Y cell cycle studies require a minimum of 2,000 cells to obtain a reliable cell cycle profile. This is not feasible using SLAM markers to purify HSCs from the endosteal compartment due to the low cell number available. For this reason, we chose LSK Flt3⁻ cells as a validated HSC-enriched population, which includes LT- and ST-HSCs (and excludes more committed MPPs). To be more accurate, in the revised manuscript we have called “HSCs” only those identified by SLAM markers, and we have referred to HSC-enriched populations/cells or “HSPCs” when the use of SLAM markers was not possible due to the low cell number available and LSK Flt3⁻ cells were used instead.

3. Fig. 3b,c, the authors claimed that *Gfra2* knockout mice showed less HSCs in the bone marrow one

week after birth, suggesting that *Gfra2*-dependent sympathetic cholinergic fibres might contribute to the developmental bone marrow HSC colonization. The phenotypic HSC data should be proven by transplantation.

The previous comment by the reviewer suggests that SLAM markers (LSKCD150⁺CD41⁻) are a better choice than LSKFlt3⁻ to define HSCs with high purity. We also agree that the important finding that SLAMs markers (which have been validated and broadly used in numerous studies) allow to estimate HSC numbers as a surrogate of long-term transplantation experiments. Therefore, SLAM and other markers have been developed precisely as a surrogate of these transplantations and are broadly used in most publications to estimate HSCs without needing to perform so many transplantations. We provided transplantation data in Figure 7 to functionally prove some of the main concepts. Unfortunately, the requested transplantations with limiting dilutions are not feasible for multiple experiments within the scope of reasonable revisions and would be hard to justify ethically due to the high number of mice needed for a purpose which might not be essential in every case.

4. Page 5 Line 121, “the frequency of quiescent HSPCs (G0 to G1 ratio) was 4-fold higher in WT endosteal marrow”, however in their transplantation data (Fig7b,c), engraftment levels showed no differences between endosteal and non-endosteal fractions even when the same number of cells were transplanted. This seems very contradictory.

As very pertinently and accurately mentioned by the reviewer in the first para., quiescent HSCs reside at the top of the hematopoietic hierarchy and harbour the highest self-renewal and long-term reconstitution capacity. Therefore, we would only expect reduced reconstitution after long-term transplantation into primary recipients, and only in the endosteal BM (which is the one regulated by the cholinergic signals). This is exactly what we observed and showed in Fig. 7b. In contrast, there are no differences in non-endosteal long-term reconstitution in primary recipients (Fig. 7c), because HSCs are not regulated by cholinergic signals in this region.

5. In Fig 4b, the authors showed no difference of G0/G1 ratio between *Gfra2* *+/+* and *-/-* for non-endosteal fraction, but in Fig 7 the authors showed that *Gfra2* *-/-* had HSC self-renewal defects even in the non-endosteal fractions. This also appears contradictory. It seems *Gfra2* affected HSC engraftment independent of its function in regulation of HSC quiescence.

We perfectly understand this comment because we were also surprised to observe that non-endosteal HSCs from *Gfra2*^{-/-} mice exhibit reduced HSC engraftment in secondary recipients. It is likely that the mechanical separation of non-endosteal HSCs also carries some endosteal cells susceptible to cholinergic regulation, which might therefore have their function compromised (albeit to a lower extent, since it does not show in primary recipients, even 6 months after transplantation; Fig. 7c). Alternatively, it is possible that some cells isolated originally from endosteal or non-endosteal BM redistribute to some degree in the BM of primary recipients, which might make the spatial separation in secondary recipients very difficult. However, we can confidently exclude the possibility that *Gfra2* directly affects HSC engraftment because adult SLAM HSCs do not express *Gfra2* protein (Supplementary Figure 6b, c). Moreover, in the separate trafficking study mentioned above we found that HSPC homing to myeloablated recipients is not different in *Gfra2*^{-/-} donor or recipient mice (**Figure R2**).

Figure R2. Unaltered HSPC homing in *Gfra2*^{-/-} donor/recipient mice.

(A) Scheme showing the protocol used for HSPC BM homing assay with irradiation (12 Gy).

(B) Frequencies of HSPCs (homing efficiency) at ZT2 that homed during the night to the BM after i.v. transplantation in lethally-irradiated mice (12 Gy) (to promote homing) at ZT10. *Gfra2*^{-/-} mice and control *Gfra2*^{+/-} mice were used as

donor or recipients in all combinations. Homing efficiency is determined as the percentage of CFU-Cs obtained from BM harvested from irradiated mice in comparison to CFU-Cs obtained from a non-irradiated mouse. The number of mice used is indicated inside the bars.

6. In Fig 4c, the authors showed that 6OHDA increased HSC quiescence in endosteal BM. Functional engrafting assays by doing *in vivo* transplantation are needed to support this conclusion.

We respectfully disagree. Quiescence (per se) is usually studied through cell cycle experiments, rather than through *in vivo* transplantation.

7. In Fig 5k, the authors showed that nicotinic antagonist could mimic the Cxcl12 phenotype of *Gfra2*^{-/-}. Does this antagonist affect HSC quiescence and self-renewal?

We have performed the requested cell cycle analysis of WT mice treated with nicotinic antagonists (Figure R3). Although not significant, a trend is observed towards reduced HSC quiescence in mice treated with nicotinic antagonists. However, this might be due to insufficient antagonism since we already showed that reduced HSC quiescence correlates with decreased Cxcl12 expression in mice lacking cholinergic nicotinic $\alpha 7$ receptor (*Chrna7*^{-/-}; Fig. 5n-o), which is a more specific and profound loss of function approach.

Figure R3. G0/G1 ratio of cell cycle phases in HSCs isolated from WT mice *in vivo* treated with nicotinic antagonists or saline. Data are means \pm SEM. Unpaired two-tailed t test. n = 3-4.

8. In Fig7, hematopoietic reconstitution was only assessed in the peripheral blood of recipient mice. Reconstitution levels in the BM would be more relevant and need to be shown. To draw conclusions from transplantation experiments, the authors must perform limiting dilution assays to quantify functional HSC frequency in *Gfra2*^{+/-} and *-/-*.

We hope that our response to the similar comment in the introductory paragraph was satisfactory.

9. In Fig 7d, the authors transplanted CD45.2+ donor cells into CD45.2+ recipients. Thus, they could not distinguish donor contributions from host recovery. They should use another congenic recipient (e.g. CD45.1+).

We agree with the reviewer that using CD45.1 recipient mice is a safer choice to circumvent potential issues of residual host hematopoiesis. We did use CD45.1 recipient mice for primary transplantations. Unfortunately, we had to use CD45.2 recipient mice for the large number of secondary transplantations performed. We agree that endogenous recovery might be a problem given the long proliferative history of donor cells. For this reason, we compared the frequency of CD45.2+ monocytes (to exclude long-term surviving lymphocytes from the analysis) between primary and secondary recipients. Overall, CD45.2+ monocyte chimerism increases with time (especially in secondary recipient mice transplanted with control *Gfra2*^{+/-} cells), suggesting a significant expansion of recipient-derived residual hematopoiesis (the 6w-old host cells likely outcompeted the 35w-old donor cells). However, additional competition with host-derived cells would equally affect all groups of mice and should not affect the comparison between the two groups. In addition, secondary recipients transplanted with *Gfra2*^{-/-} BM cells exhibit 6-10 months after transplantation a 2-fold-reduction of CD45.2+ monocyte chimerism, circulating red blood cells (RBC) and platelets, correlating with reduced mouse survival (Fig. 7e-g). These data strongly suggest a critical haematopoietic failure in secondary recipients transplanted with *Gfra2*^{-/-} BM cells (despite the presence of residual host hematopoiesis).

10. In order to prove that decreased HSC quiescence in *Gfra2* knockout mice is caused by extrinsic mechanisms, the authors need to transplant wild type bone marrow cells or HSCs into wild type and *Gfra2* knockout recipient mice.

To dissect between cell-autonomous (intrinsic) and non-cell autonomous (extrinsic) contributions we did perform reciprocal BM transplantations using *Gfra2*^{+/-} and *Gfra2*^{-/-} mice as donor and recipients in all possible combinations. In agreement with an altered BM microenvironment in *Gfra2*^{-/-} mice, only *Gfra2*^{-/-} recipient mice exhibit a significant decrease in *Cxcl12* expression (**Figure R4** shown introduced the revised manuscript as new Fig. 4d), regardless whether these mice were transplanted with WT, *Gfra2*^{+/-} or *Gfra2*^{-/-} BM cells. Although we did not perform cell cycle analysis in these mice, we have already shown that reduced *Cxcl12* level correlates with decreased HSPC quiescence. Therefore, our data strongly suggest that decreased HSPC quiescence in *Gfra2*^{-/-} mice is caused by impaired cholinergic signalling in the BM microenvironment.

Figure R4. *Cxcl12* concentration in BM extracellular fluid 16 weeks after BM transplantation in lethally-irradiated mice. Control *Gfra2*^{+/-} and *Gfra2*^{-/-} mice were used as donor or recipients in all combinations. Data are means \pm SEM. One-way ANOVA and Bonferroni comparisons. n = 5. * p < 0.05; ** p < 0.01.

11. Cholinergic nicotinic antagonist decreased bone marrow *Cxcl12* content by 40%. The authors should check if this antagonist changes the quiescence of endosteal HSCs.

Please kindly see response to the similar question in the comment number 7.

12. *Gfra2* knockout mice and *Chrna7* knockout mice showed augmented hematopoietic recovery after 5-FU challenge. Other stress situations such as recovery after irradiation needs to be evaluated?

We have already shown a differential HSC response to acute myelosuppressive stress in models of impaired cholinergic neurotransmission (Fig. 6a), and the relevance of the cholinergic signalling under chronic stress in rats (old Fig. 6b-d). To further convince the reviewer, we have tested haematopoietic reconstitution in a different stress scenario (bacterial infection), which also demonstrates poor survival of *Gfra2*^{-/-} mice under haematopoietic stress (**Figure R5**, added to the revised manuscript as new Fig. 6B).

Figure R5. Survival rate of WT or *Gfra2*^{-/-} mice after BM transplantation and infection induced by *Streptococcus agalactiae*. Data are means \pm SEM. Gehan-Breslow-Wilcoxon test for survival curves. $n = 25-27$. ** $p < 0.01$.

13. The authors need perform a limiting dilution assay to quantify the repopulating HSCs in *Gfra2*^{-/-} mice and *Chrna7*^{-/-} mice.

We have not quantified or reported absolute HSC numbers, which strictly requires limiting dilution assays. However long-term (6 month) bone marrow transplantation, as shown in Fig. 7, has been considered sufficient in most studies to suggest differential HSC enrichment between two groups, as we demonstrate in the present study.

14. Page 7. If *Gfra2*^{-/-} HSC are hyperproliferative, one wouldn't expect more HSC numbers after treatment of mice with 5FU. Explain.

Please be aware that *Gfra2*^{-/-} mice have increased fraction of endosteal HSCs in G1, not in S phase. 5-FU kills cells in S phase, which is not significantly different in HSCs from *Gfra2*^{-/-} and control mice. Therefore, we would not expect 5-FU to kill more HSCs in *Gfra2*^{-/-} mice. We showed and explained that the HSPC cycling difference between *Gfra2*^{-/-} mice and control mice concerns the G0-G1 transition. To clearly illustrate that all the cell cycle results were shown as G0/G1 fraction. As indicated in the revised manuscript (L. 157-160), "Consistent with the increased fraction of endosteal HSPCs in the G1 phase of the cell cycle (when these cells are insensitive to 5-fluorouracil (5-FU) but are ready to enter G2), *Gfra2*^{-/-} mice and *Chrna7*^{-/-} mice exhibited augmented haematopoietic recovery in the stress phase acutely after 5-FU (days 14-18)."

15. Do *Gfra2*^{+/-} mice have any functional phenotype since you used them for a control as noted on line 187 of page 8.

We understand and shared this concern regarding the use of heterozygous *Gfra2*^{+/-} mice as control of *Gfra2*^{-/-} mice when we started this project. We carefully selected the best possible control mice for the

experiments. Whenever it was possible, we chose heterozygous mice as ideal littermate controls (frequently coming from the same cages) because circulating white blood cells (WBC) and response to 5-FU were similar in *Gfra2*^{+/+} and *Gfra2*^{+/-} mice (Figure R6). Thus, *Gfra2*^{+/-} mice can be used as ideal controls of their littermate *Gfra2*^{-/-} mice for these experiments.

Figure R6. *Gfra2*^{+/+} and *Gfra2*^{+/-} mice exhibit similar circulating white blood cell (WBC) counts at night (left) and similar hematopoietic recovery after myeloablation with 5-FU. Data are means \pm SEM. Unpaired two-tailed t test (left) and multiple t test (right); n = 5-20.

16. Page 5, line 107. What about protein levels?

As requested we have measured Cxcl12 protein in postnatal BM from control *Gfra2*^{+/-} and *Gfra2*^{-/-} mice. Following the reduction of Cxcl12 mRNA gene expression level, Cxcl12 protein level was also lower in 1-week old (P7) *Gfra2*^{-/-} mice. We have included this data in the revised manuscript together with the Cxcl12 mRNA expression data and removed *Kitl* mRNA expression data which is unaltered (Figure R7 added to the revised manuscript as new Fig. 3d).

Figure R7. Cxcl12 protein concentration in BM extracellular fluid (BMECF) in 1-week old (P7) *Gfra2*^{+/-} and *Gfra2*^{-/-} mice. Data are means \pm SEM. Unpaired two-tailed t test. n = 4-6. * p < 0.05.

17. Show all data where you note data not shown.

This happened only twice in the text and was only intended to simplify the manuscript. We did not show that 6OHDA treatment ablates noradrenergic fibres because it has been previously shown in multiple publications. Please see the representative examples illustrating the dramatic reduction of TH⁺ innervation after 6OHDA treatment (Figure R8).

Figure R8. Immunofluorescence of TH (red) and CD31 (blue) in skull BM from control mice treated with 6OHDA postnatally. Scale bar, 100 μm .

The other data previously not shown (7-fold more nucleated cells in WT non-endosteal BM, compared with the endosteal BM) is shown below for clarity (**Figure R9**).

Figure R9. Bone marrow nucleated cells measured in endosteal and non-endosteal fractions from WT mice. Data are means \pm SEM. Unpaired two-tailed t test. $n = 12$. **** $p < 0.0001$

Both results have been now included in the revised manuscript in Supplementary Fig. 3b, c and Supplementary Fig. 5e, respectively.

18. Fig. 7b,c, why are the authors only showing the data of donor derived monocytes? What about whole CD45.2 chimerism and other cell populations? Besides, for non-endosteal cells, why was there no difference in the primary transplantation, but significant difference in the secondary recipients?

As explained above we had to use CD45.2 recipient mice for the large number of secondary transplants performed (we could not possibly get enough CD45.1 mice for these experiments). We agree that endogenous recovery might be a problem given the long proliferative history of donor cells. For this reason we decided to focus on the frequency of CD45.2⁺ monocytes and exclude long-term surviving lymphocytes from the analysis between primary and secondary recipients. It is likely that the mechanical separation of non-endosteal HSCs also carries some endosteal cells susceptible to cholinergic regulation, which might therefore have their function compromised (albeit to a lower extent, since it does not show in primary recipients, even 6 months after transplantation (Fig. 7c), but only in secondary recipients). Alternatively, it is possible that cell isolated originally from endosteal or non-endosteal BM redistribute to some degree in the BM of primary recipients, which might make the spatial separation in secondary recipients very difficult.

Other Comments

1. The English presentation can be improved.

We have carefully revised the English presentation and have it proofread by a native UK scientist.

2. Make sure that the Figures and Supplemental Figures are cited in the correct order.

We apologise because there was a jump and we have now fixed it in the revised manuscript.

3. The authors failed to describe current knowledge about regulation of HSC quiescence. More information on this is needed.

We have expanded the discussion section on the regulation of HSC quiescence.

4. Fig. 1c. The authors used red fluorescence labeled antibody to stain GFP, and at the same time used green fluorescence labeled antibody to stain VACHT. Will the fluorescence of GFP interfere with the VACHT staining?

No, it does not interfere with VACHT fluorescence (green). In fact we used anti-GFP antibody because the endogenous GFP fluorescence was not intense enough to be clearly visualised at such low magnification (purportedly used to show a large BM region and the specific anatomical location of this innervation).

5. Fig. 1 l and m. From the pictures, it looks like myelinated axons were also reduced and significantly enlarged. This should be clarified.

Myelinated and non-myelinated axons were identified and quantitated by two independent neuroscientists with experience in TEM. Additional examples are provided below (**Figure R10**). The two experienced observers identified 27 unmyelinated axons in WT endosteum and only 4-5 in the same region of *Gfra2* KO mice. The conclusions were pragmatically phrased "Whereas myelinated axons appeared unchanged in the same region, unmyelinated (compatible with cholinergic) axons appeared reduced in *Gfra2*^{-/-} mice."

TEM in WT samples

TEM in *Gfra2*^{-/-} samples

Figure R10. Additional examples of unmyelinated axons (arrowheads) and myelinated axon (arrow) revealed by transmission electron microscopy in WT and *Gfra2*^{-/-} BM sections. Note the reduction in *Gfra2*^{-/-} samples.

Reviewer #2 (Remarks to the Author):

This is a very interesting manuscript. Cholinergic sympathetic neurons are a fascinating population of neurons and there is a lot of interest in what they do. The idea that they are the specific population of sympathetic neurons that regulate haematopoietic stem cell quiescence and self renewal is exciting.

Haematopoietic stem cell quiescence and self-renewal are not the main focus of my research, so I will confine my comments to data and interpretations related to the neuronal analyses, and cholinergic denervation/inhibition presented in this paper.

The authors posed the question of whether cholinergic innervation in bone exists. VAcHT is used by the authors as a marker for cholinergic innervation and is appropriate, but it labels all cholinergic neurons not just those that are sympathetic. Importantly, the authors have claimed to show with ChAT-IRES-cre mice intercrossed with Ai35D reporter mice (to genetically label cholinergic neurons) that the VAcHT labeling they report at the endosteal surface is indeed related to cholinergic innervation. However, the panels in fig 1c-f that they refer to are not of high enough power to show convincing labelling of cholinergic nerve terminals, and the supplementary figure 2b, presented at higher power, shows at best poor GFP labelling in structures that do not at all look like nerve terminal endings. This draws attention to whether their labelling of cholinergic nerve terminals in bone is adequate. Whilst some controls were performed (for example primary antibody omission), other more appropriate controls (for example pre-adsorption or validation in knockout animals) were not. There was not any information about specificity controls for the goat polyclonal Merck VAcHT antibodies available on the Merck website either.

We thank the Reviewer for considering the manuscript very interesting.

To further convince this reviewer about the specificity of our cholinergic staining we now show close-up examples of VAcHT staining in thick BM sections from *ChAT-IRES-cre;Ai35D* mice. The data fully supports our previous contentions by clearly showing co-expression of GFP and VAcHT in endosteal BM, where this innervation is most abundant (**Figure R11** and new Fig. 1 and Supplementary Fig. 2). However, we agree that there might be other non-neural structures labelled in *ChAT-IRES-cre;Ai35D* bone and have indicated this possibility in the revised manuscript (P8. L. 207-209).

Additionally, we have performed isotype antibody control staining for VAcHT antibody and the result confirms the specificity of our staining (**Figure R12**).

Figure R11. Immunofluorescence of vesicular acetylcholine transporter (VACHT) to label cholinergic nerve fibers (green) and anti-GFP (red) in adult femoral BM sections from *ChAT-IRES-cre;Ai35D* mice (which have GFP expression in genetically-traced cholinergic neurons). Nuclei were counterstained with DAPI (blue). Yellow arrows indicate co-staining of VACHT and GFP. Scale bar, 100 μ m.

Figure R12. Control isotype staining for VACT using goat isotype (green). Nuclei were counterstained with DAPI (blue). Endothelial cells were labelled with endomucin (red). Scale bar, 100 μ m.

The authors used a *GFRa2*^{-/-} mouse model to study cholinergic regulation of HSCs. *GFRa2* is present in both sympathetic and sensory neurons, including in sensory neurons that innervate the bone marrow.

We are not aware of any study showing *GFRa2* expression in sensory neurons that innervate the bone marrow; please share this evidence with us. In contrast, our findings (Fig 1A-D) in combination with a previous study (Kupari & Airaksinen PlosONE 2014) suggest that virtually all *GFRa2*⁺ sensory neurons in adult mice are *MrgD*⁺ neurons, which are known to exclusively innervate the epidermis (Zylka et al 2005 Neuron). Thus, we would expect little if any *GFRa2*⁺ sensory nerve fibres in adult mouse bone marrow.

The authors also cite/provide evidence that GFRa2 mRNA is present on HSCs themselves.

There might be some confusion since we clearly state in the manuscript that, although a previous study reported *Gfra2* mRNA expression on HSCs, we only detect protein expression in 1% (negligible) of immunophenotypically-defined HSCs (now P5. L125-128 in the revised manuscript).

In addition, whilst they showed that these mice have reductions in VACHT labelling, there was still a lot of VACHT labelling present. These findings highlight the notion that the use of GFRa2 is not adequate as a marker for cholinergic sympathetic neurons alone, and that the use of GFRa2^{-/-} mice is therefore confounded by effects on other cell types (not just sympathetic cholinergic neurons) that might be causing the effects they are seeing on HSC function.

As previously commented, there might be other non-neuronal cells positive for VACHT which might not be affected in *Gfra2*^{-/-} mice. This has been clearly recognized in the revised manuscript (P8. L. 207-209). However, the fact that we detect a significant reduction in VACHT staining in endosteal BM suggest that VACHT staining mostly label cholinergic neuronal structures. Additionally, we have analysed BMSC populations by FACS (Chan CKF *et al.* Cell 2015;160:285-298). We have not observed any significant difference between *Gfra2*^{-/-} and control *Gfra2*^{+/-} mice, again suggesting the absence of other bystander effects in the microenvironment (**Figure R13**).

Figure R13. Unchanged frequencies of stromal cell populations in the BM of *Gfra2*^{-/-} mice. Collagenase I-digested BM samples were incubated with the following fluorescent fluorochrome-conjugated antibodies: CD45, CD31, Ter119, α V integrin, Thy1, 6C3 and CD200. Data are means \pm SEM. Unpaired two-tailed t test. n = 3.

Moreover, we also show that *Chrna7*^{-/-} mice exhibit reduced HSPC quiescence in endosteal BM as shown in *Gfra2*^{-/-} model (Fig. 5o). Therefore, we have validated our results using a completely independent mouse model of impaired cholinergic signalling.

Lineage tracing using the TH-cre; Ai14D mice was used to confirm the VACHT labelling was at least in some cases (those presented in figure 2c-f) of sympathetic origin. This was useful. However, in supplementary figure 4 the authors also show evidence that there are many VACHT+ profiles that are not coincident with TH+ sympathetic neuronal labeling in these same TH reporter mice.

Most likely VACHT staining does not coincide in all cases with tomato staining in bone-associated fibres due to limited recombination efficiency since the treatment of postnatal mice with 6-OHDA established that most (if not all) VACHT+ cholinergic nerve fibers in femoral BM are sympathetic cholinergic.

Chemical sympathectomy with 6OHDA reduced but did not abolish VACht labelling. These findings reinforce the idea that there is other cholinergic innervation, not of sympathetic origin, that could be contributing to the effects seen on HSC function.

As commented above, it is possible that a minor population of non-neuronal VACht⁺ structures remain after chemical sympathectomy and this might explain why VACht labelling is not completely abolished. However, this does not diminish the relevance of our findings since our data reveal that 6OHDA treatment dramatically reduces VACht⁺ staining, suggesting that the vast majority of this staining labels sympathetic cholinergic neural fibres.

Additionally, we have stained skull samples from 6OHDA-treated mice or control mice for Gfr α 2, as an alternative marker for cholinergic fibres. Similarly to VACht labelling, we detect 4-fold-decreased Gfr α 2 staining in mice treated with 6OHDA early postnatally (**Figure R14** added to the revised manuscript as new Supplementary Fig. 3d-f).

Figure R14. Early postnatal chemical sympathectomy targets Gfr α 2⁺ cholinergic fibers. Left: Immunofluorescence of Gfr α 2 (red) and CD31 (blue) in skull BM from mice treated early postnatally with 6OHDA or saline. Scale bar, 100 μ m. Right: Quantification of Gfr α 2⁺ area from these images. Data are means \pm SEM. Unpaired two-tailed t test. n = 5. ** p < 0.01.

The electron microscopy to show apparent reduced numbers of unmyelinated fibers is presented with only a single sample and without quantification, so it is difficult to gauge if the result occurs from sampling bias.

Additional examples are provided above in response to Minor Comment #5 by Reviewer 1 (**Figure R10**). Unmyelinated axons were identified and quantitated by two independent neuroscientists with experience in TEM. The two observers identified 27 unmyelinated axons in WT endosteum and only 4-5 in the same region of Gfr α 2^{-/-} mice. The conclusions were pragmatically phrased “Whereas myelinated axons appeared unchanged in the same region, unmyelinated (compatible with cholinergic) axons appeared reduced in Gfr α 2^{-/-} mice.”

Whilst sympathetic cholinergic innervation exists and is documented clearly in the limbs of mice, there is still some debate about whether it exists in other species, particularly the rat. The authors again use GFR α 2 as the marker for cholinergic innervation in the rat exercise stress experiments, but there is no evidence that this is appropriate for this species. This needs to be confirmed, along with providing clear evidence of cholinergic innervation in the femur of rats, if one is to accept the interpretation of the authors relating to the role of sympathetic cholinergic neurons on stress induced effects to HSC function. In addition, the sympathectomy in the rats was performed in the early postnatal period,

before the proposed cholinergic switch documented in mice, and so the changes observed could be the result of both adrenergic and cholinergic sympathetic innervation.

The Gfra2 staining in rat samples seems very clear to us, and the cholinergic switch in periosteum was confirmed in rats (Asmus *et al.*, 2000). However, we agree that rat data adds little or would require extra validation so we have removed it in the revised manuscript because it does not substantially add to the main messages and might be distracting.

Overall, many of the findings are consistent with the hypothesis that sympathetic cholinergic neurons might regulate HSC proliferation and self renewal, but given the limitations of the findings described above, it is difficult to see how this could be asserted with the strictest of confidence. The effects ACh through nicotinic receptors on BM stromal cells could be the result of ACh released from cells other than of sympathetic cholinergic origin.

We hope that the new data provided will convince the Reviewer that most VAcH⁺ structures in the BM are sympathetic cholinergic fibres and that these fibres are drastically reduced in *Gfra2*^{-/-} mice.

Reviewer #3 (Remarks to the Author):

This manuscript is the first to demonstrate a switch in bone marrow sympathetic innervation from noradrenergic to cholinergic within first week after birth in mice with functional consequences for HSC quiescence and self-renewal. The manuscript introduces novel concepts regarding maintenance of haematopoietic stem cell (HSC) quiescence. Experimental design is elegant with extensive use of lineage tracing and gene-deleted animals to characterise nerve fibres innervating the bone marrow complemented by functional transplant assays.

Thank you very much for the positive comments.

General Comment

1. The cholinergic switch and innervation studies described in this manuscript are innovative, interesting and well performed. However a role for bone marrow cholinergic nerve fibres in regulation of haematopoiesis is limited to direct role on Nestin⁺ MSC and CXCL12 as sole mediator. Why are possible roles for cholinergic signalling in regulation of other niche cell populations (eg. endothelial cells, tissue-resident macrophages and Nestin negative MSC) largely ignored? Most of these niche cells also express cholinergic receptors and reported to regulate HSC via CXCL12 as well as other factors.

We have measured the mRNA expression of cholinergic receptors in BM endothelial cells and we found that these cells express both nicotinic and muscarinic receptors. Interestingly and contrasting nestin⁺ BMSCs, we did not detect *Chrna7* mRNA expression in BM endothelial cells (**Figure R15**), suggesting to some degree target cell-type specificity in this regulation. Although our study clearly involves the regulation of nestin⁺ niche as one mechanism by which the cholinergic signals regulate quiescence in endosteal HSCs, we have not excluded the possible contribution of other cells. We have discussed this possibility in the revised manuscript (P10. L238-244). In fact we hope that this will be the subject of future studies by our lab and other labs, once these novel fibres and some of their targets and functions have been described in the current manuscript.

Figure R15. Cholinergic receptors (nicotinic and muscarinic) mRNA expression in sorted CD31⁺ endothelial cells. Data are means \pm SEM.

Specific comments on Figures and associated text are listed stepwise below.

Fig 1. (g-j) Comment regarding 'Endomucin .. marks endothelial cells'. Endomucin is commonly utilised to distinguish venous endothelium (endomucin positive) from arteriole vasculature (largely negative -see M Corada et al., Nature Comms 2013, PMID 24153254 and Fig2 in review article Arterial versus venous endothelial cells, PMID: PMC4105978). This is relevant as bone marrow HSC residing near

arterioles (which may not stain with endomucin) are reported to have differing self-renewal potential compare to HSC near the (endomucin positive) post-capillary venules. Could this impact interpretation of the staining data in Figure 1 g-h as endomucin may only be staining a portion of the vasculature (not arterioles) at the bone marrow endosteum?

We fully agree with the Reviewer that endomucin is used to distinguish venous endothelium from arteriole vasculature; however, it is also used to label transition zone vessels, which connect arterioles and sinusoids in the endosteal BM (where we find sympathetic cholinergic innervation) (Itkin *et al.* 2016; Kusumbe *et al.* 2014).

SuppFig2 trabecular bone marrow stains (a-d) with associated controls (h-j). These data suggest that genetically-traced Cholinergic neurons co-stain with CD31+ endothelial cells at the endosteum. This is not mentioned in manuscript text and suggest the manuscript's final discussion - regarding tentative direct role of Nestin-GFP+ MSC in regulating HSC quiescence via direct cholinergic sympathetic signalling - should be more nuanced as several other cells of the HSC niche, including endothelium and macrophages (see example refs below) may also express cholinergic receptors.

Cooke JP *et al.*, 2008. Trends in Cardiovascular Medicine Endothelial Nicotinic Acetylcholine Receptors and Angiogenesis (PMC 2673464) Chernyavsky AI, Int Immunopharmacol. 2010. Auto/paracrine control of inflammatory cytokines by acetylcholine in macrophage-like U937 cells through nicotinic receptors.

We agree that cholinergic nerve fibres are closely associated with endothelial cells in endosteal niches and it is possible that some of them also express cholinergic markers, as mentioned by the Reviewer. However, we would like to highlight that most VAcHT⁺ cholinergic fibres do not overlap with endothelial cells (please see Fig. 1d-g and Supplementary Fig. 1a-d). As clarified above, we do not exclude the possibility that other cells, such as endothelial cells or macrophages, might be affected by cholinergic signalling, and we have clearly mentioned this possibility (which requires further investigation by our group and others in future studies) in the revised discussion (P9-10. L239-243).

Fig 3 and associated text.

Line 111 comment 'HSC...enter quiescence in a Cxcl12-dependent manner ..in postnatal weeks 3 to 4' ... Reference 23 has been misquoted here. Although correct that reference 23 (PMC1578623) does indeed state that HSC enter quiescence between postnatal weeks 3 to 4 (see Ref 23, Fig3), at no stage does it state that this is in a 'CXCL12-dependant manner'. Instead reference 23 claims that post-natal HSC had an engraftment defect while cycling (which could be rectified by administering CXCL12 blockade to host) and that FACs sorted 3wk postnatal mouse HSC separated based on cell-cycle (using phenotype lin-Sca-1+, CD43+, Mac1+) show different levels of CXCL12 mRNA expression in themselves. This is quite distinct from 'HSC enter quiescence in CXCL12 dependant manner.' Please remove sentence.

The Cxcl12 dependency in this sentence is supported by additional studies. Christensen *et al.*, 2004 reported that HSC migration to foetal BM is enhanced by Cxcl12 and stem cell factor, both of which are highly expressed and progressively upregulated in BM nestin⁺ cells at perinatal stages (Isern *et al.*, 2014). Furthermore, Ara *et al.*, 2003 revealed that Cxcl12 is produced by different BM stromal cells and is required for developmental BM colonization by HSCs.

Similarly the comment (also line 111) 'HSC abruptly enter quiescence... between postnatal weeks 3 to 4.....concurrently with the completion of the cholinergic switch' ... has also not been yet proven. Figs 1 to 3 in current manuscript were performed in 1 week old mice only. Later figures refer to adult

mice or rats. No time-course to demonstrate that cholinergic switch occurs ‘concurrently’ with abrupt HSC quiescence has been performed.

The time-course of the cholinergic switch has been previously described in several publications and takes place during the first 3-4 postnatal weeks (Landis & Keefe, 1983; Leblanc & Landis, 1986; Landis *et al.*, 1988; Asmus *et al.*, 2000). Therefore, it is probably not necessary to further demonstrate the concurrency between the cholinergic switch and HSC quiescence in the first postnatal weeks.

Many other biological processes also happen to mice between post-natal week 3 to 4 including weaning and exposure to new food and associated bacteria, a shift in microbiota and immune settings etc.

We agree that the cholinergic switch is not the only process that happens during the first postnatal weeks (and have acknowledged this in the revised discussion; P9. L. 214-215). However, the cholinergic switch takes place at the time when HSCs migrate in the BM and become quiescent, but not in *Gfra2*^{-/-} mice. Given that the mechanisms triggering this change in HSCs remain elusive and that we present this as one possible explanation, we respectfully consider our point fair and interesting for the reader.

As the data contained in Fig 1 to 4 are all correlative only, perhaps the wording in places could be more nuanced to reflect this such as the statement in line 108 should be clearly explained as hypothesis.

We have changed this in the revised manuscript to “These results may suggest the possibility that sympathetic cholinergic fibres contribute to developmental BM HSPC colonization by inducing Cxcl12 expression in postnatal BM”.

Fig 4d. (RE *Gfra2* chimeras, lines 134 to 139). Normally to confirm an effect is HSC extrinsic, wildtype HSC would be also injected into *Gfra2*^{-/-} recipients . Without data from such a reverse chimera it is not possible to ... “clearly dissect haematopoietic cell autonomous and non-autonomous effects” .. as stated in line 134. The wording ‘clearly dissect’ should be modified accordingly. The reasons why the reverse chimera was not shown could include irradiation-induced nerve damage in recipient mice confusing the result. However if so this raises further questions of whether BM cholinergic nerves do survive the conditioning regimes used in these transplants and how data from Fig4d and should then be interpreted.

We have addressed this question in response to point 10 from Reviewer 1. To dissect between cell-autonomous (intrinsic) and non-cell autonomous (extrinsic) contributions we did perform reciprocal BM transplantations using *Gfra2*^{+/-} and *Gfra2*^{-/-} mice as donor and recipients in all possible combinations. In agreement with altered *Gfra2*^{-/-} BM microenvironment, only *Gfra2*^{-/-} recipient mice (regardless of the genotype of donor cells) exhibit a significant decrease in Cxcl12 expression (please see **Figure R4** and new Fig. 4d). Although we did not perform cell cycle analysis in these mice, we have already shown that reduced Cxcl12 level correlates with decreased HSC quiescence. Therefore, our data strongly suggest that decreased HSC quiescence in *Gfra2*^{-/-} mice is caused by impaired cholinergic signalling in the BM microenvironment.

Fig 5(a-h) statement on line 140 ‘ We found that endosteal Nes-GFP+ cells are innervated with cholinergic fibres’. This is possibly true from the data shown, however in supplementary Fig2(a-e) there is far more convincing co-localisation with CD31+ endothelial cells which also express nicotinic receptors as do macrophages (see example refs provided in SuppFig2 above). This is not at all mentioned in text and raises questions on why wasn’t a more open approach taken to investigate the

contributions of a range of known HSC niche cells including; (1) other MSC populations especially non-Nestin+ CXCL12 abundant PDGFRa+ MSC (as described in Greenbaum et al., Nature 2013), (2) bone marrow endothelium and (3) tissue resident perivascular macrophages.

We do not exclude the possibility that other cells, such as endothelial cells or macrophages, might be affected by cholinergic signalling, and have clearly mentioned this possibility (which requires further investigation in future studies) in the revised discussion (P9-10. L239-243). We hope that these independent investigations will be the subject of future studies by our lab and other labs, once these novel fibres and some of their targets and functions have been described here.

Fig5(i). Control Nes-GFP neg cells used for CXCL12 Q-PCR were of phenotype CD45-CD31-Ter119-. A far more appropriate and relevant Nes-GFP neg control cell population could have been achieved by Sca-1+ PDGFRa+ co-staining. These important non-Nestin MSC also express abundant CXCL12 and depletion of CXCL12 within this population which are within the Nestin neg Prx1-Cre x fl CXCL12 MSC population results in a complete loss of dormant BM phenotypic HSC and severely reduced reconstitution potential in primary transplant (see Ref Greenbaum A D et al., Nature 201). This relevant study should be included in references.

There are multiple markers which could be potentially used to label BMSCs (including Sca-1 and PDGFRa). However, the recent studies by the groups of Dr. Sean Morrison and Dr. Paul Frenette clearly demonstrate that all these markers label partially overlapping BMSC populations. As the reviewer highlights, several studies have attempted to find the most relevant source/s of Cxcl12 for HSCs using conditional KO mice and different genetic drivers. We are aware that the study by Dr Link's group mentioned by the reviewer proposed that Nes- Lepr- mesenchymal progenitors targeted by the Prx1-cre driver are the only relevant source of Cxcl12 for adult HSC maintenance. However, nestin expression was not found only in a very small subpopulation of Prx1-cre-targeted cells (the only subpopulation that was isolated and profiled in detail). Moreover, as the Reviewer probably knows Prx1 is expressed in different derivatives from somatic lateral plate mesoderm, and lineage tracing with Prx1-cre drivers marks a wide variety of mesenchymal lineages (osteoblasts/osteocytes, chondrocytes, perivascular stromal cells including Lepr+ cells, CAR cells, and periosteal cells). We discussed this concept in a review (Mendez-Ferrer S et al. Annals NY Acad Sci 2015). Therefore, Prx1-cre cannot be considered BMSC-specific based on the broad recombination pattern (Kawanami et al., 2009). A similar problem (albeit to lower extent, because it spares the prenatal period) affects studies using constitutive Lepr-cre mice.

Additionally, we have analysed BMSC populations by FACS (Chan CKF *et al.* Cell 2015;160:285-298). We have not observed any significant difference between *Gfra2*^{-/-} and control *Gfra2*^{+/-} mice, again suggesting the absence of other bystander effects in the microenvironment (**Figure R13**). For the current study, we focused on *Nestin-GFP*⁺ cells as one clear target and have not excluded other possible target populations, which may be the subject of study in the future.

Reviewers' comments:

Reviewer #1 (Remarks to the Author):

General Comments

The authors have addressed some of my concerns. The authors argue that limiting dilution assays are not necessary to access HSC function. Although I do not really agree with this conclusion I can accept them not doing limiting dilution for the experiment already shown. However, they should provide better in vivo data to support their conclusions as noted below.

Specific Comments

- In Figure 7B&C, similar engraftment levels were shown between endosteal and non-endosteal fractions in the control mice. However, in Figure 4B, the authors showed much higher frequency of quiescent HSPCs in the WT endosteal fraction. Thus, this in vivo data does not correlate with their in vitro data. If the in vitro data was correct, there should be much higher engraftment levels using WT endosteal marrow compared with using non endosteal marrow because quiescence HSCs harbor the highest self-renewal and long-term reconstitution capacity. The authors still have not addressed this point adequately as it is a key point of the paper. Did the authors only perform this transplantation once? Are these results even reproducible?
- The authors claimed that transplants for the 6OHDA quiescence experiments were not necessary because "quiescence is usually studied through cell cycle experiments, rather than through in vivo transplantation". However, the results from the endosteal and non endosteal transplants mentioned in the above comment would suggest that the authors' conclusions are incorrect. HSC function in the literature is strongly associated with the percent of HSC that are in quiescence. However, the authors with their own research presented in this paper demonstrated that these results could be artifacts without addition of extensive corresponding transplantation results. Since their manuscript is mainly studying HSC quiescence and self-renewal, they need to provide more convincing data to support their conclusion. The authors MUST provide primary limiting dilution analysis and secondary transplants for all experiments where they claim that their knockout and/or treatment results in an increase in self-renewal or quiescence.
- I still feel that it is not acceptable to transplant CD45.2+ donor cells into CD45.2+ recipients. CD45.1+ mice are commercially available and not being able to get these mice is not an excuse. This in vivo assay is important for their conclusions and should be proven by having different genotyped donor vs. recipient cells. Trying to argue your way out of this is really not appropriate.

--

Reviewer #2 (Remarks to the Author):

The response has adequately addressed some of the comments I made, but I am still left without confidence in some of the conclusions, because it is difficult to ignore effects of other cell types as an alternative hypothesis to actions through sympathetic cholinergic neurons. My overall thought remains similar to after the first review, that many of the findings are consistent with the hypothesis that sympathetic cholinergic neurons might regulate HSC proliferation and self renewal, but it is difficult to see how this could be asserted with the strictest of confidence.

--

Reviewer #3 (Remarks to the Author):

Readability, layout and improved Figure content of resubmitted manuscript is much improved.
Comments.

1. Fig 4. Could authors please confirm that cholinergic nerve fibres are indeed only in the endosteal prep using this flushing method. Much data (Fig 4b, Fig4c, Fig 4e) is based on this assumption, however a clean separation of cholinergic nerve fibres between endosteal and central bone marrow preparations using this technique has not been confirmed. This could be easily confirmed by repeating the flushing technique (shown in Fig4a) on bones from reporter ChAT-IRES-cre: Ai35D mice (anticipate label only in endosteal prep) compared to Th-cre: Ai14D mice (potentially labels both endosteal and central bone marrow preparations). This would validate the approach used to generate the data in Fig 4.

2. Results section (pg6 line 135) Comment re Fig 4d... 'suggesting that HSPC quiescence is extrinsically regulated by cholinergic signals through the microenvironment'. Have the authors considered that these results could be also due to nerve damage following the conditioning irradiation needed for HSC transplant? Have the authors checked the degree to which the BM nerve fibres are altered by the transplant conditioning irradiation they use when generating transplant chimera data? Again this could be easily checked by comparing bone marrow innervation in irradiated ChAT-IRES-cre: Ai35D and Th-cre: Ai14D recipient mice. The concern is that transplant conditioning may be damaging the very nerves being studied.

3. Are there any changes in BM adrenergic nerves in Gfra2^{-/-} mice? That is, is there any compensation effects when cholinergic nerves are absent that may indirectly affect HSC quiescence shown in Fig 4. Has this been checked? This is relevant as (Gfra2^{+/-} wildtype like HSC) appear to home and engraft normally in Gfra2^{-/-} hosts. This alone suggests that absence of cholinergic fibres in steady-state does not greatly affect HSC reconstitution potential unless stress is applied (such as shown in Fig 7). A complementary explanation is that adrenergic signalling is indirectly altered as well.

4. Fig R5. Interesting and convincing data. Would the authors find similar result with same experimental design in Chrna7^{-/-} mice? This may help confirm whether Chrna7 is absolutely essential or partially redundant in function with other chrna or chrm.

5. Statistics are appropriate however the use of SEM throughout is not. In virtually all figures Standard Deviations should be shown not SEM. SEM may look 'better' as they are smaller but are not appropriate.

Reviewers' comments:

Reviewer #1 (Remarks to the Author):

General Comments

The authors have addressed some of my concerns. The authors argue that limiting dilution assays are not necessary to assess HSC function. Although I do not really agree with this conclusion I can accept them not doing limiting dilution for the experiment already shown. However, they should provide better in vivo data to support their conclusions as noted below.

We thank the reviewer for the constructive criticism. We agree and have followed the advice and performed the following new in vivo experiments over the past 1.5 year:

- 1) Myelosuppression through 5-FU or irradiation increases the expression of cholinergic markers related to *Gfra2*-*Nrtn* signalling (Fig. 3b,c) or the presence of cholinergic innervation (Fig. 3d,e and Supplementary Fig. 3).
- 2) Serial transplantation of HSCs purified from *Gfra2* KO/control mice into *Gfra2* KO/control recipients (new Fig. 4 and Fig. 5). The data demonstrates that the lack of cholinergic regulation of the BM microenvironment renders HSCs hyperproliferative and more prone to exhaust, as an indication of decreased self-renewal.
- 3) Chronic treatment with nicotine (as a gain-of-function model to complement the *Gfra2*^{-/-} and the *Chrna7*^{-/-} loss-of-function models) induces endosteal HSC quiescence under steady state (Supplementary Fig. 5c-e) and after transplantation (Fig. 6e and Supplementary Fig. 6) leading to reduced haematopoietic recovery after transplantation (Fig. 6b-c).
- 4) To investigate the possible association of cholinergic nicotinic signalling with human HSC proliferation after transplantation, we retrospectively analysed the kinetics of platelet recovery in 201 patients undergoing allogeneic HSC transplantation, taking into consideration their smoking history before transplantation. Interestingly, a history of smoking correlated with a significant delay in the normalisation of circulating platelets after transplantation (Fig. 6f). Multivariate analysis indicated that normalisation of platelet counts was significantly delayed in smokers ($p=0.001$; Multivariate Cox Analysis), independently of the type of donor (matched family vs. alternative donor) and the number of CD34⁺ cells infused, which also influenced platelet reconstitution kinetics, as expected. Although other factors could also contribute to delayed haematopoietic recovery in smokers, these results suggest that cholinergic nicotinic signalling might impact clinical haematopoietic recovery after transplantation.

Specific Comments

- In Figure 7B&C, similar engraftment levels were shown between endosteal and non-endosteal fractions in the control mice. However, in Figure 4B, the authors showed much higher frequency of quiescent HSPCs in the WT endosteal fraction. Thus, this in vivo data does not correlate with their in vitro data. If the in vitro data was correct, there should be much higher engraftment levels using WT endosteal marrow compared with using non endosteal marrow because quiescent HSCs harbor the highest self-renewal and long-term reconstitution capacity. The authors still have not addressed this

point adequately as it is a key point of the paper. Did the authors only perform this transplantation once? Are these results even reproducible?

We thank the reviewer for bringing up this important point. However, whilst generally quiescent HSCs harbour the highest self-renewal and long-term reconstitution capacity, as the reviewer knows HSCs are very heterogeneous (Uchida N et al *Exp Hematol.* 1996;24:649; Mazurier F et al;2004;103:545; Sieburg HB et al *Blood* 2006;107:2311; Lu R et al. *Nat Biotechnol* 2011;29:928; Gibbs KD et al 2011;117:4226; Morita Y et al *J Exp Med* 2010;207:1173; Benz C et al *Cell Stem Cell* 2012;10:273; Wilson NK et al *Cell Stem Cell* 2015;16:712; Eaves CJ *Blood* 2015;125:2605; Sanjuan-Pla A et al *Nature* 2013;502:232; Carrelha J et al *Nature* 2018;554:106; Ganuza M et al *Blood* 2019).

In that regard, it has been recognised that highly quiescent (latent or dormant) HSCs only show robust multilineage reconstitution potential upon secondary transplantation (Yamamoto R et al. *Cell Stem Cell* 22:600; 2018). It is also likely that different HSC populations contribute to steady-state vs. stress haematopoiesis. It is in the latter setting that the new cholinergic regulation described here appears to be important to maintain HSC quiescence. To emphasize this notion, we have added the word “stress” to the title.

However, we took the reviewer’s point quite seriously and performed the additional long-term transplantations and experiments detailed below over the past year and a half:

- 1) Myelosuppression through 5-FU or irradiation increases the expression of cholinergic markers related to *Gfra2*-*Nrtn* signalling (Fig. 3b,c) or the presence of cholinergic innervation (Fig. 3d,e and Supplementary Fig. 3).
- 2) Serial transplantation of HSCs purified from *Gfra2* KO/control mice into *Gfra2* KO/control recipients (new Fig. 4 and Fig. 5). The data demonstrates that the lack of cholinergic regulation of the BM microenvironment renders HSCs hyperproliferative and more prone to exhaust, as an indication of decreased self-renewal.
- 3) Chronic treatment with nicotine (as a gain-of-function model to complement the *Gfra2*^{-/-} and the *Chrna7*^{-/-} loss-of-function models) induces endosteal HSC quiescence under steady state (Supplementary Fig. 5c-e) and after transplantation (Fig. 6e and Supplementary Fig. 6) leading to reduced haematopoietic recovery after transplantation (Fig. 6b-c). The effect is more pronounced under the stress induced by irradiation, fully consistent with the loss-of-function experiments and the contentions in our study.
- 4) To investigate the possible association of cholinergic nicotinic signalling with human HSC proliferation after transplantation, we retrospectively analysed the kinetics of platelet recovery in 201 patients undergoing allogeneic HSC transplantation, taking into consideration their smoking history before transplantation. Interestingly, a history of smoking correlated with a significant delay in the normalisation of circulating platelets after transplantation (Fig. 6f). Multivariate analysis indicated that normalisation of platelet counts was significantly delayed in smokers (p=0.001; Multivariate Cox Analysis), independently of the type of donor (matched family vs. alternative donor) and the number of CD34⁺ cells infused, which also influenced platelet reconstitution kinetics, as expected. Although other factors could also contribute to delayed haematopoietic recovery in smokers, these results suggest that cholinergic nicotinic signalling might impact clinical haematopoietic recovery after transplantation.

We would like to emphasize that all the previous and new experimental paradigms (response to 5-FU, haematopoietic reconstitution after transplantation of BM cells or purified HSCs, RNAseq from endosteal/non-endosteal HSCs, experiments with nicotine, clinical correlations) all strongly convey the same message: that this new cholinergic regulation is important to maintain HSC quiescence under stress. We hope that the new evidence provided can convince the reviewer.

- The authors claimed that transplants for the 6OHDA quiescence experiments were not necessary because “quiescence is usually studied through cell cycle experiments, rather than through in vivo transplantation”. However, the results from the endosteal and non endosteal transplants mentioned in the above comment would suggest that the authors’ conclusions are incorrect. HSC function in the literature is strongly associated with the percent of HSC that are in quiescence. However, the authors with their own research presented in this paper demonstrated that these results could be artifacts without addition of extensive corresponding transplantation results. Since their manuscript is mainly studying HSC quiescence and self-renewal, they need to provide more convincing data to support their conclusion. The authors **MUST** provide primary limiting dilution analysis and secondary transplants for all experiments where they claim that their knockout and/or treatment results in an increase in self-renewal or quiescence.

We agree and have provided now additional cell cycle experiments of endosteal and central BM HSCs of chimeric mice generated after long-term transplantation of Gfra2 KO/WT cells into Gfra2 KO/recipient mice. The results demonstrate that the lack of cholinergic regulation in the endosteal BM compromises HSC quiescence (new Fig. 4). We have also performed new serial transplantations of purified HSCs that demonstrated decreased HSC self-renewal (Fig. 5 and Fig. 6), as explained above.

- I still feel that it is not acceptable to transplant CD45.2+ donor cells into CD45.2+ recipients. CD45.1+ mice are commercially available and not being able to get these mice is not an excuse. This in vivo assay is important for their conclusions and should be proven by having different genotyped donor vs. recipient cells. Trying to argue your way out of this is really not appropriate.

We agree and for this reason we have deferred resubmission until completing a new set of serial transplantations using purified HSCs.

--

Reviewer #2 (Remarks to the Author):

The response has adequately addressed some of the comments I made, but I am still left without confidence in some of the conclusions, because it is difficult to ignore effects of other cell types as an alternative hypothesis to actions through sympathetic cholinergic neurons. My overall thought remains similar to after the first review, that many of the findings are consistent with the hypothesis that sympathetic cholinergic neurons might regulate HSC proliferation and self renewal, but it is difficult to see how this could be asserted with the strictest of confidence.

We thank the reviewer for considering that we have adequately addressed the comments in the previous round of revision. Although our study clearly involves the regulation of nestin⁺ niche cells as one mechanism by which local cholinergic neural signals regulate quiescence in endosteal HSCs, we have presented the revised manuscript in a more open and balanced way. Particularly, we have emphasised this broadly new concept of cholinergic regulation of stem cell quiescence under stress and we have avoided overstating the specific contribution of sympathetic cholinergic neurons and possible target cells. In the revised manuscript we have concentrated focus on the regulation of haematopoietic recovery after transplantation and we have strengthened the conclusions with new

long-term in vivo experiments and correlations with clinical HSC transplantation. We believe that the description of the cholinergic innervation, its origin, possible target cells and other functions outside the regulation of HSC proliferation do not fit in the current manuscript and should be the subject of future studies by our lab and other labs, hopefully prompted by this publication.

Reviewer #3 (Remarks to the Author):

Readability, layout and improved Figure content of resubmitted manuscript is much improved.

Thank you very much for the positive comment.

Comments.

1. Fig 4. Could authors please confirm that cholinergic nerve fibres are indeed only in the endosteal prep using this flushing method. Much data (Fig 4b, Fig4c, Fig 4e) is based on this assumption, however a clean separation of cholinergic nerve fibres between endosteal and central bone marrow preparations using this technique has not been confirmed. This could be easily confirmed by repeating the flushing technique (shown in Fig4a) on bones from reporter ChAT-IRES-cre: Ai35D mice (anticipate label only in endosteal prep) compared to Th-cre: Ai14D mice (potentially labels both endosteal and central bone marrow preparations). This would validate the approach used to generate the data in Fig 4.

The approach illustrated in Supplementary Fig. 1a for clarity has been used in multiple previous studies to separate endosteal cells from non-endosteal cells (Haylock DN et al Stem Cells 2007;25:1062; Grassinger J et al Methods Mol Biol 2011;750:197; Balduino A et al Exp Cell Res 2012;318:2427; Guarneiro J et al Stem Cell Rep 2014;2:794; Baustian C et al Stem Cell Res Ther 2015;6:151; Chen C et al Med Sci Monit 2015;21:2757; Duarte D et al Cell Stem Cell 2018;22:64; etc...). The new in vivo experiments further demonstrate that cholinergic nicotinic signalling regulates HSC proliferation in the endosteal (but not the central) BM niches (new Fig. 4b,c; new Fig. 6c,d; new Supplementary Fig. 5c,d).

2. Results section (pg6 line 135) Comment re Fig 4d...' suggesting that HSPC quiescence is extrinsically regulated by cholinergic signals through the microenvironment". Have the authors considered that these results could be also due to nerve damage following the conditioning irradiation needed for HSC transplant? Have the authors checked the degree to which the BM nerve fibres are altered by the transplant conditioning irradiation they use when generating transplant chimera data? Again this could be easily checked by comparing bone marrow innervation in irradiated ChAT-IRES-cre: Ai35D and Th-cre: Ai14D recipient mice. The concern is that transplant conditioning may be damaging the very nerves being studied.

Thanks for the very insightful comment. We have studied this axis in the two haematopoietic stress settings (myeloablation with 5-FU or after irradiation) and have observed that this regulatory circuit is not only maintained, but increased upon haematopoietic stress, fully supporting our previous contention. Particularly:

- 1) Myeloablation with 5-FU increases after 1 week the mRNA expression of Gfra2 and its ligand Neurturin in the BM (new Fig. 3b,c).
- 2) Irradiation increases the presence of Gfra2+ cholinergic nerve fibres in the BM after 2 weeks and actually reduces the presence of Th+ noradrenergic nerve fibres in the BM after 4 weeks (new Fig. 3d,e). To thoroughly identify these fibres we used a genetic model (Wnt1-Cre2;TdTomato mice. To evaluate the impact of irradiation and transplantation on BM nerve fibres expressing cholinergic markers, we used Ai14 reporter mice intercrossed with Wnt1-

cre2 mice, which express tdTomato fluorescent protein in neural-crest-derived cells, including nerve fibres and their associated Schwann cells¹⁹. *Wnt1-cre2;Ai14* mice were lethally-irradiated and transplanted with 10⁶ BM cells. Immunofluorescence for the cholinergic marker Gfra2 and the noradrenergic marker tyrosine hydroxylase (Th) was performed at baseline, and 2 or 4 weeks after irradiation and transplantation to study BM innervation during the haematopoietic recovery phase. Importantly, 7-fold more cholinergic nerve fibres were detected in the BM acutely (2 weeks) after transplantation, whilst the abundance of noradrenergic nerve fibres dropped 5-fold 4 weeks after transplantation (Fig. 3d,e and Supplementary Fig. 3). This result is consistent with the effects of 5-FU (Fig. 3b,c) and suggests that the cholinergic regulation of HSCs is activated in the BM during haematopoietic stress.

3. Are there any changes in BM adrenergic nerves in *Gfra2*^{-/-} mice? That is, is there any compensation effects when cholinergic nerves are absent that may indirectly affect HSC quiescence shown in Fig 4. Has this been checked? This is relevant as (*Gfra2*^{+/-} wildtype like HSC) appear to home and engraft normally in *Gfra2*^{-/-} hosts. This alone suggests that absence of cholinergic fibres in steady-state does not greatly affect HSC reconstitution potential unless stress is applied (such as shown in Fig 7). A complementary explanation is that adrenergic signalling is indirectly altered as well.

Thanks for the insightful thought. We have addressed this in the revised manuscript (P.4 L.84-92). "We have previously shown that decreased parasympathetic activity in *Gfra2*^{-/-} mice derepresses sympathetic activity and causes abnormal BM egress of HSCs and leukocytes via sympathetic activation of the β 3-adrenergic receptor¹². To investigate the possible contribution of the noradrenergic system to decreased endosteal HSC quiescence in *Gfra2*^{-/-} mice, we intercrossed these mice with *Adrb3*^{-/-} mice. Unlike HSC mobilization¹², increased endosteal HSC proliferation was not the consequence of derepressed noradrenergic activity in *Gfra2*^{-/-} BM because it was not rescued in *Gfra2*^{-/-}*Adrb3*^{-/-} compound mice (Figure 1C). This result suggested that cholinergic signals inhibit HSC proliferation specifically in the endosteal BM niche". Moreover, BM cholinergic fibres increase but BM noradrenergic fibres are unchanged soon (2 weeks) after transplantation (Fig. 3d,e). Altogether, we believe that this is compelling evidence for the direct role of this cholinergic signalling.

The reviewer is correct that this cholinergic regulation is especially relevant under haematopoietic stress. This is further emphasised by the increased cholinergic markers after 5-FU or irradiation and by the effects of cholinergic deficiency on haematopoietic recovery after transplantation or nicotine exposure (new Figures 4, 5 and 6 and new Supplementary Figures 3, 5 and 6). For clarity, we have highlighted this notion in the new title provided as well.

4. Fig R5. Interesting and convincing data. Would the authors find similar result with same experimental design in *Chrna7*^{-/-} mice? This may help confirm whether *Chrna7* is absolutely essential or partially redundant in function with other *chrna* or *chrn*.

Unfortunately we were not able to repeat this experiments with *Chrna7*^{-/-} mice but we provided additional experiments with nicotine treatment demonstrating that chronic treatment with nicotine (as a gain-of-function model to complement the loss-of-function models) induces endosteal HSC quiescence under steady state (Supplementary Fig. 5c-e) and after transplantation (Fig. 6c,d) and is associated with reduced haematopoietic recovery after transplantation (Fig. 6b). These results add to the previous results showing a similar deregulation of *Cxcl12* and HSC quiescence in the endosteal BM of *Chrna7*^{-/-} mice and *Gfra2*^{-/-} mice, the effect of nicotine inducing *Cxcl12* and the lack of effect

of muscarinic antagonist, which perfectly match the higher expression of nicotinic receptors (compared with muscarinic receptors) (Figures 1, 2 and Supplementary Fig. 2).

To investigate the possible association of cholinergic nicotinic signalling with human HSC proliferation after transplantation, we retrospectively analysed the kinetics of platelet recovery in 201 patients undergoing allogeneic HSC transplantation, taking into consideration their smoking history before transplantation. Interestingly, a history of smoking correlated with a significant delay in the normalisation of circulating platelets after transplantation (Fig. 6f). Multivariate analysis indicated that normalisation of platelet counts was significantly delayed in smokers ($p=0.001$; Multivariate Cox Analysis), independently of the type of donor (matched family vs. alternative donor) and the number of CD34⁺ cells infused, which also influenced platelet reconstitution kinetics, as expected. Although other factors could also contribute to delayed haematopoietic recovery in smokers, these results suggest that cholinergic nicotinic signalling might impact clinical haematopoietic recovery after transplantation.

5. Statistics are appropriate however the use of SEM throughout is not. In virtually all figures Standard Deviations should be shown not SEM. SEM may look 'better' as they are smaller but are not appropriate.

We have changed the graphs to indicate standard deviations instead of SEM throughout the manuscript.

Reviewers' Comments:

Reviewer #1:

Remarks to the Author:

Thank you for responding so thoroughly to all my concerns.

Reviewer #2:

Remarks to the Author:

Whilst much of the narrative has changed to focus on cholinergic signaling, some of the data still rely on the premise that GFRA2 is a marker for cholinergic neurons. Whilst this is not incorrect, the authors have not provided any additional data to exclude the possibility that the effects they observe in the GFRA2 knockouts are only through cholinergic neurons, nor did they adequately acknowledge this in the manuscript. This confounds a number of their data sets. The GFRA2 immunostaining presented in Supp Fig 3 does not help the authors claim. It shows labeling of many non-neuronal profiles, which if true suggests that non-neuronal cells might also be involved. In the absence of more careful characterization of the neuronal profiles that are there, it also fails to provide evidence that the neuronal profiles that are there are all cholinergic. In addition, important controls for this immunostaining were not shown, or even described adequately in the results. It seems like the authors should have at least shown lack of labeling in the neuronal profiles, but not the non-neuronal cells, in the bone marrow of GFRA2 knockout mice, that they clearly had access to, to properly support their claim.

It is also not clear how the data presented in Fig 3d and e were quantified. For example, if quantification was done using thresholding of fluorescence, then this would have also included the significant contribution to the labeling from the non-neuronal cells.

Reviewer #3:

Remarks to the Author:

The authors have made a great effort to address the reviewers concerns and have mostly succeeded in this revised manuscript.

The resubmitted manuscript is very much improved with stronger data that has emerged from a substantial number of new additional experiments including time-consuming serial transplantations (shown in Figures 3, 4, 5, 6) as well as with a more focused, nuanced and engaging presentation narrative.

I only have one major concern.

MAJOR CONCERN.

This reviewer very much appreciates the substantial amount of time and resources that have been put into generating the additional data shown in Figure 4 and 5 of this revised manuscript.

However an important control group is missing in both these transplant experiments - meaning that several of the conclusions such as the wording of the titles for Figures 4 and 5 may not be correct and in fact at this point remain unproven.

In Figure 4 and 5 the authors aim was to show that absence of *Gfra2*^{-/-} from the recipient (but not from donor) leads to altered WT HSC cycling and reconstitution post HSC transplant.

The method selected is via HSC transplants where WT donor (CD45.1+) HSC are transplanted into *Gfra2*^{-/-} (CD45.2+) recipients. Then they compared these data with those obtained from CD45.2+

donors into congenic CD45.1+ recipients.

The major problem is that WT CD45.1+ HSC do not have the same reconstitution potential (or possibly even HSC cycling kinetics) as WT CD45.2+ HSC. Please see original ref in <https://ashpublications.org/blood/article/115/2/408/26962/Congenic-interval-of-CD45-Ly-5-congenic-mice>. Between other things, the CD45.1 and CD45.2 intergenic region is known to contain several SNIPs including for important HSC regulatory genes such as Cxcr4 as well as Jarid1b (Lysine (K)-specific demethylase 5B or Kdm5) which are both expressed in HSPCs.

These data (that CD45.2+ HSC have around 2 to 3 fold greater reconstitution potential than CD45.1+ in transplant) has been confirmed many times in several different labs and is even reproducible on different mouse backgrounds (ie. same difference whether on C57BL/6 or balb/c background).

Importantly both of these SNIPs (in Cxcr4 and Jarid1d) may interfere with the interpretation of the data in Figure 4. That is the 'red' and 'green' mice are both CD45.2+ \diamond CD45.1+ recipients while the 'yellow' mice are CD45.1+ \diamond CD45.2+ recipients. This means any apparent differences in the 'yellow' group may not solely be due to absence of Gfra2 from environment, but could be also/instead be due to the differences in the CXCR4 (the receptor for CXCL12) and Jarid1b SNIPs between these two congenic strains. The solution is to perform a matching control to the Gfra2^{-/-} host transplant. That is to include as a WT (CD45.1+) donor \diamond WT (CD45.2+) recipient control transplant.

Again this reviewer appreciates the considerable and proactive effort these authors have put into this resubmission. However at present the new data and new experiments presented in Figure 4 and 5 can not be fully interpreted at this stage until an appropriate WT (CD45.1+ donor) control has been performed at least in a primary transplant setting.

MINOR CONCERNS.

METHODS.

Line 481. 'SEM' The authors have indicated they are now using SD throughout including in Figure legends. Could authors please edit the term 'SEM' in methods line 481 if no longer correct.

RESULTS

1))Number of mice per group.

Many of the experiments involve only n=3 per group. This includes Figure 1, Figure 2 c,e, 3d, e, 6 b,c and perhaps others. There is also no confirmation that experiments have been repeated to ensure reproducibility in the text. From this perspective it would be recommended to show the data (in most figures) as individual dots (one dot per individual mouse) in order to help readers appreciate the inherent variability of the data and the actual (often low) numbers of biological replicates that have been used to reach these conclusions.

Figure 3a. RE. The description of these findings are that 'Gfra2^{-/-} mice show an 'augmented haematopoietic recovery' post 5-FU. (line 115).

This figure (3a) rather suggests that each of the mouse genotypes had a similar starting leukocyte count of 10- 13mill per mL blood prior to chemotherapy and that each recovered back to that level at approximately the same time (around day 12 post 5-FU) thus no 'augmented recovery' by any one group.

What is observed though is that Gfra2^{-/-} mice appear to then 'over-shoot' with blood leukocytes reaching over 20million/mL blood, or levels double that of recovered wildtype control or Chrma7^{-/-} mice. Instead of 'augmented recovery' these data suggest the Gfra2^{-/-} mice have a decreased ability

to fine tune immune responses (such as not to over shoot). Please change text wording to reflect the data.

Line 542. Figure legend 3. ...donor cells comprise 'CD45.1' Please edit this to match what is shown in associated figure outline 3f (ie. CD45.2 donors). At least edit one of these to be consistent.

Figure 3b,c. Could authors use a different colour for mRNA level data.

In this figure red is consistently WT mice (or WT-like reporters), while green is Gfra2^{-/-} .

Thus it is very confusing that Figure 3b instead shows WT mice treated with 5-FU as green (the colour of Gfra2^{-/-} mice). If the colours cant be changed, please put a 'WT' on the X-axis under the ' C and 5-FU ' in order to help readers understand the sudden shift in meaning.

I have a similar comment for the data in Figure 4b,c. In the rest of the whole Figure 4 yellow, green, red indicate mouse genetic background. Oddly Figure 4b,c however uses the same colour scheme to indicate something very different. Please don't do this. Please change 4 b,c to a different colour range (even different shades of grey or blue would be better).

Figure 3h.

Can authors explain why the % circulating Gfra2^{+/-} HET control cells increase suddenly at week 24 post transplant? This is as curious as why % Gfra2^{-/-} reconstitution drops at 24 weeks. These experiments are listed as containing 4 to 10 mice per group. Could it be that a small number of mice at the 24 week timepoint is what is generating this shift in % apparent blood reconstitution? Could authors please show data as a dot per individual mouse to aid interpretation of these data

We thank all 3 reviewers for their constructive criticism, which has allowed to greatly improve our study. We agree that additional studies were required to investigate the neuronal and non-neuronal contributions to cholinergic regulation. Therefore, we have performed extensive studies regarding the source and function of cholinergic signals in the endosteal and central niches. Below is a summary of the new experiments performed over the last year. Notably, the new results fully support our original contentions.

- We investigated the source of acetylcholine during haematopoietic regeneration making use of two validated transgenic lines (*ChAT-Ires-Cre* mice and *ChAT-Gfp* mice) to further demonstrate the contribution of stromal cells to the cholinergic expansion after transplantation (new Fig. 1e-h). These experiments are further supported by increased bone marrow acetylcholine levels (measured by ELISA) 2 weeks after transplantation (new Fig. 1a).
- Controls for GFR α 2 and TH immunofluorescence studies have been included in Supplementary Fig. 1.
- We have generated conditional KO mice for α 7nAChR in haematopoietic cells (iVav-Cre) and HSC niche cells. “ α 7nAChR deletion in either *Nes-Cre*^{ERT2}-targeted HSC niche cells^{19, 22} or in Leptin-receptor-Cre-targeted HSC niche cells²³ (which overlap with *Nes-GFP*⁺ BMSCs^{24, 25}), but not in *Vav1-iCre*-targeted haematopoietic cells, similarly decreased endosteal (not central) BM HSC quiescence (Fig. 3m and Supplementary Fig. 5). These results suggest that ACh limits HSC proliferation by inducing CXCL12 expression in endosteal HSC niche-forming BMSCs via α 7nAChR.” (P.7, L129-134). These targeted deletion experiments further highlight our initial contentions regarding the key importance of niche cells in the cholinergic regulation of HSCs.
- We carried out new alternative competitive long-term transplantations requested as a necessary control by Reviewer 3 (CD45.1>CD45.2). The results support even further our original claims by demonstrating a higher proliferation of WT HSCs in a cholinergic-neural-deficient niche.
- We have interrogated in depth the RNA-Seq data sets of endosteal and central HSCs isolated from WT or *Gfra2*^{-/-} mice. “Notably, gene set enrichment analysis (GSEA) showed that central (compared with endosteal) BM WT HSCs exhibited increased ribosomal, mitochondrial and GFR α 1-related pathways (Fig. 5a, Supplementary Fig. 7a-d and Supplementary Table 1), suggesting a higher activation and different response to GFR signalling. Furthermore, the expression of target genes of the GFR α 1/2 co-receptor RET, which promotes HSC self-renewal and expansion^{15, 16}, was reduced in HSCs from *Gfra2*^{-/-} mice (Fig. 5b). In contrast, myc-related glycolysis and Notch1-dependent pathways, which regulate HSC maintenance^{27, 28} and GFR α 1/2-dependent, but RET-independent, maintenance of cardiac progenitors²⁹, were enriched in endosteal (compared with central) WT HSCs (Supplementary Fig. 7e,f). Similarly, neurite outgrowth-related pathways and interleukin-6 (IL-6)-dependent transcription, which drives cholinergic expansion (Related Manuscript in Supplementary File), were reduced in HSCs from *Gfra2*^{-/-} mice (Supplementary Fig. 7g,h). In

contrast, targets co-activated by Notch1 and Myc³⁰ were enriched in endosteal (compared with central) HSCs from WT, but not in *Gfra2*^{-/-} mice (Fig. 5c and Supplementary Table 2).” (P. 8, L.148-161).

- As a complementary loss-of-function model, transplanted WT mice were treated with a nicotinic receptor antagonist (Fig. 8a). Nicotinic receptor blockade increased BM HSC proliferation after transplantation (Fig. 8b-c). These results suggest that nicotinic signalling reduces HSC proliferation and hematopoietic reconstitution after transplantation.” (P. 9, L.187-191).
- Increased *n* numbers and all figures have now been changed to display the data as individual dots to aid the reader appreciated the inherent variability of the data (unless the data cannot be comprehensively displayed in such way).
- We have changed the colour scheme of graphs to make them more coherent throughout the figures.

Below is a point-by-point response to all of the reviewers’ comments. Changes in the manuscript have been highlighted with blue font. We look forward to the final outcome of the revision process and truly hope that the revised study can be considered acceptable for publication.

Reviewer #1 (Remarks to the Author):

Thank you for responding so thoroughly to all my concerns.

Thank you very much for the positive appreciation.

--

Reviewer #2 (Remarks to the Author):

Whilst much of the narrative has changed to focus on cholinergic signaling, some of the data still rely on the premise that GFRA2 is a marker for cholinergic neurons. Whilst this is not incorrect, the authors have not provided any additional data to exclude the possibility that the effects they observe in the GFRA2 knockouts are only through cholinergic neurons, nor did they adequately acknowledge this in the manuscript. This confounds a number of their data sets. The GFRA2 immunostaining presented in Supp Fig 3 does not help the authors claim. It shows labeling of many non-neuronal profiles, which if true suggests that non-neuronal cells might also be involved. In the absence of more careful characterization of the neuronal profiles that are there, it also fails to provide evidence that the neuronal profiles that are there are all cholinergic. In addition, important controls for this immunostaining were not shown, or even described adequately in the results. It seems like the authors should have at least shown lack of labeling in the neuronal profiles, but not the non-neuronal cells, in

the bone marrow of GFRA2 knockout mice, that they clearly had access to, to properly support their claim.

We agree that additional studies were required to investigate the neuronal and non-neuronal contributions to cholinergic regulation. Over the past year and a half, we have performed extensive studies about the source and function of cholinergic signals in bone and bone marrow. The large new body of exciting results have forced us to split the results into two manuscripts. The scope and focus of the current manuscript remains the function of cholinergic signals in haematopoiesis (and particularly hematopoietic regeneration). The new results are presented in the new Figures 1,4,5,8 and Supplementary Fig 4-7. Controls for immunostaining have been added to Supplementary Fig. 1.

Regarding the source of acetylcholine (ACh) of relevance during hematopoietic regeneration, we have made use of two lines (*ChAT-Ires-Cre* mice and *ChAT-Gfp* mice) and we measured bone marrow ACh after HSC transplantation. The new results have been incorporated in the following paragraph added to the revised manuscript (Page 3, beginning of Results, L.35-53):

“Cholinergic expansion during stress haematopoiesis

The autonomic nervous system has been evolutionarily selected to efficiently respond to stress. Therefore, we have studied the role of sympathetic cholinergic signals during stress haematopoiesis. WT mice underwent BM transplantation (BMT) following lethal irradiation. BM ACh concentration transiently increased 2w after BMT (Fig. 1a), suggesting a role for cholinergic signals during stress haematopoiesis. To investigate the source of BM ACh, we performed genetic lineage tracing of neuroglial cells in *Wnt1-Cre2* mice, and cholinergic cells using *ChAT-Ires-Cre* mice and *ChAT-Gfp* mice. Matching the transient ACh increase, cholinergic nerve fibres peaked 2w after transplantation (Fig. 1b and Supplementary Fig. 1a). Signalling through the GDNF family receptor alpha 2 (GFR α 2) promotes the survival of cholinergic neurons^{12, 13}. In agreement with the transient increase in cholinergic innervation, BM GFR α 2 mRNA expression increased 9-fold 2 weeks after transplantation (Fig. 1c). In sharp contrast, BM noradrenergic fibres steadily decreased over 4w (Fig. 1d and Supplementary Fig. 1b). In a separate study, we found that cholinergic neural signals have bone-anabolic effects and are amplified in the skeletal system through cholinergic osteoprogenitors (Related Manuscript in Supplementary File). Therefore, we measured ChAT⁺ osteoprogenitors and found them similarly expanded 2w after irradiation (Fig. 1e-h and Supplementary Fig. 2). These results suggest that both neural and non-neuronal BM cholinergic signals increase during haematopoietic regeneration.”

[redacted]

It is also not clear how the data presented in Fig 3d and e were quantified. For example, if quantification was done using thresholding of fluorescence, then this would have also included the significant contribution to the labeling from the non-neuronal cells.

Quantification was carried out via manual counting of the fibre staining, utilising the *Wnt1-Cre2;tdTomato* marker as further confirmation of neuronal identity. Therefore, we were able to exclude the labelling from the non-neuronal cells.

--

Reviewer #3 (Remarks to the Author):

The authors have made a great effort to address the reviewers concerns and have mostly succeeded in this revised manuscript.

The resubmitted manuscript is very much improved with stronger data that has emerged from a substantial number of new additional experiments including time-consuming serial transplantations (shown in Figures 3, 4, 5, 6) as well as with a more focused, nuanced and engaging presentation narrative.

I only have one major concern.

MAJOR CONCERN.

This reviewer very much appreciates the substantial amount of time and resources that have been put into generating the additional data shown in Figure 4 and 5 of this revised manuscript. However an important control group is missing in both these transplant experiments - meaning that several of the conclusions such as the wording of the titles for Figures 4 and 5 may not be correct and in fact at this point remain unproven.

In Figure 4 and 5 the authors aim was to show that absence of *Grfra2*^{-/-} from the recipient (but not from donor) leads to altered WT HSC cycling and reconstitution post HSC transplant. The method selected is via HSC transplants where WT donor (CD45.1⁺) HSC are transplanted into *Gfra2*^{-/-} (CD45.2⁺) recipients. Then they compared these data with those obtained from CD45.2⁺ donors into congenic CD45.1⁺ recipients.

The major problem is that WT CD45.1⁺ HSC do not have the same reconstitution potential (or possibly even HSC cycling kinetics) as WT CD45.2⁺ HSC. Please see original ref in <https://ashpublications.org/blood/article/115/2/408/26962/Congenic-interval-of-CD45-Ly-5-congenic-mice>. Between other things, the CD45.1 and CD45.2 intergenic region is known to contain

several SNIPs including for important HSC regulatory genes such as *Cxcr4* as well as *Jarid1b* (Lysine (K)-specific demethylase 5B or *Kdm5*) which are both expressed in HSPCs. These data (that CD45.2⁺ HSC have around 2 to 3 fold greater reconstitution potential than CD45.1⁺ in transplant) has been confirmed many times in several different labs and is even reproducible on different mouse backgrounds (ie. same difference whether on C57BL/6 or balb/c background).

Importantly both of these SNIPs (in *Cxcr4* and *Jarid1d*) may interfere with the interpretation of the data in Figure 4. That is the ‘red’ and ‘green’ mice are both CD45.2⁺ \diamond CD45.1⁺ recipients while the ‘yellow’ mice are CD45.1⁺ \diamond CD45.2⁺ recipients. This means any apparent differences in the ‘yellow’ group may not solely be due to absence of *Gfra2* from environment, but could be also/instead be due to the differences in the CXCR4 (the receptor for CXCL12) and *Jarid1b* SNIPs between these two congenic strains. The solution is to perform a matching control to the *Gfra2*^{-/-} host transplant. That is to include as a WT (CD45.1⁺) donor \diamond WT (CD45.2⁺) recipient control transplant.

Again this reviewer appreciates the considerable and proactive effort these authors have put into this resubmission. However at present the new data and new experiments presented in Figure 4 and 5 can not be fully interpreted at this stage until an appropriate WT (CD45.1⁺ donor) control has been performed at least in a primary transplant setting.

Thanks for the comment. We have carried out the additional WT primary purified HSC transplant (CD45.1 > CD45.2), which demonstrates a reduced engraftment potential compared to the other WT group (CD45.2 > CD45.1), further supporting our original claims by emphasizing the increased chimerism observed in *GFR α 2* knockout recipients. Additionally, we would like to highlight that “Multiple independent experimental paradigms (response to 5-FU, haematopoietic reconstitution under standard conditions, RNAseq from endosteal/non-endosteal, gain of function of cholinergic signalling via nicotine, loss of function via nicotinic antagonist, targeted loss of function using α 7nAChR deletion in *Nes-Cre^{ERT2}*- or *LepR-Cre*-targeted HSC niche cells (but not in haematopoietic cells) all strongly convey the same conclusion: that this newly discovered cholinergic signalling is required to maintain HSC quiescence under stress” (P.11, L.235-240).

MINOR CONCERNS.

METHODS.

Line 481. ‘SEM’ The authors have indicated they are now using SD throughout including in Figure legends. Could authors please edit the term ‘SEM’ in methods line 481 if no longer correct.

Thanks for this comment, we have indicated individual points as frequently as possible following the journal’s policy and have used SEM to indicate the variability as customary.

RESULTS

1))Number of mice per group.

Many of the experiments involve only n=3 per group. This includes Figure 1, Figure 2 c,e, 3d, e, 6 b,c and perhaps others. There is also no confirmation that experiments have been repeated to ensure reproducibility in the text. From this perspective it would be recommended to show the data (in most figures) as individual dots (one dot per individual mouse) in order to help readers appreciate the inherent variability of the data and the actual (often low) numbers of biological replicates that have been used to reach these conclusions.

All figures have now been changed to display the data as individual dots unless the data cannot be comprehensively displayed (cell cycle, response to 5-FU). Let me kindly remind the reviewer that we have repeated all long-term serial transplantation experiments using sorted HSCs instead of total BM cells during the extensive revision of this study. The new serial competitive long-term transplantation experiments performed have independently confirmed our initial results (which have now been removed from the manuscript since they are redundant) using BM cells (instead of purified HSCs, as we show in the revised manuscript). We believe that this complete repetition of experiments using an alternative strategy and leading to the robust observation of identical results should be convincing enough.

Figure 3a. RE. The description of these findings are that ‘*Gfra2*^{-/-} mice show an ‘augmented haematopoietic recovery’ post 5-FU. (line 115). This figure (3a) rather suggests that each of the mouse genotypes had a similar starting leukocyte count of 10- 13mill per mL blood prior to chemotherapy and that each recovered back to that level at approximately the same time (around day 12 post 5-FU) thus no ‘augmented recovery’ by any one group. What is observed though is that *Gfra2*^{-/-} mice appear to then ‘over-shoot’ with blood leukocytes reaching over 20million/mL blood, or levels double that of recovered wildtype control or *Chrna7*^{-/-} mice. Instead of ‘augmented recovery’ these data suggest the *Gfra2*^{-/-} mice have a decreased ability to fine tune immune responses (such as not to over shoot). Please change text wording to reflect the data.

The manuscript has been edited to reflect this comment. We wrote in the revised abstract that “lack of cholinergic innervation impairs balanced responses to chemotherapy or irradiation and reduces HSC quiescence and self-renewal”. We clarified in the revised Results that “*Gfra2*^{-/-} mice and *Chrna7*^{-/-} mice exhibited a decreased ability to regulate haematopoietic recovery and fine-tune immune responses acutely after 5-FU (14-18d); in contrast, haematopoiesis normalised in these mice after 20d, when stress haematopoiesis reverted to homeostasis (Fig. 3j), underscoring the role of cholinergic signals during stress haematopoiesis” (P.6-7, L.119-122).

Line 542. Figure legend 3. ...donor cells comprise 'CD45.1' Please edit this to match what is shown in associated figure outline 3f (ie. CD45.2 donors). At least edit one of these to be consistent.

As mentioned above, this data has now been removed from the manuscript and replaced with the results of transplantations using purified HSCs.

Figure 3b,c. Could authors use a different colour for mRNA level data. In this figure red is consistently WT mice (or WT-like reporters), while green is *Gfra2*^{-/-} . Thus it is very confusing that Figure 3b instead shows WT mice treated with 5-FU as green (the colour of *Gfra2*^{-/-} mice). If the colours cant be changed, please put a 'WT' on the X-axis under the ' C and 5-FU ' in order to help readers understand the sudden shift in meaning.

I have a similar comment for the data in Figure 4b,c. In the rest of the whole Figure 4 yellow, green, red indicate mouse genetic background. Oddly Figure 4b,c however uses the same colour scheme to indicate something very different. Please don't do this. Please change 4 b,c to a different colour range (even different shades of grey or blue would be better).

Apologies for the confusion generated. Colours have been updated throughout the figures to avoid confusion.

Figure 3h. Can authors explain why the % circulating *Gfra2*^{+/-} HET control cells increase suddenly at week 24 post transplant? This is as curious as why % *Gfra2*^{-/-} reconstitution drops at 24 weeks. These experiments are listed as containing 4 to 10 mice per group. Could it be that a small number of mice at the 24 week timepoint is what is generating this shift in % apparent blood reconstitution? Could authors please show data as a dot per individual mouse to aid interpretation of these data

As mentioned above, this data has now been removed from the manuscript and replaced with the results of transplantations using purified HSCs.

Reviewers' Comments:

Reviewer #2:

Remarks to the Author:

The authors continue to provide clear evidence to support of cholinergic signaling in the context of HSC quiescence.

Some of the conclusions still rely on the premise that GFRa2 is a marker for cholinergic neurons.

However, the authors have not resolved any of my concerns relating to whether GFRa2 is an appropriate marker to use to identify sympathetic cholinergic neurons in isolation of other cell types in the BM.

Reviewer #3:

Remarks to the Author:

Separating this research into two separate manuscripts is welcome has greatly improved the narrative and rationale of each. The bone maintenance manuscript is interesting and well-written, although the poor resolution of figures in the pdf provided precluded details.

The second manuscript characterizes the role of cholinergic signaling in regulating proportion of quiescent HSC in the bone marrow/endosteum and overall impact on HSC serial reconstitution potential. Comments on the HSC manuscript are below.

Comments.

1. Manuscript title.

The manuscript title states 'cholinergic signals preserve HSC quiescence under stress' however little confirmatory data for this statement (eg. % quiescent HSC before and under stress) could be found by this reviewer. The closest is data in Supp Fig 9 (7 weeks post transplantation +/- nicotine, with no indication of the number of biological replicates presented). In order to justify title of this manuscript and the last line of abstract, at least one additional figure containing this crucial data (changes in % HSC quiescence, before and following partial irradiation and 7 days following 5FU showing impact of cholinergic signaling) is absolutely required.

2. Analysis of HSC quiescence.

Current analysis of HSC quiescence (dotplots shown in Fig 2, 8, supp 5,8) is unconvincing. The G0/G1 boundary seems to have been drawn arbitrarily at different levels depending on sample. This impacts the whole interpretation of the data. The G0 population is not really distinct from G1. A simple reanalysis may improve this, otherwise repeat experiments and/or additional data showing BrdU incorporation should be shown/performed. Could the authors please include their Ki67 isotype control stain? which is a must for these types of assays.

For reanalysis, Ki67 expression should be shown using biexponential scale to more clearly distinguish G0 from G1. The gate defining G0/G1 boundary should also remain constant within a group. If the current problem (constant realignment of gates for each sample within each group) derives from an unequal input cell number per stain (meaning unequal staining intensity), then the correct location of gates for each sample should be objectively made (by a blinded user) by gating and analysis of: a) cells in Lin- Sca+ Kit- gate which are all in G0, compared to ii) cells in Lin- Sca+ Kit+ CD48+ gate which are nearly all in G1. Only once these gates have been set, should the HSC be gated and % HSC quiescence reanalysed. These steps would need to be repeated for each individual mouse in a blinded fashion. It would be helpful to include a supplementary figure showing both the LSK CD48-CD150+ and LSK Flt3- gating used for HSC (currently not shown anywhere), plus the Ki67/DNA content stain in each of these technical confirmation gates (L-S+K- compared to L-S-K+ CD48+) for setting G0 and

G1 boundary. Any repeat experiments could also include confirmatory BrdU staining, ideally as pulse chase for label retaining cells (with ~70 day chase), or if too onerous, then as % BrdU negative (non-cycling) HSC after 3 to 5 days BrdU administration in mouse prior to euthanasia .

3. Although there is an association between increased HSC quiescence and increased HSC self-renewal (or serial reconstitution potential), these properties are distinct and may not always overlap. In some places the manuscript text appears to use these terms interchangeably. One example is Figure 4 title 'Cholinergic deficient niche renders endosteal HSCs more proliferative' . No HSC proliferation (or even cell cycle) data are shown in this figure. Instead this figure contains data on blood reconstitution of recipient mice following transplantation of 500 sorted HSC. Please modify title to more correctly reflect content and please consider the precision of wording used throughout text.

Other comments.

Figure 1 g,h,d regarding relative enrichment of ChAT-GFP+ cells 2 weeks post transplant. At two weeks post irradiation (transplantation) the bone marrow is still depleted of haematopoietic progenitor cells .

-Could the relative enrichment of the mesenchymal cell populations shown in Fig 1 simply be due to the depletion of most hematopoietic cells at this timepoint post transplant conditioning?

-Would it be possible to confirm these surprising data (strong increase in ChAT expressing cells at 2 weeks post conditioning) by qRT-PCR for ChAT on sorted cells? Is it possible for Acetylcholine to be quantified (such as by mass spec) in BM fluids at 0, 2 ,4 weeks post-transplant/conditioning or 7 days post 5FU to for further interpretation of these observations?

Figure 3j. Blood Leukocyte recovery post 5-FU. The increase in blood leukocytes at days 14-18 post 5-FU in *Gfra2*^{-/-} and *Chrna*^{-/-} mice (compated to WT mice) is very interesting.

-What is happening to bone marrow HSC (number and quiescence) at this timepoint. What happens to CXCL12 (ng per unit BM fluid) and nicotinic receptors at this time point ?

Figure 4 (primary transplant) and Figure 6 (secondary transplant). The difference in reconstitution potential in a *Gfra2*^{-/-} compared to wildtype receipient (orange vs blue lines) is impressive.

For the secondary transplant 500 BM HSC were sorted for transplant. As this means the authors have this data, could the number of phenotypic HSC in the BM of primary recipient mice 16 weeks post transplant also be included as an additional figure. It will be very interesting to observe if differences in reconstitution potential are reflected in difference in original (primary transplanted) BM phenotypic HSC numbers.

The use of only n=3-4 mice per group for these type of transplants (where large variation within groups are common) is an extremely low number. Could some more indication (individual dots) be provided of the actual % donor derived cells for each mouse in each group? This could be an additional supp figure for single timepoint, however the use of SEMs in these figures may mask a potential large variation within the mice of each group (as is normal in these type of experiments). The Anova performed (Sidaks comparison) also is likely based on the assumption these data are parametrically distributed. Perhaps a discussion with statistician regarding analysis of this data and others in manuscript regarding use of SEM and analysis may be warranted.

Figure 5. RNA seq analysis. These data are very interesting. A volcano plot with key mRNAs highlighted for readers would be a welcome visual addition. In addition this RNA-seq on sorted HSC (from WT and *Gfra2*^{-/-} mice) appears to validate the cell cycle/quiescence flow cytometry data.

Figure 9. Lovely observation. Could it be also possible that the delayed platelet normalization in patients post allogeneic HSC transplant is not simply due to nicotine-mediated alterations in HSC quiescence regulation, but also due to other differences such as the increased vascular damage

observed in smokers etc. Either way it is an interesting observation.

Figure 10. Graphical overview is not as intuitive as it could be. Without reading the manuscript this reviewer would not gain much from glancing at this current figure. I would not know where the ACh maybe expressed or where AChR expressing cells locate. Could the words/acronyms mentioned in Figure 9 legend also be the same as shown in the Figure (eg. please show the 'BMSC' and 'HSPC' in their respective places visually on the figure).

Line 86, text refers to Supp Fig 2 d-e. These don't exist.

REVIEWERS' COMMENTS

Reviewer #2 (Remarks to the Author):

The authors continue to provide clear evidence to support of cholinergic signaling in the context of HSC quiescence.

Some of the conclusions still rely on the premise that GFR α 2 is a marker for cholinergic neurons.

However, the authors have not resolved any of my concerns relating to whether GFR α 2 is an appropriate marker to use to identify sympathetic cholinergic neurons in isolation of other cell types in the BM.

GFR α 2 is a well-established marker of cholinergic neurons, since binding of the neurotrophic factor Neurturin (Heuckeroth et al., 1999) to GFR α 2 (Hiltunen and Airaksinen, 2004; Rossi et al., 1999) promotes the development and survival of cholinergic neurons (parasympathetic or sympathetic, but not noradrenergic). To further validate GFR α 2 as a marker for cholinergic neurons in the BM, we performed genetic lineage tracing of neuroglial cells in *Wnt1-Cre2* mice and quantified *Wnt1-Cre2*-traced GFR α 2⁺ cholinergic nerve fibres (Fig. 1b and Supplementary Fig. 1). In addition, our conclusions do not rely solely on GFR α 2 as a marker of cholinergic neurons. A variety of other well-established genetic tracing models to label cholinergic cells (*B6.129S-Chat^{tm1(cre)Lowl}/MwarJ* mice (*Chat-IRES-Cre*) (Stock number 031661), *ChATBAC-eGFP* (Stock number 007902), BMSCs (*Nes-gfp⁶¹*, *B6.129(Cg)-Lep^{tm2(cre)Rck/J}* (Stock number 008320), C57BL/6-Tg(*Nes-cre/ERT2*)KEisc/J (Stock number 016261), or to dissect the role of cholinergic (*B6.129S7-Chrna7^{tm1Bay}/J* (Stock number 003232, *α 7nAChRflox* (Stock number 026965)), adrenergic (*FVB/N-Adrb3tm1Lowl/J* (Stock number 006402) or neurotrophic *B6;129X1-Nrtn^{tm1Jmi}/J* (Stock number 012238) signals have been used, together with other validated markers for flow cytometry and immunofluorescence. Additional experimental evidence provided previously include:

- Transplantation of *VavCre;Chrna7-floxed* BM into WT did not affect endosteal or central BM HSPC proliferation (Supplementary Fig. 5e-f), highlighting the key importance of the niche.
- To complement the GFR α 2 transplantation studies, we have performed WT transplantations with the addition of nicotinic antagonist hexamethonium bromide, resulting in reduced HSCs in G0 phase of the cell cycle .

Additionally, we have performed new experiments that further confirm our previous contention. "Importantly, α 7nAChR deletion in either Leptin-receptor-Cre-targeted HSC niche cells²², which overlap with *Nes-GFP⁺* BMSCs^{23, 24} or *Nes-Cre^{ERT2}*-targeted HSC niche cells, similarly increased endosteal (not central) BM HSC proliferation (Fig. 3m and Supplementary Fig. 5)" (P.7 2nd para.).

Furthermore, we provided as supporting material for review a related manuscript under review elsewhere which has characterised the sympathetic origin and bone maintenance functions of cholinergic fibres. The extensive evidence provided in that study shows that early postnatally, a subset of sympathetic nerve fibers undergoes an Interleukin-6-induced cholinergic switch upon contacting the bone. A neurotrophic dependency between GFR α 2 and its ligand, Neurturin, is established between sympathetic cholinergic fibers and bone-embedded osteocytes, which require this cholinergic innervation for their survival and connectivity. Bone-lining osteoprogenitors amplify and propagate cholinergic signals in the bone marrow. Moderate exercise augments trabecular bone

partly through an IL-6-dependent expansion of sympathetic cholinergic nerve fibers. Consequently, loss of cholinergic skeletal innervation reduces osteocyte survival and function, causing osteopenia and impaired skeletal adaptation to moderate exercise. These results uncover a cholinergic neuro-osteocyte interface that regulates skeletogenesis and skeletal turnover through bone-anabolic effects.

These conclusions are supported by the evidence provided in the related manuscript, including: 1) cholinergic nerve fibers being the main source of NRTN near bone (Figure 6K); 2) NRTN directly promoting osteocyte survival (Figure 6I-J) in GFR α 2- and RET-expressing osteocytes (Figure 6D-E); 3) treatment with GDNF or NRTN, alone or combined with their soluble receptors, improves growth and survival in WT primary OBs, while NRTN's trophic effect is reduced in Gfra2 $^{-/-}$ primary OBs (Figure 6I-J); 4) MLO-Y4 osteocyte-like cells (Kato et al., 1997) similarly exhibit reduced apoptosis upon GFR treatment (Figure S6F-G), suggesting a trophic effect of NRTN produced by the cholinergic fibers on GFR α 2-expressing osteocytes; 5) osteocyte numbers are reduced and atrophic in Nrtn KO mice (Figure 6O-Q) or after neonatal sympathectomy of cholinergic fibers (Figure 6L-N), but not after adult sympathectomy of noradrenergic fibers (Figure S6H-K); 6) bone adaptation to moderate exercise is impaired in cholinergic-neural-deficient mice (Figure 7L) or in rats, after neonatal sympathectomy of cholinergic fibers (Figure S7F-I); 7) deficient bone-anabolic responses in cholinergic-neural-deficient mice are explained by the incapacity of osteocytes to repress sclerostin, which is a key inhibitor of bone-anabolic Wnt signalling (Figures 6D-E and S6E); and 8) the key role of deregulated sclerostin in the absence of sympathetic cholinergic fibers is demonstrated by the rescue of osteopenia and bone strength in GFR α 2 KO mice treated with sclerostin-blocking antibody (Figure 6F-H).

Reviewer #3 (Remarks to the Author):

Separating this research into two separate manuscripts is welcome and has greatly improved the narrative and rationale of each. The bone maintenance manuscript is interesting and well-written, although the poor resolution of figures in the pdf provided precluded details.

We thank the reviewer very much for considering that an increased focus on haematopoietic vs skeletal regulation by cholinergic signals has improved the narrative and accessibility of the study. We are also thankful for considering the related bone maintenance manuscript interesting and we are sorry to learn that the reviewer could not download the high resolution images that were embedded in the PDF of the related manuscript for review (bone maintenance). The PDF was necessarily compressed but all the figures had a message (click here to access/download) in the upper right corner with a hyperlink to the high resolution images.

The second manuscript characterizes the role of cholinergic signaling in regulating proportion of quiescent HSC in the bone marrow/endosteum and overall impact on HSC serial reconstitution potential. Comments on the HSC manuscript are below.

Comments.

1. Manuscript title.

The manuscript title states 'cholinergic signals preserve HSC quiescence under stress' however little confirmatory data for this statement (eg. % quiescent HSC before and under stress) could be found by

this reviewer. The closest is data in Supp Fig 9 (7 weeks post transplantation +/- nicotine, with no indication of the number of biological replicates presented). In order to justify title of this manuscript and the last line of abstract, at least one additional figure containing this crucial data (changes in % HSC quiescence, before and following partial irradiation and 7 days following 5FU showing impact of cholinergic signaling) is absolutely required.

All experiments are performed in the context of stress or emergency haematopoiesis, i.e following irradiation and transplantation, since the sympathetic nervous system has been evolutionary selected as a stress response system. Thus, we have demonstrated the role of the cholinergic signals in regulating stress/emergency/regenerative haematopoiesis by comparing control mice with either loss or gain of function cholinergic models in different settings of stress haematopoiesis (after myeloablation, or irradiation and transplantation).

2. Analysis of HSC quiescence.

Current analysis of HSC quiescence (dotplots shown in Fig 2, 8, supp 5,8) is unconvincing. The G0/G1 boundary seems to have been drawn arbitrarily at different levels depending on sample. This impacts the whole interpretation of the data. The G0 population is not really distinct from G1. A simple reanalysis may improve this, otherwise repeat experiments and/or additional data showing BrdU incorporation should be shown/performed. Could the authors please include their Ki67 isotype control stain? which is a must for these types of assays.

For reanalysis, Ki67 expression should be shown using biexponential scale to more clearly distinguish G0 from G1. The gate defining G0/G1 boundary should also remain constant within a group. If the current problem (constant realignment of gates for each sample within each group) derives from an unequal input cell number per stain (meaning unequal staining intensity), then the correct location of gates for each sample should be objectively made (by a blinded user) by gating and analysis of: a) cells in Lin- Sca+ Kit- gate which are all in G0, compared to ii) cells in Lin- Sca+ Kit+ CD48+ gate which are nearly all in G1. Only once these gates have been set, should the HSC be gated and % HSC quiescence reanalysed. These steps would need to be repeated for each individual mouse in a blinded fashion. It would be helpful to include a supplementary figure showing both the LSK CD48-CD150+ and LSK Flt3- gating used for HSC (currently not shown anywhere), plus the Ki67/DNA content stain in each of these technical confirmation gates (L-S+K- compared to L-S-K+ CD48+) for setting G0 and G1 boundary. Any repeat experiments could also include confirmatory BrdU staining, ideally as pulse chase for label retaining cells (with ~70 day chase), or if too onerous, then as % BrdU negative (non-cycling) HSC after 3 to 5 days BrdU administration in mouse prior to euthanasia .

Thanks for highlighting the best procedure to perform cell cycle studies, which we have followed to the best of our abilities. Some representative examples shown before were derived from different experiments where the gates need to be necessarily adjusted given the known variability of these assays. The FACS plots have been updated in the revised figures to include samples from the same experiment and consistent gating strategy. The ki67 isotype control stain has been included in Supplementary Fig 4d.

3. Although there is an association between increased HSC quiescence and increased HSC self-renewal (or serial reconstitution potential), these properties are distinct and may not always overlap. In some

places the manuscript text appears to use these terms interchangeably. One example is Figure 4 title 'Cholinergic deficient niche renders endosteal HSCs more proliferative'. No HSC proliferation (or even cell cycle) data are shown in this figure. Instead this figure contains data on blood reconstitution of recipient mice following transplantation of 500 sorted HSC. Please modify title to more correctly reflect content and please consider the precision of wording used throughout text.

The manuscript has been altered to reflect this comment. Figure 4 Legend has been revised as "Cholinergic-deficient niche compromises HSC quiescence, leading to increased reconstitution of primary recipients".

Other comments.

Figure 1 g,h,d regarding relative enrichment of ChAT-GFP+ cells 2 weeks post transplant. At two weeks post irradiation (transplantation) the bone marrow is still depleted of haematopoietic progenitor cells. -Could the relative enrichment of the mesenchymal cell populations shown in Fig 1 simply be due to the depletion of most hematopoietic cells at this timepoint post transplant conditioning?

The data is presented as the abundance of *ChAT*-GFP-positive within each stromal cell population, and is therefore not influenced by reduced bone marrow haematopoietic cells following irradiation. In addition, when comparing the actual number of *ChAT*-GFP+ cells following transplantation, the data indicates that the population expands, as opposed to the population being more resistant to the effects of irradiation (New Supplementary Fig 1c,d).

-Would it be possible to confirm these surprising data (strong increase in ChAT expressing cells at 2 weeks post conditioning) by qRT-PCR for ChAT on sorted cells? Is it possible for Acetylcholine to be quantified (such as by mass spec) in BM fluids at 0, 2, 4 weeks post-transplant/conditioning or 7 days post 5FU to for further interpretation of these observations?

We had provided measurements of acetylcholine levels using ELISA (Fig. 1a), which is a validated method.

In the related manuscript provided as materials for review, we provided measurements of acetylcholine levels in sorted osteolineage cells (Fig. 4H).

As suggested, we have performed qRT-PCR for ChAT on sorted osteolineage cells at steady state (New Supplementary Fig. 1e, reproduced here as Figure R1). The results correlate well with the flow cytometry data from *ChAT-IRES-Cre;Ai14D* mice previously provided (Fig. 1h).

Figure R1. mRNA expression of *ChAT* in sorted osteolineage cells of WT mice (N=4).

Figure 3j. Blood Leukocyte recovery post 5-FU. The increase in blood leukocytes at days 14-18 post 5-FU in *Gfra2*^{-/-} and *Chrna*^{-/-} mice (compared to WT mice) is very interesting.

-What is happening to bone marrow HSC (number and quiescence) at this timepoint. What happens to CXCL12 (ng per unit BM fluid) and nicotinic receptors at this time point ?

We agree that those are interesting follow up studies for the future on the possible regulation of nicotinic receptor expression and CXCL12 regulation by cholinergic signals after 5-FU, since it is known that loss of CXCL12 alters the patterns of haematopoietic regeneration after myelosuppression, leading to faster haematologic re-establishment under 5-FU challenge (Tzeng YS et al. Loss of Cxcl12/Sdf-1 in adult mice decreases the quiescent state of hematopoietic stem/progenitor cells and alters the pattern of hematopoietic regeneration after myelosuppression. Blood 117(2):429-39 (2010).

Figure 4 (primary transplant) and Figure 6 (secondary transplant). The difference in reconstitution potential in a *Gfra2*^{-/-} compared to wildtype recipient (orange vs blue lines) is impressive. For the secondary transplant 500 BM HSC were sorted for transplant. As this means the authors have this data, could the number of phenotypic HSC in the BM of primary recipient mice 16 weeks post transplant also be included as an additional figure. It will be very interesting to observe if differences in reconstitution potential are reflected in difference in original (primary transplanted) BM phenotypic HSC numbers.

Please find in the right hand side the numbers of HSCs (2 femurs and 2 tibias) in primary recipient mice 4 months after transplantation.

The use of only n=3-4 mice per group for these type of transplants (where large variation within groups are common) is an extremely low number. Could some more indication (individual dots) be provided of the actual % donor derived cells for each mouse in each group? This could be an additional supp figure for single timepoint, however the use of SEMs in these figures may mask a potential large variation within the mice of each group (as is normal in these type of experiments). The Anova performed (Sidaks comparison) also is likely based on the assumption these data are parametrically distributed. Perhaps a discussion with statistician regarding analysis of this data and others in manuscript regarding use of SEM and analysis may be warranted.

Below are examples of the distribution at 4 weeks in both the primary (left) and secondary (right) transplantations.

Homocedasticity and normality tests have been carried out to choose suitable statistical tests. We have displayed individual dot plots for all bar charts, clearly showing the intrinsic variability of the parameters. Showing individual dot plots and SEM is an accepted and commonly used way to present this type of data.

Figure 5. RNA seq analysis. These data are very interesting. A volcano plot with key mRNAs highlighted for readers would be a welcome visual addition. In addition this RNA-seq on sorted HSC (from WT and *Gfra2*^{-/-} mice) appears to validate the cell cycle/quiescence flow cytometry data.

We agree that the RNAseq data is an independent confirmation of the cell cycle results. As suggested, we have added Volcano plots of differentially-expressed genes in HSCs from the central or endosteal BM of WT and *Gfra2*^{-/-} mice to the manuscript (new Supplementary Fig. 7m-n).

Figure 9. Lovely observation. Could it be also possible that the delayed platelet normalization in patients post allogeneic HSC transplant is not simply due to nicotine-mediated alterations in HSC quiescence regulation, but also due to other differences such as the increased vascular damage observed in smokers etc. Either way it is an interesting observation.

It is possible that other alterations induced by high nicotine or inflammation associated with smoking contribute to phenotypes. We have suggested a possible association with HSC proliferation given the results obtained with animal models, but we do not claim or imply causality.

Figure 10. Graphical overview is not as intuitive as it could be. Without reading the manuscript this reviewer would not gain much from glancing at this current figure. I would not know where the ACh maybe expressed or where AChR expressing cells locate. Could the words/acronyms mentioned in Figure 9 legend also be the same as shown in the Figure (eg. please show the 'BMSC' and 'HSPC' in their respective places visually on the figure).

Graphical overview (Figure 10) has been modified to be clearer and more visual and accessible.

Line 86, text refers to Supp Fig 2 d-e. These don't exist.

Supplementary Fig. 2d-e is cited on P.5 2nd para: "The decreased quiescence of endosteal HSCs in *Gfra2*^{-/-} mice (see Fig. 2c) was not observed in WT mice carrying *Gfra2*^{-/-} haematopoietic cells (Supplementary Fig. 2d-e), suggesting that HSC quiescence is extrinsically regulated by cholinergic signals through the microenvironment."